# Synergistic phase separation of two pathways promotes integrin clustering and nascent adhesion formation

**Lindsay B Case[1,2]\*, Milagros De Pasquale[2], Lisa Henry[1], Michael K Rosen[1]\***

[1]Department of Biophysics, Howard Hughes Medical Institute, The University of Texas Southwestern Medical Center, Dallas, United States; [2]Department of Biology, Massachusetts Institute of Technology, Cambridge, United States

**Abstract** Integrin adhesion complexes (IACs) are integrin-based plasma-membrane-associated compartments where cells sense environmental cues. The physical mechanisms and molecular interactions that mediate initial IAC formation are unclear. We found that both p130Cas ('Cas') and Focal adhesion kinase ('FAK') undergo liquid-liquid phase separation in vitro under physiologic conditions. Cas- and FAK- driven phase separation is sufficient to reconstitute kindlin-dependent integrin clustering in vitro with recombinant mammalian proteins. In vitro condensates and IACs in mouse embryonic fibroblasts (MEFs) exhibit similar sensitivities to environmental perturbations including changes in temperature and pH. Furthermore, mutations that inhibit or enhance phase separation in vitro reduce or increase the number of IACs in MEFs, respectively. Finally, we find that the Cas and FAK pathways act synergistically to promote phase separation, integrin clustering, IAC formation and partitioning of key components in vitro and in cells. We propose that Cas- and FAK-driven phase separation provides an intracellular trigger for integrin clustering and nascent IAC formation.

## Editor's evaluation

This paper identifies phase separation as underlying mechanism that imitates/contributes to the formation of Integrin-containing adhesion sites. The authors demonstrate that p130Cas and FAK undergo phase separation, concentrate kindlin and eventually integrins. The experiments and findings are well described and controlled. This paper is an important contribution to the understanding of how integrins and focal adhesion proteins cluster to form cell attachment sites.

**\*For correspondence:**
lcase@mit.edu (LBC);
michael.rosen@utsouthwestern.edu (MKR)

## Introduction

Integrin-mediated adhesion complexes (IACs) are plasma membrane-associated compartments that provide specific adhesion between cells and their surroundings (***Case and Waterman, 2015***; ***Chastney et al., 2021***). IACs serve as sites of force transmission between the actin cytoskeleton and the extracellular matrix (ECM) to drive tissue morphogenesis, cell movement, and ECM remodeling. Additionally, they serve as signaling hubs where cells sense biochemical and physical cues in their environment to regulate the cell cycle, differentiation and death. Thus, IACs mediate an array of functions involving biochemical and physical interactions between the cell and its environment.

Integrins are heterodimeric transmembrane receptors that specifically bind to ligands within the ECM or to receptors on adjacent cells (***Hynes, 2002***). Both chains of the α/β heterodimer have a large extracellular domain that imparts ligand specificity and a small cytoplasmic domain (20–50 amino acids) that mediates intracellular signaling. Since integrins lack catalytic activity and do not directly bind actin (***Hynes, 2002***), downstream signaling requires the macromolecular assembly of

integrins with cytoplasmic adaptor proteins, signaling molecules, and actin binding proteins to form IACs. Thus, molecular interactions that regulate integrin clustering and initial IAC formation control integrin-dependent signaling (*Hotchin and Hall, 1995*; *Miyamoto et al., 1995*; *Robertson et al., 2015*; *Theodosiou et al., 2016*).

IACs initially form as puncta (~120 nm diameter) within the lamellipodia, termed nascent adhesions. Nascent adhesion formation is dependent on ligand binding, integrin conformational activation, and mechanical forces imparted by retrograde actin flow (*Changede et al., 2015*; *Choi et al., 2008*). Although nascent adhesion formation and cell spreading are impaired on soft substrates (where mechanical forces are reduced), this can be rescued by inducing the integrin conformational change with $Mn^{2+}$, suggesting that during the initial steps of nascent adhesion assembly force is only required to activate integrins (*Oakes et al., 2018*). Kindlin and talin are adaptor proteins that directly bind the β integrin cytoplasmic domain (*Sun et al., 2019*). Kindlin binding is required for integrin clustering and nascent adhesion assembly, although it is not clear how kindlin orchestrates this higher order assembly (*Theodosiou et al., 2016*; *Ye et al., 2013*). Many proteins appear to simultaneously assemble into nascent adhesions, including α5β1 integrin, αVβ3 integrin, talin, kindlin, focal adhesion kinase (FAK), paxillin, and p130Cas (*Bachir et al., 2014*; *Changede et al., 2015*; *Choi et al., 2008*; *Donato et al., 2010*; *Lawson et al., 2012*; *Yu et al., 2011*). After initial formation, nascent adhesions undergo dramatic growth and compositional maturation dependent on talin, increased forces, and bundled actin filaments (*Choi et al., 2008*; *Kuo et al., 2011*; *Oakes et al., 2012*; *Schwarz and Gardel, 2012*). While IAC maturation is reasonably well understood, the specific molecular interactions that regulate initial nascent adhesion formation are unclear. In this study, we investigate how specific protein-protein interactions contribute to nascent adhesion formation.

Liquid-liquid phase separation driven by weak interactions between multivalent molecules has emerged as an important mechanism that can drive the formation of micron-sized, membraneless cellular compartments, termed 'biomolecular condensates' (*Banani et al., 2017*; *Case et al., 2019a*; *Shin and Brangwynne, 2017*). At the plasma membrane, phase separation can promote the assembly of transmembrane proteins and their cytoplasmic binding partners into dynamic, micron-sized clusters (*Banjade and Rosen, 2014*; *Beutel et al., 2019*; *Case et al., 2019a*; *Su et al., 2016*; *Zeng et al., 2016*). Phase separation is highly dependent on molecular valence (*Li et al., 2012*), and increasing the valence of interactions reduces the concentration threshold required for phase separation. Condensates often possess liquid-like material properties, can contain hundreds of molecular constituents, and form over a wide range of molecular stoichiometries (*Banani et al., 2017*; *Case et al., 2019b*). Condensate composition can vary dramatically and change rapidly in response to signals (*Banani et al., 2016*; *Markmiller et al., 2018*; *Youn et al., 2018*).

IACs are discrete, micron-sized structures that exhibit many characteristics of phase separated compartments. They are highly enriched in multivalent adaptor proteins whose valence can be modulated by phosphorylation and mechanical force (*Pellicena and Miller, 2001*; *Schiller and Fässler, 2013*; *Yao et al., 2016*), concomitant with changes in IAC size. IACs often exhibit liquid-like material properties such as rapid constituent exchange (*Supplementary file 1*; *Lavelin et al., 2013*; *Pasapera et al., 2010*; *Stutchbury et al., 2017*) and the ability to fuse (*Berginski et al., 2011*; *Changede et al., 2015*). IACs contain hundreds of different proteins (*Schiller and Fässler, 2013*), are stoichiometrically undefined (*Bachir et al., 2014*), and their composition can change dramatically under different cellular conditions (*Horton et al., 2015*; *Kuo et al., 2011*; *Schiller and Fässler, 2013*). Because of their properties and molecular composition, we hypothesized that multivalent oligomerization and phase separation of IAC-associated proteins might contribute to integrin clustering and nascent adhesion assembly.

Here, we investigated biochemical interactions among the earliest proteins to arrive at nascent adhesions. We found that several of these undergo liquid-liquid phase separation when combined in vitro at physiologic concentrations and under physiologic buffer conditions. Multivalent interactions between phosphorylated p130Cas ('pCas'), Nck and N-WASP are sufficient for phase separation. 'FAK' also phase separates, and multivalent interactions between FAK and paxillin enhance this behavior. pCas and FAK undergo phase separation synergistically to form droplets that more strongly concentrate kindlin, and consequently integrin. Using a novel experimental platform to reconstitute β1 integrin clustering on supported phospholipid bilayers, we found that kindlin likely plays a central role in nascent adhesion assembly by coupling the pCas and FAK pathways to integrins. In vitro

condensates and cellular IACs exhibit similar sensitivity to temperature and pH. Further, mutations in Cas or FAK that inhibit phase separation in vitro reduce the number IACs in cells, while paxillin mutations that enhance phase separation in vitro increase the number of IACs in cells. Finally, we find that the Cas and FAK pathways act synergistically to promote phase separation, integrin clustering, nascent adhesion formation and partitioning of key components in vitro and in cells. We propose that pCas- and FAK-driven phase separation provides an intracellular trigger for integrin clustering and nascent adhesion formation.

## Results

### Multivalent interactions promote phase separation of p130Cas under physiologic conditions

The adaptor protein Cas (p130Cas/BCAR1) is present in nascent adhesions (*Donato et al., 2010*) and has been implicated in promoting IAC formation (*Meenderink et al., 2010*). Cas contains an N-terminal SH3 domain, a central disordered 'substrate' domain, a Src-binding domain, and a C-terminal Cas-family homology domain (*Figure 1—figure supplement 1*; *Meenderink et al., 2010*). Within the substrate domain, there are 15 YXXP motifs that, when phosphorylated, can bind to the SH2 domain-containing adaptor proteins, Crk-II and Nck (*Iwahara et al., 2004*; *Pellicena and Miller, 2001*; *Schlaepfer et al., 1997*). In cells, Cas is robustly phosphorylated on 10 of its YXXP motifs, and proper cell migration requires a minimum of four phosphorylation sites (*Shin et al., 2004*). Crk-II contains two SH3 domains which can bind adaptor proteins such as SOS and C3G that contain multiple proline-rich motifs (PRMs) (*Birge et al., 2009*). Similarly, Nck contains three SH3 domains which can bind adaptor proteins such as N-WASP, SOS, and Abl that contain multiple PRMs (*Li et al., 2001*). In other signaling pathways, phosphorylation of proteins on multiple tyrosine residues and subsequent binding by multivalent adaptor proteins can promote phase separation (*Banjade and Rosen, 2014*; *Kim et al., 2019*; *Su et al., 2016*). For example, multivalent interactions between phosphorylated Nephrin, Nck, and N-WASP or phosphorylated Lat, Grb2 and SOS drive phase separation to form droplets in solution and clusters at membranes (*Banjade and Rosen, 2014*; *Banjade et al., 2015*; *Kim et al., 2019*; *Li et al., 2012*; *Su et al., 2016*). Thus, we sought to determine if the multiply-phosphorylated Cas substrate domain could promote phase separation, similar to phosphorylated Nephrin and Lat. Nck has been localized within IACs (*Goicoechea et al., 2002*; *Horton et al., 2015*), N-WASP colocalizes with Cas within the lamellipodia where nascent adhesions form (*Zhang et al., 2014*), and both Nck and N-WASP have been implicated in regulating cell adhesion to fibronectin (*Misra et al., 2007*; *Ruusala et al., 2008*). Thus, we chose to use Nck as the SH2/SH3 domain containing adaptor protein and N-WASP as the PRM containing adaptor protein for our studies (*Figure 1a*, *Supplementary file 2*). To assess phase separation in vitro, we purified recombinant proteins (*Figure 1—figure supplement 2*), phosphorylated Cas on tyrosine to an average of ~17–19 sites/molecule (pCas, *Figure 1—figure supplement 3*), and assessed phase separation by both measuring solution turbidity (Absorbance at 350 nm) and visualizing droplets with fluorescence microscopy. For microscopy measurements, we conjugated specific proteins with Alexa fluorophores, as indicated in figure legends. Although purified pCas is more highly phosphorylated than in cells, the valency for Nck is likely similar (five of the six Nck-binding motifs are robustly phosphorylated in cells) (*Shin et al., 2004*). For all in vitro experiments, we used buffer containing 50 mM HEPES (pH 7.3), 50 mM KCl, 1 mM TCEP, and 0.1% BSA. No crowding agents were used in any experiments. In mammalian cells, the concentration of Nck is ~200 nM, that of N-WASP (and its homolog, WASP) ranges from 150 nM to 10 µM, and that of Cas is ~70 nM (*Supplementary file 3*, *Hein et al., 2015*; *Higgs and Pollard, 2000*; *Isaac et al., 2010*; *Roybal et al., 2016*). We combined Nck and N-WASP at physiologic concentrations (200 nM Nck/1 µM N-WASP and 500 nM Nck/1 µM N-WASP) and titrated increasing concentrations of unphosphorylated or phosphorylated Cas. We specifically observed an increase in solution turbidity at physiologic concentrations (10–200 nM) of pCas (*Figure 1b*). Spinning disk confocal fluorescence microscopy confirmed that the increase in solution turbidity was due to formation of condensed foci at physiologic concentrations (*Figure 1—figure supplement 4*). We increased protein concentrations to 1 µM to ensure that the foci were larger than the point spread function of the microscope, revealing that they are spherical (*Figure 1c*) and suggesting that they behave as liquid droplets. We measured the partition coefficient (PC) of molecules into the droplets (Intensity inside droplet/[Intensity in bulk

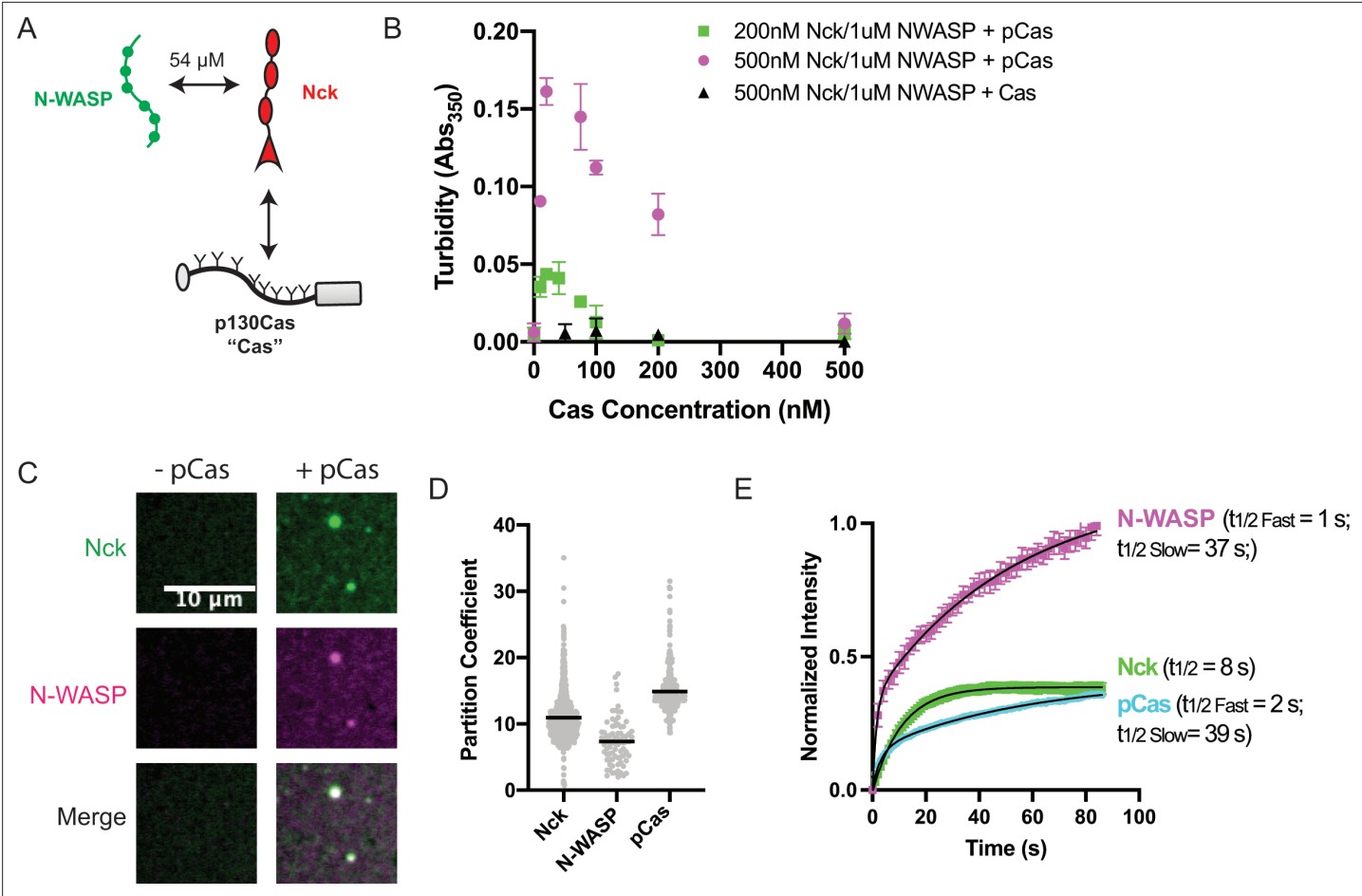

**Figure 1.** p130Cas, Nck, and N-WASP undergo liquid-liquid phase separation. (**A**) Molecular interactions of IAC proteins, $K_D$ values indicated where known. Details and references in *Supplementary file 1*. (**B**) Solution turbidity measurements. Nck (200 nM, green; or 500 nM, magenta+ black) and N-WASP (1 μM) were combined with increasing concentrations of phosphorylated Cas (pCas, green+ magenta) or unphosphorylated Cas (Cas, black). Each point represents the mean ± SEM of three independent measurements. (**C**) Spinning disk confocal fluorescence microscopy images of droplets. Nck (1 μM, 15% Alexa568 labeled) and N-WASP (1 μM, 15% Alexa647 labeled) were combined ± pCas (1 μM, unlabeled). (**D**) Quantification of constituent partitioning into droplets. Each grey point represents an individual measurement, and the mean indicated by black line. Each condition contains at least 75 measurements from two or more independent experiments. (**E**) Fluorescence Recovery After Photobleaching (FRAP) measurements of droplets. Droplets formed from 1 μM each of Nck (15% Alexa568 labeled), N-WASP (15% Alexa647 labeled) and pCas (5%–647 labeled). Each point represents the mean ± SEM of at least six independent measurements. Recovery curves were fit with a single exponential (Nck) or biexponential (pCas, N-WASP) model and the fits are overlayed on the graph (black line). Detailed fit information in *Supplementary file 4*. All scale bars = 10 μm.

The online version of this article includes the following source data and figure supplement(s) for figure 1:

**Figure supplement 1.** Domain organization of proteins used in this study.

**Figure supplement 2.** Purification of recombinant integrin adhesion complex proteins.

**Figure supplement 2—source data 1.** Uncropped gel images and unprocessed.tif files from *Figure 1—figure supplement 2*.

**Figure supplement 3.** Intact mass spectrometry of p130Cas proteins.

**Figure supplement 4.** Droplets form with physiological protein concentrations.

**Figure supplement 5.** Measuring the point spread function (PSF).

**Figure supplement 6.** Representative fluorescence recovery after photobleaching (FRAP) data.

solution]; *Figure 1d*, *Figure 1—figure supplement 5*, see Materials and methods) and performed fluorescence recovery after photobleaching (FRAP) analysis (*Figure 1—figure supplement 6*). We found that pCas, Nck and N-WASP rapidly exchanged between droplets and bulk solution (*Figure 1e*; *Supplementary file 4*; pCas $t_{1/2 fast}$ = 2 s; pCas $t_{1/2 slow}$ = 39s; Nck $t_{1/2}$ = 8 s; N-WASP $t_{1/2 fast}$ = 1 s; N-WASP $t_{1/2 slow}$ = 37 s), although a fraction of Nck and pCas molecules did not recover in the 100 s timeframe of

the experiment, suggesting a slower phase of recovery and/or an immobile fraction (*Supplementary file 4*). In cells, IAC-associated proteins, including Cas, often contain a population of fast exchanging molecules ($t_{1/2} < 30$ s) as well as an immobile fraction (up to 50% immobile for some proteins; *Supplementary file 1*). Thus, similar to cellular IACs, droplets exhibit liquid-like material properties with some solid-like elements as well. We conclude that interactions between multiply-phosphorylated Cas, Nck, and N-WASP are sufficient to promote liquid-liquid phase separation at physiologic protein concentrations.

## FAK and paxillin phase separate under physiologic conditions

Like many condensates, IAC assembly is robust to changes in composition. Although Cas knockout (KO) fibroblasts still form IACs, they exhibit decreased adhesion assembly rates (*Meenderink et al., 2010*). These observations suggest a potential role for Cas in nascent adhesion assembly that is at least partially compensated for by additional proteins (*Honda et al., 1998*). Thus, we sought to identify additional interactions that may promote integrin clustering at IACs, perhaps through phase separation (*Figure 2a*, *Supplementary file 2*). Talin, kindlin, paxillin, and FAK regulate nascent adhesion assembly, are present in nascent adhesions, and rapidly exchange between IACs and the cytoplasm (*Supplementary file 1*; *Bachir et al., 2014*; *Changede et al., 2015*; *Choi et al., 2008*; *Meenderink et al., 2010*; *Swaminathan et al., 2016*; *Theodosiou et al., 2016*). Full-length talin is autoinhibited (*Dedden et al., 2019*), but the talin head domain (talinH) is sufficient to rescue nascent adhesion formation in talin depleted cells (*Changede et al., 2015*). We expressed and purified recombinant talinH, kindlin, paxillin, and FAK (*Figure 1—figure supplement 2*) and screened for phase separation in vitro by measuring solution turbidity. We observed an increase in solution turbidity with increasing concentrations of FAK, starting at ~100 nM, but not with increasing concentrations of talinH, kindlin, or paxillin (*Figure 2b*, *Figure 2—figure supplement 1a*). Note that we purify FAK in buffer containing 300 mM NaCl, preventing self-assembly prior to dilution into the experimental buffer. Using spinning disk confocal fluorescence microscopy, we observed small FAK foci at 40 nM concentration (*Figure 2—figure supplement 2a-b*), the upper end of FAK concentrations in mammalian cells (5–40 nM) (*Brami-Cherrier et al., 2014*; *Hein et al., 2015*; *Supplementary file 3*). With 1 µM FAK (1% Alexa-647-labeled), we observe larger, micron-scale spherical droplets (*Figure 2c*) that exchange molecules with bulk solution as assessed by FRAP (*Figure 2g*. *Figure 2—figure supplement 3a*; *Supplementary file 4*; $t_{1/2\ fast}$ = 6 s; $t_{1/2\ slow}$ = 56 s), although a fraction of molecules (38%) did not recover. Thus, droplets exhibit liquid-like material properties, albeit with some solid-like elements as well. We conclude that FAK undergoes liquid-liquid phase separation in vitro under physiologic conditions.

## Paxillin enhances FAK phase separation

Next, we sought to identify additional interactions that might enhance FAK phase separation. FAK can interact directly with paxillin and talin (*Lawson et al., 2012*; *Thomas et al., 1999*). Thus, we added increasing concentrations of paxillin, talinH, or kindlin to 250 nM FAK. We found that paxillin, but not talinH or kindlin, increased solution turbidity (*Figure 2d*). Turbidity of a solution containing 250 nM FAK and paxillin was not increased by addition of talinH or kindlin (*Figure 2—figure supplement 1b*). When paxillin (5% Alexa546) and FAK (1% Alexa647) were combined at 1 µM concentration, micron-sized spherical droplets containing both proteins were formed (*Figure 2e*). In the presence of equimolar paxillin, FAK enrichment in the droplets increased twofold (*Figure 2f*). Both FAK and paxillin exchange between droplets and bulk solution in FRAP analyses (*Figure 2g*, *Figure 2—figure supplement 3b-c*; *Supplementary file 4*); FAK $t_{1/2\ fast}$ = 10 s; FAK $t_{1/2\ slow}$ = 98 s; paxillin $t_{1/2\ fast}$ = 7 s, paxillin $t_{1/2\ slow}$ = 78 s, although FAK recovery is slowed relative to droplets containing FAK alone. Thus, similar to cellular IACs (*Supplementary file 1*), droplets formed with FAK and paxillin exhibit predominantly liquid-like material properties with some solid-like elements. Finally, we found that paxillin enhances droplet formation even at low nanomolar concentrations of FAK and paxillin (*Figure 2—figure supplement 2c-d*). We conclude that interactions with paxillin enhance FAK phase separation.

## The pCas and FAK pathways phase separate synergistically

Our results suggest that pCas-dependent and FAK-dependent phase separation can promote higher order assembly of IAC-associated proteins. Furthermore, pCas directly interacts with both paxillin and

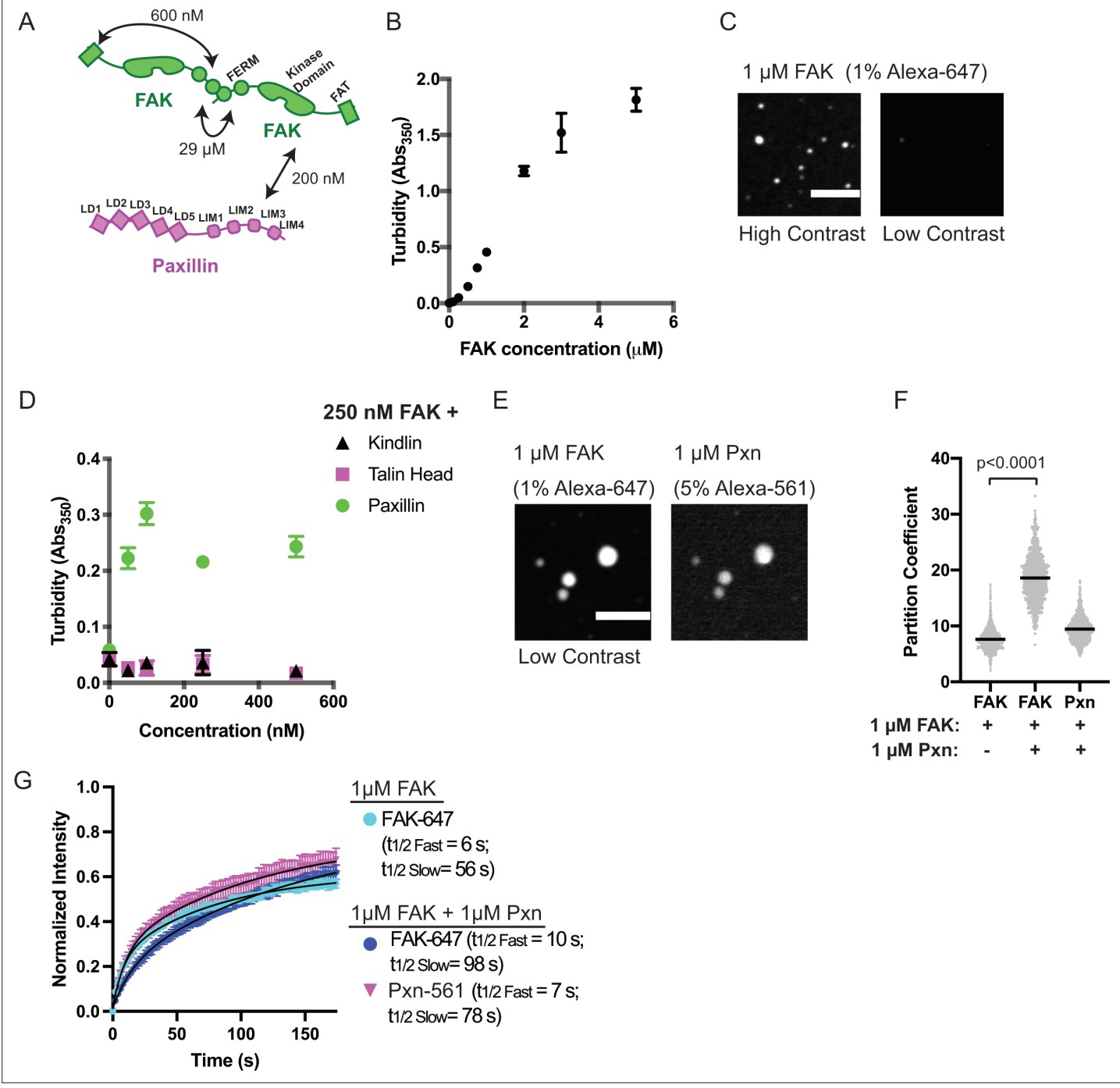

**Figure 2.** FAK and paxillin undergo liquid-liquid phase separation. (**A**) Molecular interactions of IAC proteins, $K_D$ values indicated where known. Details and references in ***Supplementary file 1***. (**B**) Solution turbidity measurements with increasing concentrations of FAK. (**C**) Spinning disk confocal fluorescence microscopy images of droplets formed from 1 µM FAK (1% Alexa647 labeled). The same image is displayed with two different contrast settings for comparison to panel E. (**D**) Solution turbidity measurements. 250 nM FAK was combined with increasing concentrations of talinH (magenta), kindlin (black), or paxillin (green). (**E**) Spinning disk confocal fluorescence microscopy images of droplets formed from 1 µM FAK (1% Alexa647 labeled) and 1 µM FAK paxillin (5% Alexa546 labeled). For comparison, contrast settings of FAK image are identical to those in 2 C, low contrast. (**F**) Quantification of constituent partitioning into droplets. Each condition contains at least 750 measurements from four or more independent experiments. Significance tested with student's t-test. (**G**) Fluorescence Recovery After Photobleaching (FRAP) measurements of droplets. Droplets formed from 1 µM of FAK (1% Alexa647 labeled, cyan) or 1 µM each of FAK (1% Alexa647 labeled, dark blue) and paxillin (5% Alexa546 labeled, magenta). Each point represents the mean ± SEM of at least 15 independent measurements. Recovery curves were fit with a biexponential model and the fits are overlayed on the graph (black line). Detailed fit information in ***Supplementary file 4***. In (**B**) and (**D**), each point represents the mean ± SEM of three independent

*Figure 2 continued on next page*

*Figure 2 continued*

measurements. In (**F**), each gray point represents an individual measurement, and the mean is indicated by black line. All scale bars = 10 µm.

The online version of this article includes the following figure supplement(s) for figure 2:

**Figure supplement 1.** Solution turbidity measurements.

**Figure supplement 2.** Droplets form with physiological protein concentrations.

**Figure supplement 3.** Representative fluorescence recovery after photobleaching (FRAP) data.

FAK (*Wisniewska et al., 2005*; *Zhang et al., 2017*) and Cas, FAK, and paxillin have been observed to associate as a complex in the cytoplasm (*Hoffmann et al., 2014*), suggesting these two pathways could function together to regulate IAC formation (*Figure 3a*, *Figure 1—figure supplement 1*, *Supplementary file 2*). Thus, we sought to better understand the relationship between pCas- and FAK- driven phase separation in vitro. When we combined 1 µ M each of pCas, Nck, N-WASP, FAK, and paxillin ('pCas + FAK mix'), we observed a single class of droplets containing all molecules (*Figure 3b*). Moreover, partitioning of all molecules into droplets increased two- to fivefold with the pCas + FAK mix (*Figure 3c* vs *Figures 1d and 2f*). Next, we combined lower concentrations of proteins (either 250 nM of each or the cytoplasmic concentrations (*Supplementary file 3*)) and measured solution turbidity in buffers with increasing concentrations of salt. Turbidity was observed at salt concentrations up to 100 mM (cytoplasmic protein concentrations) or 150 mM (uniform 250 nM protein concentration), and generally decreased with increasing salt, consistent with phase separation driven by electrostatic interactions (*Figure 3—figure supplement 1*). We conclude that the pCas and FAK pathways undergo phase separation synergistically to form droplets that more strongly concentrate cytoplasmic adaptor proteins.

## Kindlin recruits integrin into pCas+FAK droplets

Next, we sought to determine whether phase separated droplets could recruit the β1 integrin cytoplasmic tail. Droplets formed from 1 µ M pCas, Nck, N-WASP, FAK, and paxillin did not recruit β1 integrin (PC = 1.2), and this was not changed by addition of 1 µM talinH (PC = 1.2). However, adding 1 µ M kindlin, significantly increased integrin partitioning in droplets (PC = 3.0, *Figure 3d*). Thus, kindlin specifically couples phase separation of pCas and FAK to the integrin cytoplasmic tail.

To better understand how kindlin recruits integrin into droplets, we measured kindlin partitioning into droplets formed from the pCas mix, FAK mix, and pCas + FAK mix (*Figure 3e–f*). We found that kindlin weakly partitioned into droplets formed with either the pCas mix or the FAK mix, but its partitioning was significantly increased in droplets formed with the pCas + FAK mix (*Figure 3f*). Although kindlin can directly bind paxillin (albeit with low affinity, $K_D$ = 200 µM) (*Böttcher et al., 2017*; *Zhu et al., 2019*), the protein has not been reported to interact with Cas, Nck, or N-WASP (*Dong et al., 2016*). Thus, we next sought to determine how kindlin partitions into droplets containing only pCas, Nck, and N-WASP (the pCas mix). First, we measured kindlin partitioning into previously described droplets formed by phosphorylated Nephrin, Nck and N-WASP but lacking pCas (*Li et al., 2012*). Kindlin did not enrich in these droplets (PC = 1), suggesting that kindlin does not interact strongly with pNephrin, Nck, or N-WASP (*Figure 3—figure supplement 2a-b*). Thus, we hypothesized that kindlin partitions into droplets containing pCas, Nck and N-WASP by binding pCas. To determine if Kindlin could directly interact with Cas, we performed a pull-down experiment. We found that His-tagged kindlin can pull-down both unphosphorylated and phosphorylated Cas (*Figure 3—figure supplement 2c-d*). We conclude that kindlin enriches in droplets formed with the pCas mix through direct interactions with Cas that do not require Cas phosphorylation (*Figure 3a*). A low affinity interaction with both Paxillin and Cas is consistent with a smaller partition coefficient.

Next, we measured β1 integrin tail partitioning into droplets formed with the pCas mix, FAK mix, and pCas + FAK mix (all containing kindlin). Similar to kindlin, β1 integrin weakly partitions into droplets formed with either the pCas mix or the FAK mix, but partitioning significantly increases when both pathways are combined to form droplets with the pCas + FAK mix (*Figure 3g–h*). We conclude that pCas and FAK undergo phase separation synergistically to form droplets that more strongly concentrate kindlin, and consequently integrin.

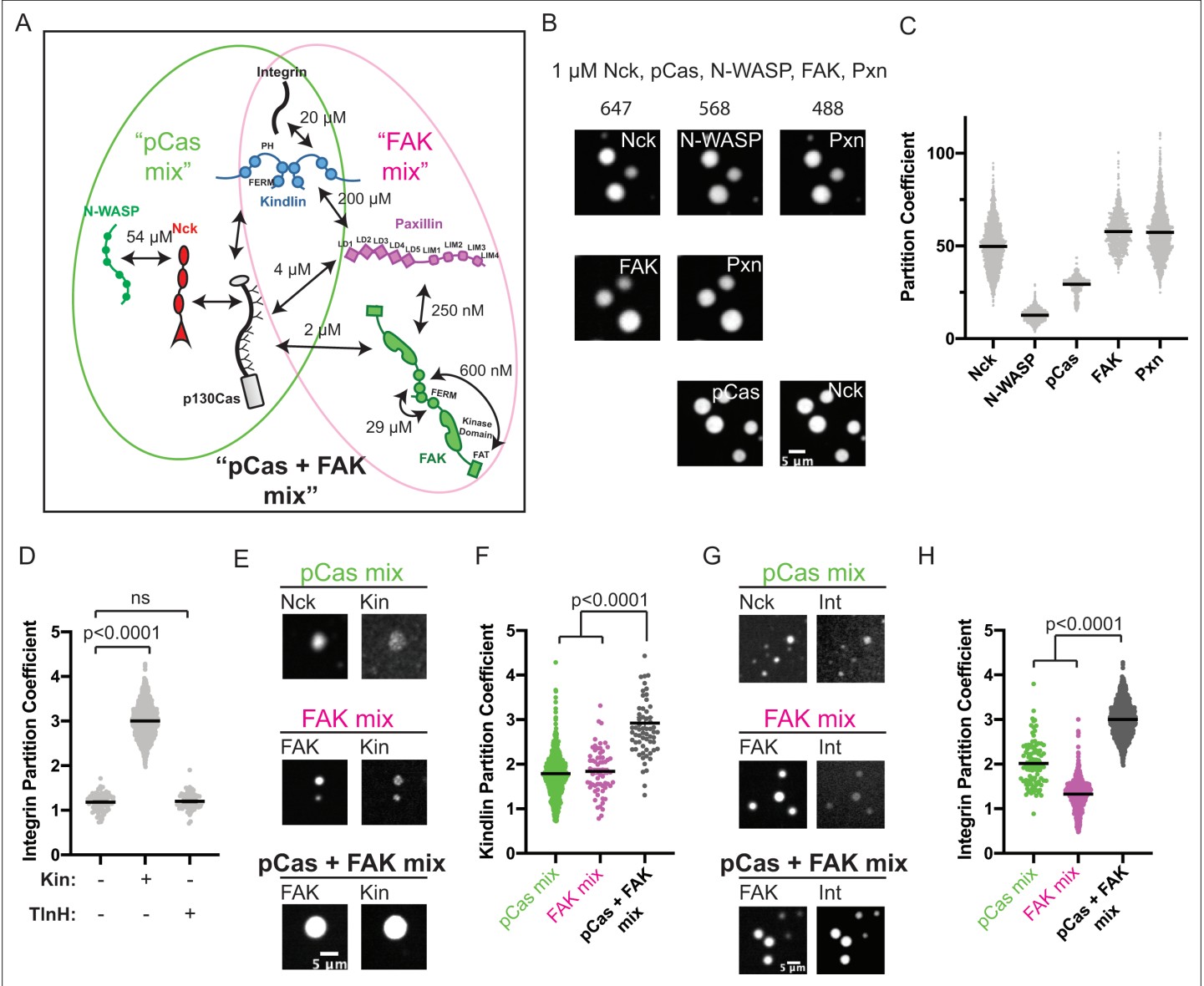

**Figure 3.** The pCas and FAK pathways phase separate synergistically. (**A**) Molecular interactions and known $K_D$ values, including between the two pathways. Details and references in ***Supplementary file 1***. (**B**) Spinning disk confocal fluorescence microscopy images of droplets formed with the *pCas + FAK* mix. TOP: 1 µM each of Nck (15% alexa647), N-WASP (15% Alexa568), pCas, FAK, paxillin (15% Alexa488); MIDDLE: 1 µM each of Nck, N-WASP, pCas, FAK (1% Alexa647 labeled), paxillin (1% Alexa488 labeled); BOTTOM: 1 µM each of Nck (15% Alexa488 labeled), N-WASP, pCas (1% Alexa647 labeled), FAK, paxillin; (**C**) Quantification of constituent partitioning into droplets. Each condition contains at least 750 measurements from four or more independent experiments. (**D**) Quantification of integrin-GFP partitioning into droplets formed with the *pCas + FAK* mix. 1 µM each of Nck, N-WASP, pCas, FAK, paxillin, and B1 Integrin (15% GFP labeled) with either 1 µM Kin or 1 µM TlnH. Each condition contains at least 70 measurements from two or more independent experiments. (**E**) Spinning disk confocal fluorescence microscopy images of droplets. Top: droplets formed with the *pCas mix* (1 µM each of Nck [15% Alexa647 labeled], N-WASP, pCas, and kindlin [5% Alexa568 labeled]). Middle: Droplets formed with the *FAK mix* (1 µM each of FAK (1% Alexa647 labeled), paxillin, and kindlin (5% Alexa568 labeled)). Bottom: Droplets formed with the *pCas + FAK* mix (1 µM each of Nck, N-WASP, pCas, FAK (1% Alexa647 labeled), paxillin, and kindlin (5% Alexa568 labeled)). Contrast settings of kindlin images are matched in all images. (**F**) Quantification of kindlin partitioning. Each condition contains at least 60 measurements from two or more independent experiments. (**G**) Spinning disk confocal fluorescence microscopy images of droplets. Top: Droplets formed with the *pCas mix* (1 µM each of Nck [15% Alexa647 labeled], N-WASP, pCas, kindlin, and integrin [15% GFP labeled]). Middle: Droplets formed with the *FAK mix* (1 µM each of FAK [1% Alexa647 labeled], paxillin, kindlin and integrin [15% GFP labeled]). Bottom: Droplets formed with the *pCas + FAK* mix (1 µM each of Nck, N-WASP, pCas, FAK [1% Alexa647 labeled], paxillin, kindlin, and integrin [15% GFP labeled]). Contrast settings of integrin images are matched in all images. (**H**) Quantification of integrin partitioning. Each condition contains at least 70 measurements from two or more independent experiments. In (**D**), (**F**) and (**H**) significance tested by one-way ANOVA followed by a Tukey multiple comparison test. All scale bars = 5 µm.

*Figure 3 continued on next page*

*Figure 3 continued*

The online version of this article includes the following source data and figure supplement(s) for figure 3:

**Figure supplement 1.** NaCl reduces solution turbidity.

**Figure supplement 2.** Kindlin interacts with p130Cas.

**Figure supplement 2—source data 1.** Uncropped gel images and unprocessed.tif files from *Figure 3—figure supplement 2*.

## pCas- and FAK-dependent phase separation synergistically promote integrin clustering on membranes

Since IACs are membrane-associated condensates, we next tested whether pCas- and FAK- dependent phase separation were sufficient to promote integrin clustering on supported phospholipid bilayers. The α and β integrin cytoplasmic tails separate upon activation (*Kim et al., 2003*; *Wegener and Campbell, 2008*), and talin and kindlin bind to the latter (*Li et al., 2017*). Thus, the β integrin cytoplasmic domain is the minimal fragment required to examine talin- or kindlin- dependent integrin clustering. Although the cell surface expression of integrin receptors varies under different conditions, integrin densities between 300–1500 molecules/$\mu m^2$ have been observed in cells (*Rossier et al., 2012*; *Wiseman et al., 2004*). We attached $His_{10}$-tagged β1 integrin cytoplasmic domain (His-β1) at a density of ~1000 molecules/$\mu m^2$ on phospholipid bilayers composed of 98% phosphatidylcholine (POPC) plus 2% Ni-NTA lipids (*Figure 4a*; *Su et al., 2017*). Using Total Internal Reflection Fluorescence (TIRF) microscopy, we confirmed our methodology consistently generated fluid phospholipid bilayers (*Figure 4—figure supplement 1*) and that His-β1 was uniformly distributed and rapidly diffusing on the bilayer (*Figure 4b–c* 'control').

After addition of 1 µM each of pCas, Nck, N-WASP and kindlin ('pCas mix'), we observe the formation of micron-sized integrin clusters (~1100 clusters/$mm^2$, *Figure 4b–d*). Similarly, after addition of 200 nM FAK plus 1 µM each of paxillin and kindlin ("FAK mix"), we observe the formation of micronsized integrin clusters (~500 clusters/$mm^2$, *Figure 4b–d*). However, when we combined 200 nM FAK and 1 µM each of pCas, Nck, N-WASP, paxillin, and kindlin ('pCas + FAK mix'), we observe more than a 10-fold increase in the number of integrin clusters (12,200 clusters/$mm^2$) compared with either the pCas mix or FAK mix alone (*Figure 4b–d*). Unexpectedly, we found that FAK interacts with Ni-NTA lipid and competes with His-tagged proteins for membrane binding (data not shown). To reduce this effect, we lowered the FAK concentration to 200 nM for experiments on supported phospholipid bilayers. We conclude that pCas- and FAK-driven phase separation synergistically promote integrin clustering on phospholipid bilayers.

FRAP analysis demonstrates that integrin rapidly exchanges between clusters and the surrounding membrane (*Figure 4c*), indicating that the clusters are dynamic assemblies. Furthermore, without the complete mixture of proteins in either the pCas mix (*Figure 4—figure supplement 2* a-b) or FAK mix (*Figure 4—figure supplement 2* c-d), clusters fail to form. Thus, both kindlin and molecules that promote phase separation are required for integrin clustering on bilayers.

Next, we compared fluorescence intensity of integrin inside clusters formed with the pCas + FAK mix with that in the surrounding bilayer. We found that integrin intensity within pCas + FAK clusters increased an average of twofold compared with unclustered integrins (*Figure 4e*). In both MEFs and CHO cells plated on fibronectin, $\alpha 5\beta 1$ integrin is 1.3–2 times more concentrated in nascent adhesions compared with the surrounding regions of the membrane (*Wiseman et al., 2004*). Thus, kindlin coupled to pCas- and FAK-driven phase separation is sufficient to reconstitute physiologically relevant enrichment of integrin within clusters.

These higher protein concentrations are sufficient for the pCas + FAK mix to form droplets in solution (*Figure 3b*). Thus, we sought to determine whether integrin clustering occurred through preformed droplets settling on the membrane, or if condensates were also nucleated on the membrane. We formed droplets with 1 µM pCas, 1 µM Nck (15% Alexa647 labeled), 1 µM N-WASP, 1 µM Kin, 1 µM Paxillin and 200 nM FAK ('1 µM concentration') and quantified the droplet density in the TIRF field on either PEG-coated glass (PEG), empty phospholipid bilayers (lipids) and phospholipid bilayers containing unlabeled membrane-bound integrin (Lipid+ Integrin). As expected, droplets nucleated in solution did settle over time, indicated by the increased droplet density on both PEG and empty lipids (*Figure 4f*). However, there was a significant increase in droplet density at 1 min and 5 min timepoints on bilayers containing integrin, suggesting integrin can promote de novo nucleation on bilayers.

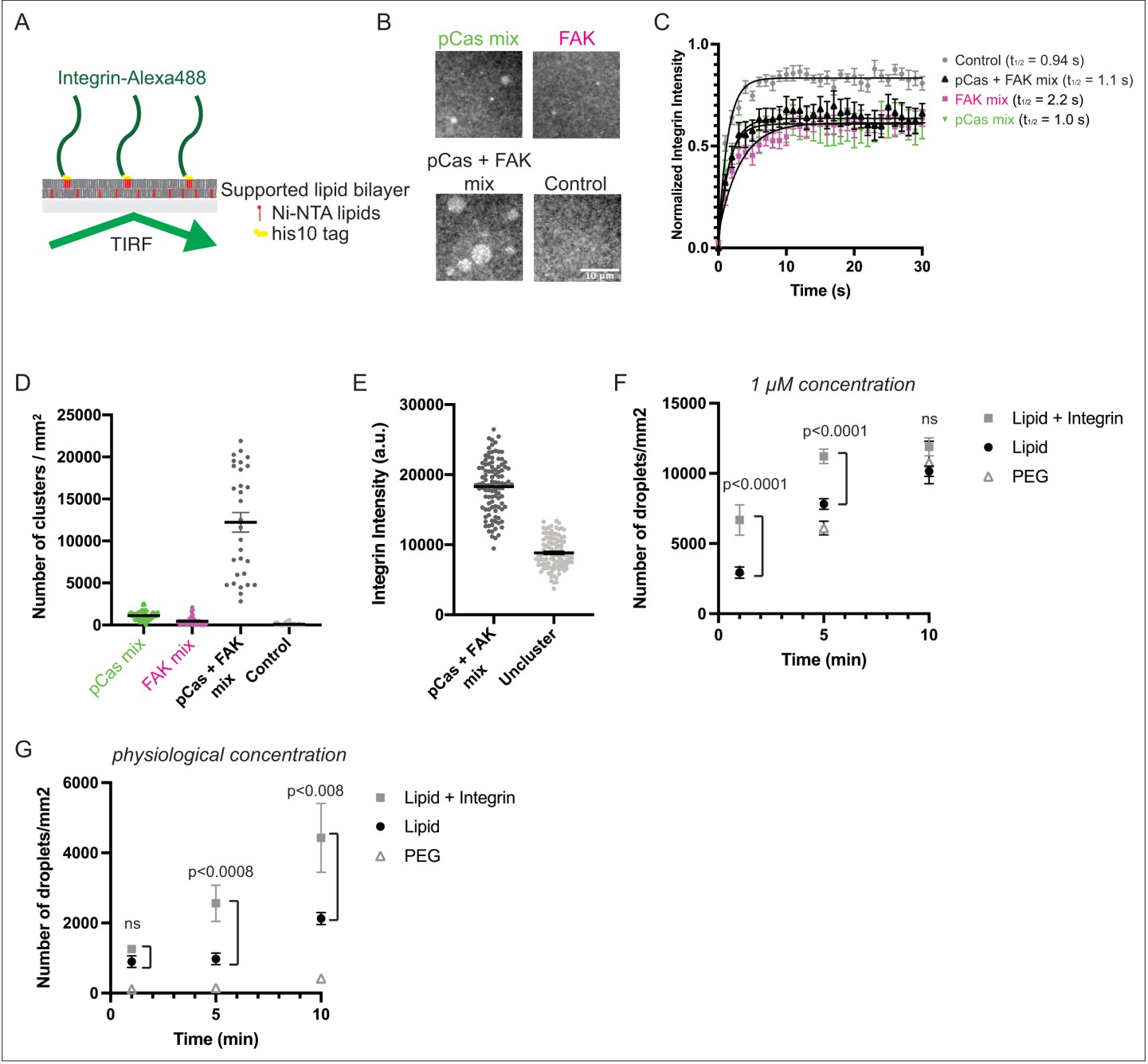

**Figure 4.** Phase separation is sufficient to reconstitute kindlin-dependent integrin clustering on supported phospholipid bilayers. (**A**) Cartoon describing phospholipid bilayer reconstitution. (**B**) Total Internal Reflection Fluorescence (TIRF) microscopy images. Different combinations of proteins (pCas mix: 1 μM each of Nck, N-WASP, Cas and kindlin; **FAK mix**: 200 nM FAK, and 1 μM each of paxillin and kindlin; **pCas + FAK mix**: 200 nM FAK and 1 μ M each of Nck, N-WASP, Cas, paxillin and kindlin; **Control**: Buffer only) were added to membrane-bound integrin (15% Alexa488 labeled, ~ 1000 molecules/μm²). (**C**) Fluorescence Recovery After Photobleaching (FRAP) measurements of his-Integrin in clusters. Each point represents the mean ± SEM of at least 12 independent measurements. The $t_{1/2}$ was calculated from a single exponential fit; fit overlayed on graph (black line). (**D**) Quantification of integrin clusters. Each gray point represents a single field of view, black lines represent mean ± SEM. (**E**) Quantification of integrin intensity within clusters formed with the *pCas + FAK* mix or in the surrounding unclustered regions ('*Uncluster*'). Each point represents a single measurement, black lines represent mean ± SEM. Each condition contains at least 150 measurements from two independent experiments. (**F, G**) Quantification of droplets visible in TIRF field. Experiments were done on PEG-coated glass (PEG), empty phospholipid bilayers (lipids) and phospholipid bilayers containing unlabeled membrane-bound integrin (Lipid+ Integrin). Droplets containing Nck-Alexa647 were visualized with TIRF illumination. 1 μM concentration: 1 μ M pCas, 1 μM Nck (15% Alexa647 labeled), 1 μM N-WASP, 1 μM Kin, 1 μM Paxillin, 200 nM FAK; physiological concentration: 70 n M pCas, 180 nM Nck (15% Alexa647 labeled), 140 nM N-WASP, 60 nM Kin, 60 nM Paxillin, 40 nM FAK. Each point represents the mean ± SEM of at least 17 measurements from

*Figure 4 continued on next page*

*Figure 4 continued*

three independent experiments. Significance tested by one-way ANOVA followed by a Tukey multiple comparison test. All scale bars = 10 μm.

The online version of this article includes the following figure supplement(s) for figure 4:

**Figure supplement 1.** Analysis of phospholipid bilayer fluidity.

**Figure supplement 2.** Analysis of proteins required for integrin clustering.

Furthermore, when we combined proteins at lower, more physiological concentrations (70 nM pCas, 180 nM Nck (15% Alexa647 labeled), 140 nM N-WASP, 60 nM Kin, 60 nM Paxillin, 40 nM FAK) we observed very few droplets on PEG (*Figure 4g*). There is a slight increase in droplet density on empty lipids, possibly due to interaction of FAK with the Ni-NTA lipids. Even with these physiological protein concentrations, we observe a significant increase in droplet density at 5 min and 10 min timepoints on bilayers containing integrin (*Figure 4g*). Thus, the β1 integrin cytoplasmic tail significantly increases the rate of condensate formation on bilayers, likely by increasing the local protein concentration at the membrane and thus accelerating condensate nucleation (*Snead et al., 2021*).

## In vitro droplets and cellular IACs similarly respond to environmental perturbations

Since phase separation is sensitive to solvent conditions, we sought to compare how in vitro droplets and cellular IACs responded to changes in environment. First, we determined the effect of temperature on in vitro phase separation. We combined 250 nM pCas, Nck, N-WASP, FAK, and paxillin, incubated at different temperatures for 30 min, and measured solution turbidity. We observed phase separation at 4 °C, 22°C, and 37°C, although there was a small but significant decrease in solution turbidity at 4 °C (*Figure 5a*). The decrease in phase separation at lower temperatures is somewhat unusual, but not unheard of *Jiang et al., 2015*; *Vrhovski et al., 1997*, as for most protein systems phase separation is enhanced as temperature decreases (*Nott et al., 2015*; *Yoshizawa et al., 2018*). To determine how IACs respond to a transient change in temperature, cells were plated on fibronectin for 3 hr and then incubated at 4 °C, 22 °C, or 37 °C for 10 min, followed by fixation and immunostaining for endogenous paxillin. IACs were observed at all temperatures, but there was a decrease in total adhesion area at 4 °C (*Figure 5b–c*), mirroring the temperature dependence of the in vitro droplets.

Next, we determined the effect of pH on phase separation. In cells, IACs are sensitive to intracellular pH (*Choi et al., 2013*). Mutations in Nhe1 that decrease intracellular pH from a typical resting value of ~7.5–7.0 cause an increase in IAC size and number (*Denker and Barber, 2002*; *Srivastava et al., 2008*). We tested the effect of buffer pH on phase separation in vitro and found that turbidity of solutions containing 250 nM pCas, Nck, N-WASP, FAK, and paxillin increased with increasing acidity. We measured a twofold decrease in turbidity between pH 6.5 and pH 7.5 (*Figure 5c*). To transiently alter intracellular pH, we treated cells with buffer containing 10 μM Nigericin+ valinomycin, ionophores that equilibrate the extracellular and intracellular pH (*Triandafillou et al., 2020*). We incubated cells with buffers for 10 min followed by fixation and immunostaining. We observed a twofold decrease in total adhesion area in cells treated with pH 7.5 compared with pH 6.5 (*Figure 5d–e*). Thus, solution pH has parallel effects in vitro and in cells, with increasing pH causing decreased pCas+ FAK phase separation and decreased total adhesion area. We conclude that solvent perturbations similarly alter phase separation in vitro and IACs in cells.

## In vitro droplets and cellular IACs respond similarly to genetic perturbations

We have identified two distinct sets of molecular interactions, one pCas-dependent and one FAK-dependent, that are sufficient to promote phase separation and drive kindlin-dependent clustering of integrins in vitro. Next, we tested whether mutations that perturb Cas or FAK phase separation in vitro similarly alter IACs in cells.

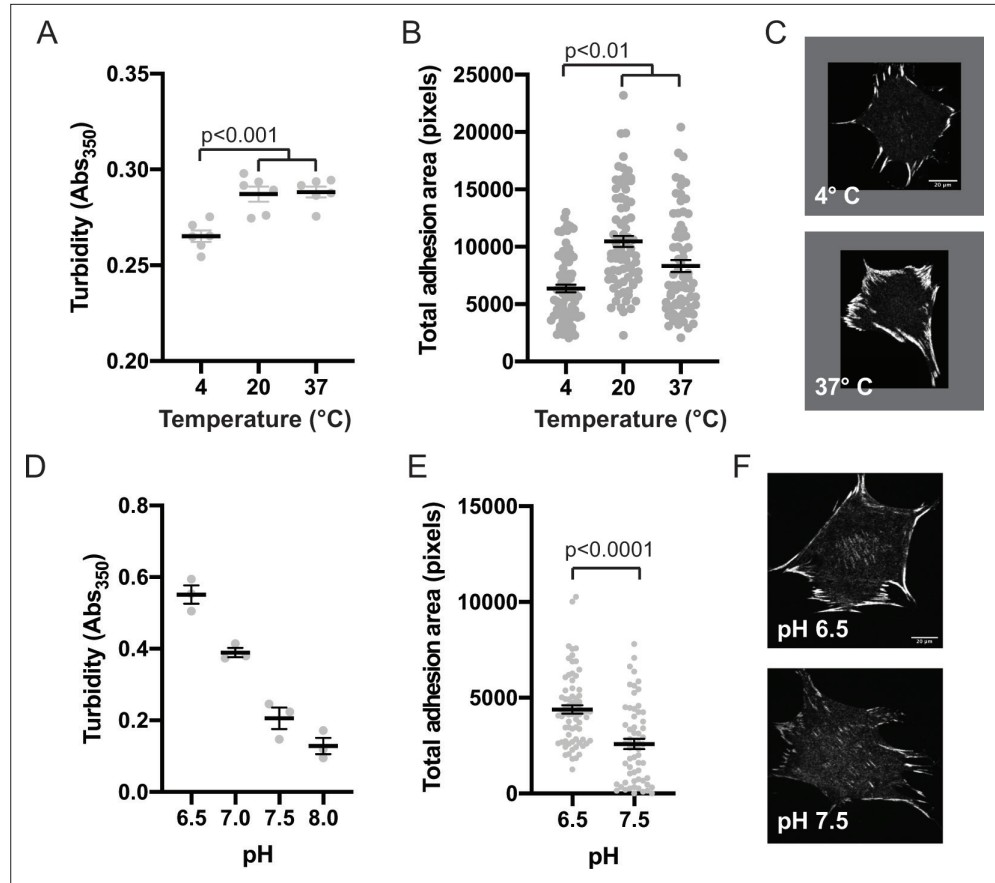

**Figure 5.** Integrin adhesion complexes are sensitive to solvent perturbations that alter phase separation. (**A**) Solution turbidity measurements of solution containing 250 nM each of Nck, N-WASP, pCas, FAK, and Paxillin in buffer containing 50 mM Hepes pH 7.3, 50 mM KCl and 0.1% BSA. Solution incubated at indicated temperatures of 30 min prior to measurements. (**B**) Total adhesion area per cell quantified from spinning disk images of endogenous paxillin. Cells were incubated at indicated temperatures for 10 min prior to fixation and paxillin immunostaining. (**C**) Representative spinning disk confocal microscopy images. (**D**) Turbidity measurements of solution containing 250 nM each of Nck, N-WASP, pCas, FAK, and Paxillin in buffer containing 50 mM KCl, 0.1% BSA and either 50 mM of Hepes or Mes at the indicated pH. (**E**) Total adhesion area per cell quantified from spinning disk confocal microscopy images of endogenous paxillin. Cells were incubated for 10 min prior to fixation and paxillin immunostaining in buffer containing 10 μM nigericin, 10 μM valinomycin, 150 mM NaCl, 50 mM KCl, 1 mM CaCl$_2$, 1 mM MgCl$_2$, 2 mM Glutamax, and either 50 mM Mes pH 6.5 or 50 mM Hepes pH 7.5. Significance tested with unpaired t-test. (**F**) Representative spinning disk confocal microscopy images. All scale bars = 20 μm.

## Preventing Cas phosphorylation reduces droplet formation and paxillin partitioning in vitro

As noted above, unphosphorylated Cas does not phase separate in the presence of Nck and N-WASP (*Figure 1b*). To parallel this material in cells, all 15 tyrosines within the Cas substrate domain were mutated to phenylalanine (Y15F), which prevents phosphorylation of Cas in cells (*Donato et al., 2010*). Using solution turbidity measurements and microscopy, we first confirmed that recombinant CasY15F protein does not phase separate in the presence of Nck and N-WASP (*Figure 6a*; *Figure 6—figure supplement 1*). Next, we determined the effect of CasY15F on in vitro droplet formation and paxillin partitioning in the Cas + FAK mix. CasY15F can still interact with FAK through its N-terminal SH2 domain and paxillin through its C-terminal Cas homology domain (*Figure 6b*; *Figure 1—figure supplement 1*). We find that CasY15F results in fewer, smaller droplets compared with pCas (*Figure 6c*). Paxillin partitioning is also decreased ~1.4 fold in droplets containing CasY15F (*Figure 6d*).

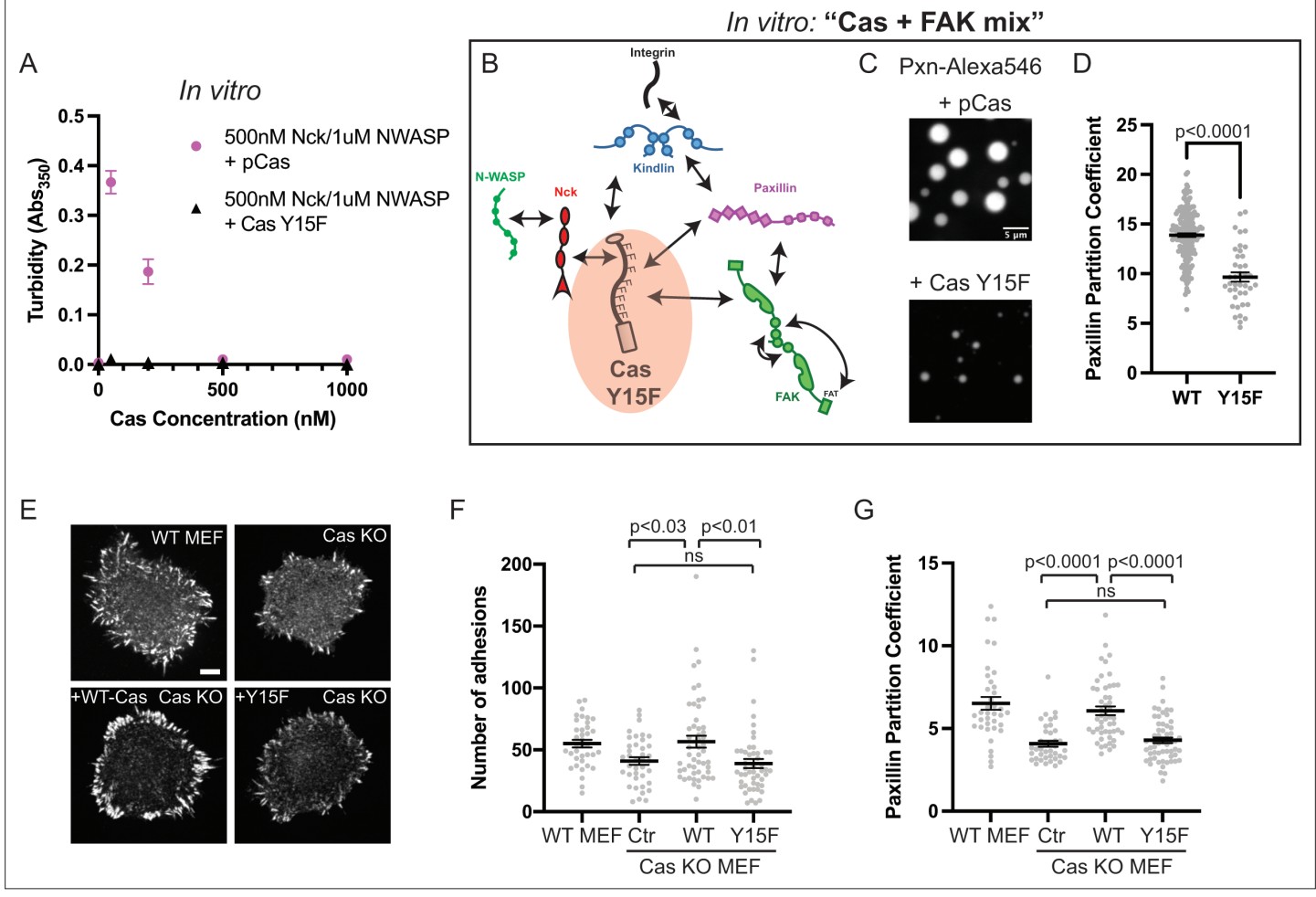

**Figure 6.** Integrin adhesion complexes are sensitive to genetic perturbations that alter Cas-dependent phase separation. (**A**) In vitro solution turbidity measurements. 500 nM Nck and 1 μM N-WASP were combined with increasing concentrations of phosphorylated Cas (pCas, magenta) or Y15F Cas (black). Each point represents the mean ± SEM of three independent measurements. (**B**) Representation of in vitro experiments in C-D. (**C**) Spinning disk confocal fluorescence microscopy images of droplets formed with the *Cas + FAK* mix. TOP: 1 μM each of pCas, Nck, N-WASP, Kin, FAK, and paxillin (15% Alexa546); BOTTOM: 1 μM each of Y15F Cas, Nck, N-WASP, Kin, FAK, and paxillin (15% Alexa546). (**D**) Quantification of paxillin partitioning into droplets. Each condition contains at least 40 measurements from two independent experiments. (**E**) Spinning disk fluorescence microscopy images of MEFs with immunostaining for endogenous paxillin. (**F**) Quantification of number of adhesions. (**G**) Quantification of paxillin partitioning into adhesions (partition coefficient). Significance tested by one-way ANOVA followed by a Tukey multiple comparison test. In (**C**) scalebar = 5 μm. In (**E**) scalebar = 10 μm. Note that the pCas protein used in *Figure 6* is a different batch than protein used in *Figures 1–5* (See *Figure 1—figure supplement 3*).

The online version of this article includes the following source data and figure supplement(s) for figure 6:

**Figure supplement 1.** Cas Y15F does not form droplets.

**Figure supplement 2.** Measuring number of integrin adhesions during cell spreading.

**Figure supplement 3.** Westernblot analysis of MEF cell lines.

**Figure supplement 3—source data 1.** Uncropped gel images and unprocessed gel.tif files from *Figure 6—figure supplement 3*.

## Preventing Cas phosphorylation reduces the number of IACs and paxillin partitioning in cells

Next, we sought to compare these in vitro measurements to the effect of CasY15F on IAC formation and paxillin partitioning in cells. Cas and FAK are important for both IAC formation and disassembly (***Donato et al., 2010***; ***Swaminathan et al., 2016***). To distinguish between potentially confounding functions during these opposing processes, we quantified the number of IACs during initial cell spreading when formation dominates. We plated cells on fibronectin-coated glass and allowed them to spread and form IACs for 20 min. After fixation and immunostaining for endogenous paxillin, we

counted the number of IACs ('number of adhesions', *Figure 6—figure supplement 2*; *Horzum et al., 2014*).

We first examined the role of Cas in regulating the number of IACs. We found that Cas KO MEFs (*Figure 6—figure supplement 3a*) formed significantly fewer IACs than WT MEFs, consistent with a potential defect in nascent adhesion formation (*Figure 6e–f*). We found that expressing WT Cas in Cas KO MEFs restored WT numbers of IACs, while expressing Cas Y15F failed to rescue the number of IACs (*Figure 6e–f*). Furthermore, the partitioning of endogenous paxillin into adhesions ([Intensity inside adhesions]/[Intensity in cytoplasm]) was decreased in Cas KO MEFs, and partitioning was rescued with WT Cas but not Cas Y15F (*Figure 6g*). Paxillin partitioning was ~1.4-fold higher in cells expressing WT Cas compared with Cas Y15F, consistent with the effect of reduced phase separation observed in vitro (*Figure 6d*). Thus, phosphorylation of Cas is required for Cas-dependent regulation of IACs during cell spreading.

## Preventing FAK oligomerization inhibits phase separation and reduces the number of IACs in cells

FAK contains several molecular features that could underlie its ability to phase separate. There are at least two distinct sets of FAK-FAK intermolecular interactions that could promote higher-order oligomerization (*Brami-Cherrier et al., 2014*). FAK can dimerize through association of two FAK FERM domains, and the FERM:FERM interaction is then further stabilized by an additional interaction between a basic patch in the FERM domain and the C-terminal FAT domain (*Figure 2a*). Importantly, these self-interactions can occur in the presence of paxillin, suggesting that higher order FAK oligomerization is compatible with paxillin binding (*Brami-Cherrier et al., 2014*). FAK also contains an intrinsically disordered region (IDR) between the kinase domain and FAT domain that has sequence features common to IDRs that phase separate (*Figure 7—figure supplement 1a*; *Vernon et al., 2018*).

We sought to determine whether perturbing any of these molecular features would reduce FAK phase separation in vitro. Pyk2 is closely related to FAK, but the Pyk2 IDR is predicted to have a lower phase separation propensity than the FAK IDR (*Figure 7—figure supplement 1a*). Fus is an RNA-binding protein with an IDR that is sufficient for phase separation in vitro (*Lin et al., 2015*). Swapping the FAK IDR for either the Pyk2 IDR or the FUS IDR did not dramatically alter FAK phase separation in vitro (*Figure 7—figure supplement 1a-b*). Furthermore, a C-terminal fragment containing the IDR and FAT domain did not undergo phase separation (*Figure 7—figure supplement 1c-d*). Thus, the IDR is not a dominant driver of FAK phase separation. In contrast, a single point mutation in the FERM domain (W226A) that weakens the FERM-FERM interaction (*Brami-Cherrier et al., 2014*) was sufficient to dramatically reduce FAK phase separation measured by solution turbidity (*Figure 7a*) or microscopy (*Figure 7—figure supplement 2*). Furthermore, titrating the FAK C-terminus inhibits phase separation of the full-length protein, suggesting that the FERM-FAT interaction is also important for phase separation (*Figure 7—figure supplement 1e*). We conclude that FAK phase separation is primarily driven by higher order oligomerization (*Figure 7—figure supplement 1f*).

Next, we further characterized the effect of FAK W266A in vitro. Unlike 1 µM WT FAK (*Figure 2c*), 1 µM FAK W266A (1% Alexa-546) does not form droplets on its own (*Figure 7—figure supplement 2*). In the pCas+ FAK mix, FAK W266A results in fewer, smaller droplets (*Figure 7c*) in which paxillin is partitioned ~1.2 more weakly compared with WT FAK (*Figure 7d*).

We next examined the role of FAK in regulating the number of IACs during cell spreading. We found that FAK KO MEFs (*Figure 6—figure supplement 3b*) formed significantly fewer IACs and were noticeably smaller than WT MEFs, consistent with the role of FAK in nascent adhesion assembly (*Figure 6d–e*; *Swaminathan et al., 2016*). While FAK KO MEFs expressing GFP-WT FAK remained noticeably smaller than WT MEFs, GFP-WT FAK still partially rescued the number of IACs. In contrast, FAK KO MEFS expressing GFP-FAK W266A had no change in the number of IACs compared with FAK KO MEFs. Furthermore, the partitioning of endogenous paxillin into adhesions was decreased in FAK KO MEFs, and partitioning was fully rescued with GFP-WT FAK but not GFP-FAK W266A (*Figure 7g*). Paxillin partitioning was ~1.4-fold higher in cells expressing GFP-WT FAK compared with GFP-FAK W266A, consistent with our observations in vitro (*Figure 7d*). Thus, FAK oligomerization is required for FAK-dependent regulation of IACs during cell spreading.

We next measured FAK partitioning into IACs. We found that the W266A mutation causes a 50% reduction in FAK partitioning (*Figure 6f–g*). While W266A FAK cannot drive phase separation, it may

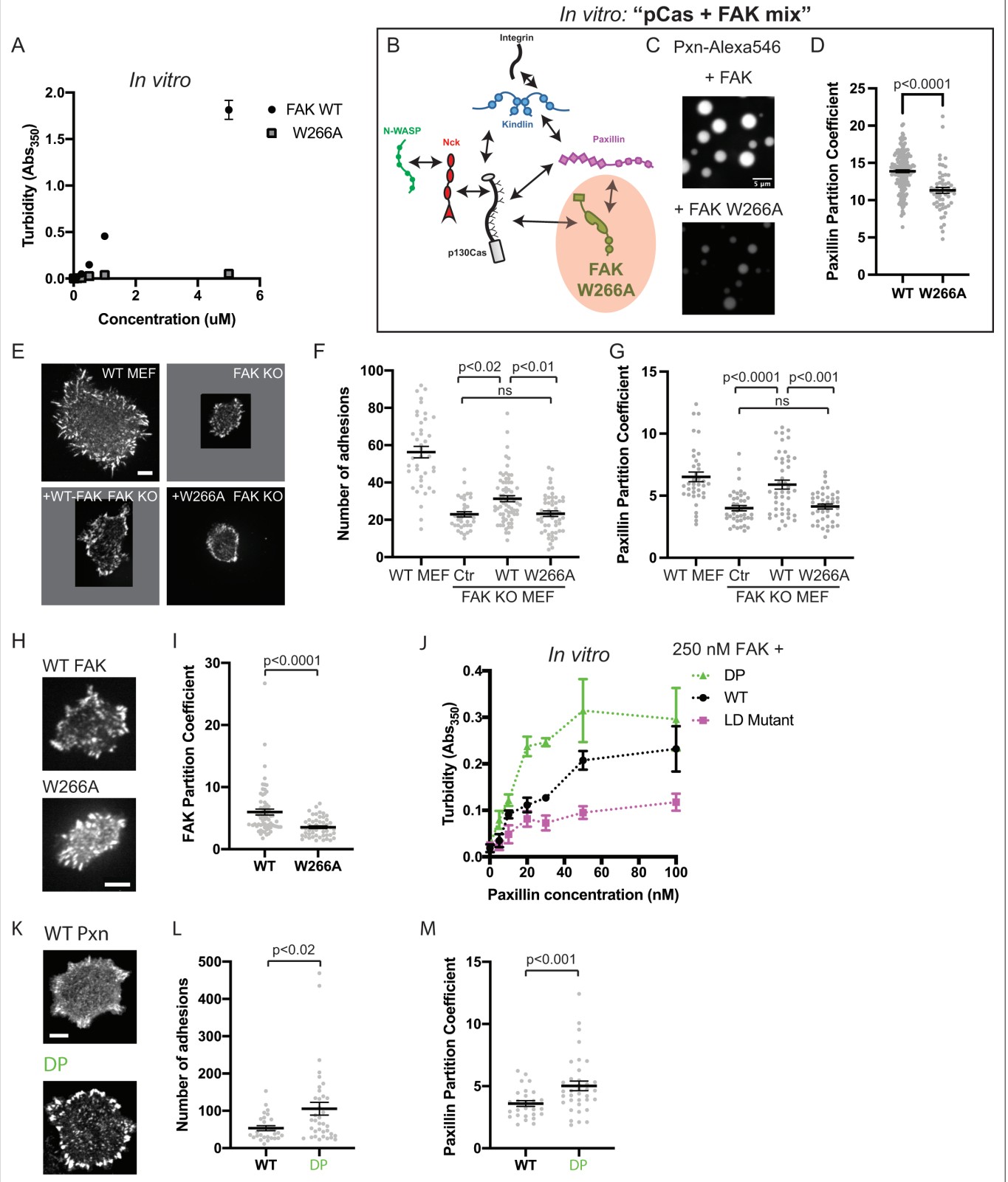

**Figure 7.** Integrin adhesion complexes are sensitive to genetic perturbations that alter FAK-dependent phase separation. (**A**) In vitro solution turbidity measurements with increasing concentrations of recombinant WT FAK or W266A FAK (to inhibit dimerization). WT FAK data is duplicated from *Figure 1B* for comparison. Each point represents the mean ± SEM of three independent measurements. (**B**) Representation of in vitro experiments in C-D. (**C**) Spinning disk confocal fluorescence microscopy images of droplets formed with the *pCas+ FAK* mix. Top: 1 μM each of pCas, Nck, N-WASP,

*Figure 7 continued on next page*

*Figure 7 continued*

Kin, FAK, and paxillin (15% Alexa546); Bottom: 1 µM each of pCas, Nck, N-WASP, Kin, FAK W266A, and paxillin (15% Alexa546). (**D**) Quantification of paxillin partitioning into droplets. Each condition contains at least 40 measurements from two independent experiments. (**E**) Spinning disk fluorescence microscopy images of MEFs transiently expressing GFP-FAK variants with immunostaining for endogenous paxillin. (**F**) Quantification of number of adhesions. (**G**) Quantification of paxillin partitioning into adhesions (partition coefficient). (**H**) Spinning disk fluorescence microscopy images of MEFs transiently expressing GFP-FAK variants. (**I**) Quantification of GFP-FAK partitioning into adhesions (partition coefficient). (**J**) In vitro solution turbidity measurements of paxillin variants. 250 nM recombinant FAK was combined with increasing concentrations of recombinant paxillin variants. (**K**) Spinning disk confocal fluorescence microscopy images of MEF cells transiently expressing GFP-paxillin variants. (**L**) Quantification of number of adhesions. (**M**) Quantification of GFP-paxillin partitioning into adhesions (partition coefficient). In (**F**), (**G**), (**I**), (**L**) and (**M**) each grey point represents a measurement from one cell, and the mean ± SEM mean is indicated by black lines. Data from at least 35 cells from two or more independent experiments. In (**D**), (**F**) and (**G**) significance tested by one-way ANOVA followed by a Tukey multiple comparison test. In (**I**), (**L**) and (**M**) significance tested by an unpaired t-test. In (**C**) scalebar = 5 µm. All other scale bars = 10 µm. Note that the pCas protein used in *Figure 7* is a different batch than protein used in *Figures 1–5* (See *Figure 1—figure supplement 3*).

The online version of this article includes the following figure supplement(s) for figure 7:

**Figure supplement 1.** In vitro analysis of FAK phase separation.

**Figure supplement 2.** FAK W266A does not form droplets.

**Figure supplement 3.** Paxillin valence variants.

still partition into IACs through additional protein-protein interactions, for example binding of the FAK proline-rich motifs to the Cas SH3 domain (*Figure 1—figure supplement 1*). In the 'scaffold/client' description of condensate composition, the mutation converts FAK from behaving more scaffold-like to more client-like (*Banani et al., 2016*; *Ditlev et al., 2018*; *Xing et al., 2020*). Thus, a decrease in FAK partitioning is consistent with loss of FAK-dependent phase separation. Together, these cellular data demonstrate that impairing the phase separation of either Cas or FAK corelates with a partial reduction in the number of IACs observed after 20 min of cell spreading.

### Increasing paxillin valence increases the number of IACs in cells

Next, we sought to identify mutations that might increase phase separation at IACs. Since interactions between paxillin and FAK enhance FAK-dependent phase separation in vitro (*Figure 2d–f*), we sought to understand the molecular basis of this enhancement. The FAK C-terminal FAT domain contains two distinct binding sites for at least two paxillin LD motifs (LD2 and LD4; *Gao et al., 2004*; *Scheswohl et al., 2008*; *Thomas et al., 1999*). To test if multivalent interaction between paxillin and FAK were necessary to enhance phase separation, we engineered paxillin mutants (*Figure 7—figure supplement 3*). To weaken paxillin-FAK binding, we mutated a key Asp residue in each of the five LD motifs to reduce the affinity for FAK ('Paxillin LD mutant'; *Thomas et al., 1999*). To increase paxillin valence, we duplicated the N-terminus (Residues 1–321) to double the number of LD motifs ('Double Paxillin', DP). We added increasing concentrations of each paxillin mutant to 250 nM FAK and measured solution turbidity (*Figure 7j*). We found that the paxillin LD mutant enhances FAK phase separation less than WT, while the DP mutant acts more strongly than WT. We conclude that multivalent interactions between paxillin LD motifs and the FAK C-term FAT domain enhance FAK phase separation.

Next, we tested whether increasing paxillin valence could alter the number of IACs formed during cell spreading. We expressed GFP-WT paxillin or GFP-DP paxillin in WT MEFs. Cells expressing DP paxillin formed twice as many IACs as cells expressing equal levels of WT paxillin (*Figure 7k–l*). Thus, doubling the number of paxillin LD motifs is sufficient to increase the number of IACs. Furthermore, we found that DP paxillin was more strongly partitioned into IACs compared with WT paxillin (*Figure 7m*), consistent with an increase in paxillin-dependent phase separation. Thus, enhancing paxillin-dependent phase separation corelates with an increase in the number of IACs. Together, these experiments demonstrate three distinct genetic perturbations that similarly alter phase separation behavior in vitro and IAC number in cells.

## pCas and FAK synergistically promote nascent adhesion assembly

Since our in vitro data demonstrate that pCas and FAK synergistically promote phase separation and kindlin-dependent integrin clustering, we sought to determine if FAK and Cas act synergistically to promote nascent adhesion formation in cells. To assess nascent adhesion assembly more specifically during cell spreading, we fixed cells after only 5 min of spreading on fibronection, immunostained for

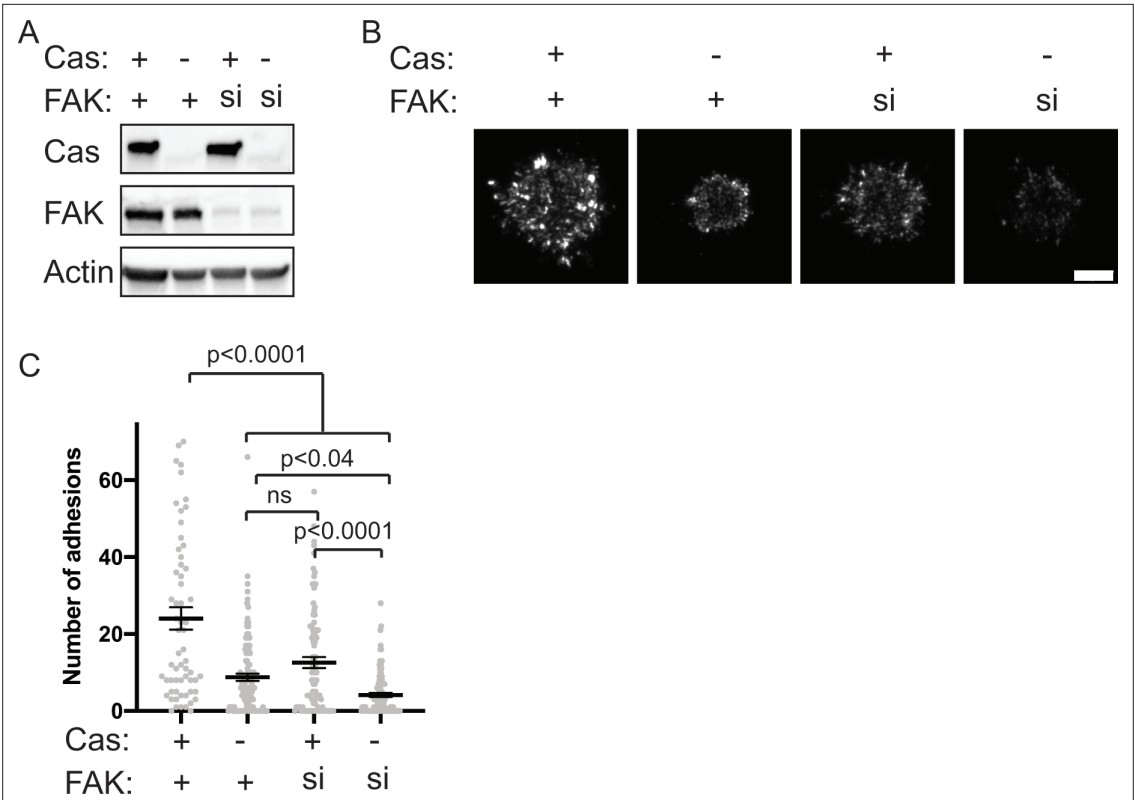

**Figure 8.** Cas and FAK synergistically promote nascent adhesion assembly during cell spreading. (**A**) Western blot analysis of FAK knockdown. WT (+) or Cas knockout (-) MEFs were treated with nontargeting (+) or FAK (si) siRNA for 48 hr and lysates were blotted with Cas, FAK or actin antibodies. (**B**) Total Internal Reflection Fluorescence (TIRF) microscopy images of MEFs fixed after 5 min of spreading with immunostaining for endogenous paxillin. Scalebar = 5 μm. (**C**) Quantification of number of adhesions. Each grey point represents a measurement from one cell, and the mean ± SEM mean is indicated by black lines. Data from at least 60 cells from two or more independent experiments. Significance tested by one-way ANOVA followed by a Tukey multiple comparison test.

The online version of this article includes the following source data and figure supplement(s) for figure 8:

**Source data 1.** Uncropped gel images and unprocessed gel.tif files from *Figure 8a*.

**Figure supplement 1.** Number of adhesions in FAK knockout cells after 5 min spreading.

**Figure supplement 1—source data 1.** Uncropped gel images and unprocessed gel tif files from *Figure 8—figure supplement 1*.

**Figure supplement 2.** Number of adhesions in Cas knockout cells after 20 min spreading.

endogenous paxillin, and imaged cells with TIRF. At this 5-min time point, cells were still predominantly undergoing isotropic cell spreading and IACs remained small and un-elongated. To simultaneously reduce Cas and FAK protein levels in cells, we first used siRNA to knock down Cas in WT or FAK KO MEFs (*Figure 8—figure supplement 1a*). We found that the number of nascent adhesions formed in 5 min was significantly reduced in cells expressing Cas siRNA, while FAK KO MEFs did not significantly differ from WT MEFs (*Figure 8—figure supplement 1b-c*). However, western blot analysis showed that Cas protein levels are elevated in FAK KO MEFs (*Figure 8—figure supplement 1a*) and can be returned to WT levels with the re-expression of WT-FAK-GFP (*Figure 6—figure supplement 3b*). Additionally, FAK KO MEFs were more resistant to Cas knockdown with siRNA (*Figure 8—figure supplement 1a*). While this genetic compensation supports our biochemical evidence suggesting that FAK and Cas may be functionally linked, this approach was not sufficient to simultaneously remove FAK and Cas protein from cells.

Next, we tried an alternative approach to remove FAK and Cas protein from cells by using siRNA to knock down FAK in WT or Cas KO MEFs. Western blot analysis showed this strategy was more effective at achieving low protein levels of both Cas and FAK (*Figure 8a*). Using this approach, we found that loss of either Cas or FAK partially reduces the number of nascent adhesions compared with WT MEFs (*Figure 8b–c*). Simultaneous loss of both Cas and FAK causes a further reduction in the number

of nascent adhesions compared with the loss of either protein alone (*Figure 8b–c*). When both Cas and FAK are absent, cells form few nascent adhesions and fail to spread in 5 min. Fixation and immunostaining of cells after 20 min of spreading showed similar results (*Figure 8—figure supplement 2*), suggesting that decreased nascent adhesion assembly in the absence of Cas and FAK leads to significantly fewer IACs and reduced cell spreading even after 20 min.

Together, these data suggest that Cas and FAK have functionally linked roles in promoting nascent adhesion assembly. Consistent with the in vitro reconstitution, maximal nascent adhesion assembly occurs in cells expressing both Cas and FAK. Furthermore, genetic studies have found that Cas and FAK are synthetic lethal in *Drosophila* (*Tikhmyanova et al., 2010*). Neither the FAK mutant nor the Cas mutant flies have a reported phenotype but combining the Cas and FAK mutations is 100% embryonic lethal with a phenotype similar to the β integrin mutant. Our data suggest that the synthetic lethality in flies may reflect the cooperativity in phase separation, integrin clustering, and nascent adhesion assembly that we have seen biochemically and in cultured cells.

## Discussion

Combining purified recombinant proteins on phospholipid bilayers, we successfully developed an in vitro reconstitution of nascent adhesions. We found that a mixture of seven proteins is sufficient to reconstitute integrin clustering on phospholipid bilayers and recapitulate the integrin enrichment observed within nascent adhesions in cells. These proteins assemble through phase separation at physiologic concentrations and buffer conditions. This phase separation is necessary and sufficient for kindlin-dependent integrin clustering on phospholipid bilayers. Our data suggest that kindlin plays a central role in nascent adhesion assembly by coupling the Cas and FAK pathways to integrins. We find that both environmental perturbations and genetic perturbations can similarly alter phase separation in vitro and the number of IACs in cells. Finally, we find that pCas-dependent and FAK-dependent phase separation synergistically promote integrin clustering on phospholipid bilayers, and that while Cas and FAK can independently promote nascent adhesion assembly, maximum assembly occurs in cells expressing both proteins. Together, these data lead to a model wherein multivalency-driven phase separation of two connected pathways—one based on pCas and the other based on FAK, both coupled to integrins through kindlin—underlies formation of nascent adhesions in cells.

Although kindlin is required for robust IAC formation (*Theodosiou et al., 2016*), the specific role of kindlin in nascent adhesion formation has remained enigmatic. We find that kindlin is required to couple Cas- and FAK-dependent phase separation to integrins through direct interactions with Cas and paxillin. Observations from our biochemical reconstitution are consistent with previous observations that integrin associates with kindlin and not talin during nascent adhesion assembly (*Bachir et al., 2014*), that kindlin does not regulate integrin affinity but rather promotes integrin clustering (*Ye et al., 2013*), and that the kindlin-paxillin interaction is required for efficient nascent adhesion formation in cells (*Theodosiou et al., 2016*).

Although protein-driven phase separation is consistent with many features of cellular IACs, phase separation likely cooperates with additional factors to regulate nascent adhesion formation in cells. Ligand binding to the extracellular domain of integrin, talin binding to the cytoplasmic domain of integrin, and retrograde actin flow are all required for nascent adhesion formation in cells. Our data suggest that while the ECM-integrin-talin-actin linkage may be critical for integrin conformational activation, it may not be required for subsequent macromolecular assembly of adaptor proteins on the integrin cytoplasmic tail. Furthermore, many IAC-associated proteins can interact with negatively charged phosphoinositides. FAK and N-WASP preferentially bind $PI(4,5)P_2$ via basic amino acids (*Goñi et al., 2014*; *Papayannopoulos et al., 2005*), while kindlin preferentially binds $PI(3,4,5)P_3$ via its PH domain (*Liu et al., 2011*). We find that phosphoinositides are not necessary to reconstitute clusters on phospholipid bilayers that resemble nascent adhesions. Consistent with our observations, reduction of $PI(4,5)P_2$ synthesis at IACs in cells leads to the formation of small IACs that contain paxillin and kindlin, but lack talin and vinculin (*Legate et al., 2011*). Furthermore, talin has been shown to directly interact with and activate PIP5K1C (*Di Paolo et al., 2002*), and PIP4K2A was identified in as a constituent of IACs whose local concentration potentially changes during adhesion assembly and growth (*Horton et al., 2015*). Thus, $PI(4,5)P_2$ may be locally generated during IAC formation or maturation, and additional interactions between phosphoinositides and cytoplasmic adaptor proteins could be important to reduce the threshold concentration of phase separation during nascent adhesion assembly or

subsequent adhesion maturation and growth (***Mitchison, 2020***). Relatedly, coupling between phase separation of lipids and phase separation of membrane-associated proteins was recently described in vitro for reconstituted clusters of T cell receptor signaling proteins (***Chung et al., 2021***). Extracellular ligand binding (***Changede et al., 2015***), steric exclusion (***Paszek et al., 2009***), differential lipid composition (***Gaus et al., 2006***; ***Sharma et al., 2004***; ***Son et al., 2017***), and the actomyosin cytoskeleton (***Changede et al., 2015***; ***Kalappurakkal et al., 2019***) likely function together with pCas- and FAK-dependent phase separation to regulate different aspects of integrin activation, nascent adhesion formation and adhesion maturation.

We have focused our investigations here on a minimal set of proteins that are known to assemble at integrins coincident with the initiation of nascent adhesion assembly and which play important functional roles in generating the cellular structures. Yet phase separation at mammalian IACs, particularly during maturation where many more proteins arrive at the structures, likely involves even more proteins. Of the 60 proteins consistently found in proteomic analysis of IACs, 46 have either three or more repeated modular binding domains and/or long intrinsically disordered regions (***Supplementary file 5***; ***Horton et al., 2015***). The multivalent nature of these proteins suggests that they could contribute to higher order assembly and phase separation within IACs. Indeed, LIMD1, a multivalent protein containing a long intrinsically disordered region (***Supplementary file 5***), was recently shown to undergo both phase separation and force-dependent localization to adhesions during maturation (***Wang et al., 2021***). As IAC composition changes during maturation, many different multivalent proteins become enriched (***Horton et al., 2015***; ***Kuo et al., 2011***; ***Schiller and Fässler, 2013***). Future work will be needed to understand how the changing composition might alter the physical properties of the condensed IAC phase and whether additional multivalent proteins, such as LIMD1, become more important at different stages of IAC formation and maturation.

Mechanical forces play a critical role in regulating IAC composition and function. During nascent adhesion assembly, forces are simply required to promote integrin activation (***Oakes et al., 2018***). However, subsequent IAC growth and compositional maturation are dependent on forces transmitted across IACs between the extracellular matrix and the actin cytoskeleton (***Choi et al., 2008***; ***Kuo et al., 2011***; ***Oakes et al., 2012***; ***Schwarz and Gardel, 2012***). Future work will be required to understand how the actin cytoskeleton interacts with and potentially reorganizes the condensed IAC phase. In other systems, actin can regulate condensates by acting as a physical barrier to prevent fusion (***Feric et al., 2016***), by providing mechanical force to promote movement across the membrane surface (***Ditlev et al., 2019***; ***Kim et al., 2019***), or by acting as a scaffold along which condensates can wet or assemble (***Case et al., 2019b***; ***Ditlev et al., 2019***; ***Su et al., 2016***). Understanding how forces are transmitted across the condensed IAC phase will require an understanding of the emergent physical properties of the phase. In vitro reconstitution will be a useful tool to directly address these questions.

Mature IACs (called focal adhesions) are organized into vertical layers, with paxillin and FAK localized near the membrane and actin and actin binding proteins localized >50 nm above the membrane (***Kanchanawong et al., 2010***). At mature focal adhesions, the plasma membrane and actin cytoskeleton provide an inherent polarity; the membrane is physically connected to the ECM through integrins on one side and actin generates forces on the other. These forces are transmitted across the components of the mature focal adhesion. While nascent adhesions may behave like an isotropic liquid condensate, the actin cytoskeleton could induce molecular reorganizations to produce order in the direction perpendicular to the membrane and generate a state more akin to a liquid crystal. While it is not known when this layered organization emerges during IAC assembly and maturation, talin plays a critical role in controlling the distance between actin and the plasma membrane (***Liu et al., 2015***). Additionally, talin undergoes force dependent conformational changes that extend the protein and expose additional vinculin binding sites (***del Rio et al., 2009***). Thus, force-dependent talin extension likely plays an important role in patterning the mature focal adhesion. Many other condensates interact with actin, and the layered organization of mature focal adhesions may be more common than previously appreciated (***Beutel et al., 2019***; ***Bresler et al., 2004***; ***Zeng et al., 2016***).

In conclusion, we have reconstituted minimal macromolecular integrin clusters that have similar composition and features to cellular nascent adhesions. Our biochemical and cellular data provide evidence that Cas- and FAK-dependent phase separation promotes integrin clustering and nascent adhesion assembly. A phase separation model is consistent with many well-documented characteristics of IACs. In other systems, phase separation of receptors enhances downstream signaling (***Case***

et al., 2019b; Huang et al., 2019; Su et al., 2016). Thus, phase separation may provide an important framework for understanding the regulation of signaling and force transmission at IACs.

# Materials and methods

## Key resources table

| Reagent type (species) or resource | Designation | Source or reference | Identifiers | Additional information |
|---|---|---|---|---|
| Strain, strain background (*Escherichia coli*) | BL21(DE3) Competent *E. coli* | New England BioLabs | Cat#C2527I | Chemically competent *E. coli*. |
| Strain, strain background (*Escherichia coli*) | MAX Efficiency DH10Bac Competent Cells | Thermo Fisher | Cat#10361012 | Chemically competent *E. coli*. |
| Cell line (*Mus musculus*) | *Ptk2*+/+, *Trp53*-/- MEFs; WT MEFs | ATCC | CRL-2645; RRID: CVCL_8955 | |
| Cell line (*Mus musculus*) | *Ptk2*-/-, *Trp53*-/- MEFs; FAK KO MEFs | ATCC | CRL-2644; RRID:CVCL_8954 | |
| Cell line (*Mus musculus*) | *Bcar1*-/- MEFs + Control Vector; Cas KO control MEFs | *Meenderink et al., 2010* | N/A | |
| Cell line (*Mus musculus*) | *Bcar1*-/- MEFs + CasWT Vector; Cas KO CasWT MEFs | *Meenderink et al., 2010* | N/A | |
| cell line (*Mus musculus*) | *Bcar1*-/- MEFs + Cas15 F Vector; Cas KO CasY15F MEFs | *Meenderink et al., 2010* | N/A | |
| Cell line (*Spodoptera frugiperda*) | Sf9 Cells | Gibco | Cat#11496015 | |
| Antibody | Mouse monoclonal anti-Paxillin (Clone 349/Paxillin) | BD Biosciences | Cat#610051; RRID: AB_397463 | (1:100) |
| Antibody | Mouse monoclonal anti-FAK (Clone 4.47) | Millipore | Cat#05–537; RRID: AB_2173817 | (1:1000) |
| antibody | Rabbit monoclonal anti-p130 Cas (E1L9H) | Cell Signaling Technology | Cat#13846; RRID: AB_2798328 | (1:1,000) |
| Antibody | Rabbit polyclonal anti-beta Actin | Abcam | Cat#ab8227; RRID: AB_2305186 | (1:5000) |
| Antibody | Goat polyclonal anti-Mouse IgG, Alexa Fluor 568 | Thermo Fisher | Cat#A-11004; RRID: AB_2534072 | (1:250) |
| Antibody | Goat polyclonal anti-Mouse IgG, HRP | Santa Cruz Biotechnology | Cat#sc-2005; RRID: AB_631736 | (1:10,000) |
| Antibody | Goat polyclonal anti-Rabbit IgG, HRP | Santa Cruz Biotechnology | Cat# sc-2030; RRID: AB_631747 | (1:10,000) |
| Sequence-based reagent | ON-TARGETplus Mouse Bcar1 (12927) siRNA - SMARTpool | Horizon Discovery (Dharmacon) | Cat#L-041961-00-0005 | Commercially available |
| Sequence-based reagent | ON-TARGETplus Non-targeting Control Pool | Horizon Discovery (Dharmacon) | Cat#D-001810-10-05 | Commercially available |
| Sequence-based reagent | ON-TARGETplus Mouse Ptk2 siRNA - SMARTpool | Horizon Discovery (Dharmacon) | Cat#L-041099-00-0005 | Commercially available |
| Chemical compound, drug | Alexa Fluor 647 C2 Maleimide | Invitrogen | Cat#A20347 | |
| Chemical compound, drug | Alexa Fluor 488 C5 Maleimide | Invitrogen | Cat#A10254 | |
| Chemical compound, drug | Alexa Fluor 568 C5 Maleimide | Invitrogen | Cat#A20341 | |
| Chemical compound, drug | SNAP-Surface Alexa Fluor 546 | New England BioLabs | Cat#S9132S | |
| Chemical compound, drug | SNAP-Surface Alexa Fluor 488 | New England BioLabs | Cat#S9129S | |
| Chemical compound, drug | Intracellular pH Calibration Buffer Kit (Nigericin/Valinomycin) | Invitrogen | Cat#P35379 | |
| Software, algorithm | MATLAB | Mathworks | | |
| Software, algorithm | FIJI | https://fiji.sc | RRID:SCR_002285 | |
| Other | POPC | Avanti Polar Lipids | Cat# 850457C | |
| Other | PEG5000-PE | Avanti Polar Lipids | Cat# 880230C | |

*Continued on next page*

Continued

| Reagent type (species) or resource | Designation | Source or reference | Identifiers | Additional information |
|---|---|---|---|---|
| Other | DGS-NTA-Ni | Avanti Polar Lipids | Cat# 790404C | |
| Other | NBD PC | Avanti Polar Lipids | Cat# 810130C | |

## Protein expression, purification, and modification

Information on different recombinant protein constructs is provided in *Supplementary file 6*.

### Lck purification:

His6-Lck was expressed from baculovirus in Spodoptera frugiperta (Sf9) cells. Cells were collected by centrifugation and lysed by douncing on ice in 50 mM Tris-HCl (pH 7.5), 100 mM NaCl, 5 mM βME, 0.01% NP-40, 1 mM PMSF, 1 μg/ml antipain, 1 μg/ml benzamidine, and 1 μg/ml leupeptin. Centrifugation-cleared lysate was applied to Ni-NTA agarose beads (Qiagen), washed with 20 mM Tris-HCl (pH 7.5), 1 M NaCl, 20 mM imidazole (pH 7.5), 5 mM βME, and 10% glycerol, and then eluted with 20 mM Tris-HCl (pH 7.5), 100 mM NaCl, 200 mM imidazole (pH 7.5), 5 mM βME, and 10% glycerol. The elute was applied to a Source 15 Q anion exchange column and eluted with a gradient of 100 → 300 mM NaCl in 25 mM HEPES (pH 7.5) and 2 mM βME. Collected fractions were concentrated (Amicon 10 K, Millipore) and applied to an SD75 column in 25 mM HEPES (pH 7.5), 150 mM NaCl, and 1 mM βME.

### p130Cas purification

$His_6$-p130Cas was expressed from baculovirus in Spodoptera frugiperta (Sf9) cells. Cells were collected by centrifugation and lysed by douncing on ice in 20 mM Tris-HCl (pH 8.0), 20 mM Imidazole (pH 8.0), 500 mM NaCl, 10% glycerol and 5 mM βME + cOmplete, EDTA-free Protease Inhibitor tablet (Roche). Centrifugation-cleared lysate was applied to Ni-NTA agarose beads (Qiagen), washed with 20 mM Tris-HCl (pH 8.0), 20 mM Imidazole (pH 8.0), 500 mM NaCl, 10% glycerol and 5 mM βME, and then eluted with 20 mM Tris-HCl (pH 8.0), 400 mM Imidazole (pH 8.0), 500 mM NaCl, 10% glycerol and 5 mM βME. The his tag was removed using TEV protease treatment for 16 hr at 4 °C. Cleaved protein was applied to a Source 15 Q anion exchange column and eluted with a gradient of 100 → 300 mM NaCl in 20 mM Immidazole (pH 7.0), 1 mM DTT and 10% glycerol. Collected fractions were concentrated (Amicon 50 K, Millipore) and applied to an SD200 column in 25 mM Hepes pH7.5, 150 mM NaCl and 10% glycerol. Cas was concentrated using Amicon Ultra Centrifugal Filter units (Millipore) to >400 μ M, mixed with 100 mM HEPES (pH 7.5), 100 mM NaCl, 15 mM ATP, 20 mM $MgCl_2$, 2 mM DTT, and 150 nM active Lck, and incubated for 16 hrs at 30°C. Phosphorylated Cas was resolved on a Mono Q anion exchange column using a shallow 100 mM → 350 mM NaCl gradient in 40 mM Imidazole (pH 7.0), 1 mM DTT and 10% glycerol. Fully phosphorylated Cas was additionally purified by size exclusion chromatography to remove any protein aggregates using a Superdex 200 prepgrade column (GE Healthcare) in 25 mM HEPES (pH 7.5), 150 mM NaCl, 1 mM βME, and 10% glycerol. Cas phosphorylation was confirmed by size shift in SDS-PAGE gel and quantified with mass spectrometry.

### Nephrin purification

BL21(DE3) cells expressing MBP-$His_8$-Nephrin were collected by centrifugation and lysed by cell disruption (Emulsiflex-C5, Avestin) in 20 mM imidazole (pH 8.0), 150 mM NaCl, 5 mM βME, 0.1% NP-40, 10% glycerol, 1 mM PMSF, 1 μg/ml antipain, 1 μg/ml benzamidine, 1 μg/ml leupeptin, and 1 μg/ml pepstatin. Centrifugation-cleared lysate was applied to Ni-NTA agarose (Qiagen), washed with 50 mM imidazole (pH 8.0), 150 mM NaCl, 5 mM βME, 0.01% NP-40, and 10% glycerol, and eluted with 500 mM imidazole (pH 8.0), 150 mM NaCl, 5 mM βME, 0.01% NP-40, and 10% glycerol. The MBP tag was removed using TEV protease treatment for 16 hr at 4 °C (for non his-tagged nephrin, the his6-tag was also removed with precision protease treatment). Cleaved protein was applied to a Source 15 Q anion exchange column and eluted with a gradient of 150 mM→350 mM NaCl in 20 mM Immidazole (pH 8.0) and 2 mM DTT followed by size exclusion chromatography using a Superdex 200 prepgrade column (GE Healthcare) in 25 mM HEPES (pH 7.5), 150 mM NaCl, and 2 mM DTT. Nephrin was concentrated using Amicon Ultra Centrifugal Filter units (Millipore) to >400 μ M, mixed with 100 mM HEPES

(pH 7.5), 100 mM NaCl, 15 mM ATP, 20 mM MgCl$_2$, 2 mM DTT, and 150 nM active Lck, and incubated for 16 hr at 30 °C. Phosphorylated Nephrin (pNephrin) was resolved on a Mono Q anion exchange column using a shallow 100 mM → 350 mM NaCl gradient in 20 mM imidazole (pH 8.0) and 2 mM DTT to separate differentially phosphorylated species of Nephrin. Fully phosphorylated pNephrin was additionally purified by size exclusion chromatography to remove any protein aggregates using a Superdex 200 prepgrade column (GE Healthcare) in 25 mM HEPES (pH 7.5), 150 mM NaCl, 5 mM βME, and 10% glycerol. Complete Nephrin phosphorylation was confirmed by mass spectrometry.

## Nck purification

BL21(DE3) cells expressing GST-Nck were collected by centrifugation and lysed by sonication in 25 mM Tris-HCl (pH 8.0), 200 mM NaCl, 2 mM EDTA (pH 8.0), 1 mM DTT, 1 mM PMSF, 1 µg/ml anti-pain, 1 µg/ml benzamidine, 1 µg/ml leupeptin, and 1 µg/ml pepstatin. Centrifugation-cleared lysate was applied to Glutathione Sepharose 4B (GE Healthcare) and washed with 20 mM Tris-HCl (pH 8.0), 200 mM NaCl, and 1 mM DTT. GST was cleaved from protein by TEV protease treatment for 16 hr at 4 °C. Cleaved protein was applied to a Source 15 Q anion exchange column and eluted with a gradient of 5 → 250 mM NaCl in 20 mM imidazole (pH 7.0) and 1 mM DTT. Eluted protein was pooled and applied to a Source 15 S cation exchange column and eluted with a gradient of 0 → 500 mM NaCl in 20 mM imidazole (pH 7.0) and 1 mM DTT. Eluted protein was concentrated using Amicon Ultra 10 k concentrators and further purified by size exclusion chromatography to remove any protein aggregates using a Superdex 75 prepgrade column (GE Healthcare) in 25 mM HEPES (pH 7.5), 150 mM NaCl, and 1 mM βME.

## N-WASP purification

BL21(DE3) cells expressing His$_6$-N-WASP were collected by centrifugation and lysed by cell disruption (Emulsiflex-C5, Avestin) in 20 mM imidazole (pH 7.0), 300 mM KCl, 5 mM βME, 0.01% NP-40, 1 mM PMSF, 1 µg/ml antipain, 1 µg/ml benzamidine, 1 µg/ml leupeptin, and 1 µg/ml pepstatin. The cleared lysate was applied to Ni-NTA agarose (Qiagen), washed with 50 mM imidazole (pH 7.0), 300 mM KCl, 5 mM βME, and eluted with 300 mM imidazole (pH 7.0), 100 mM KCl, and 5 mM βME. The elute was further purified over a Source 15 Q column using a gradient of 250 → 450 mM NaCl in 20 mM imidazole (pH 7.0), and 1 mM DTT. The His$_6$-tag was removed by TEV protease at 4 °C for 16 hr (for His-N-WASP, no TEV treatment occurred). Cleaved N-WASP (or uncleaved for His-N-WASP) was then applied to a Source 15 S column using a gradient of 110 → 410 mM NaCl in 20 mM imidazole (pH 7.0), 1 mM DTT. Fractions containing N-WASP were concentrated using an Amicon Ultra 10 k concentrator (Millipore) and further purified by size exclusion chromatography to remove any protein aggregates using a Superdex 200 prepgrade column (GE Healthcare) in 25 mM HEPES (pH 7.5), 150 mM KCl, 1 mM βME, and 10% glycerol.

## Kindlin purification:

BL21(DE3) cells expressing His$_6$-sumo-kindlin were collected by centrifugation and lysed by cell disruption (Emulsiflex-C5, Avestin) in 20 mM Tris-HCl (pH 8.0), 20 mM Imidazole (pH 8.0), 150 mM NaCl, 0.01% NP-40, 5 mM βME, 1 µg/ml antipain, 1 µg/ml benzamidine, 1 µg/ml leupeptin, and 1 µg/ml pepstatin. The cleared lysate was applied to Ni-NTA agarose (Qiagen) and washed with 20 mM Tris-HCl (pH 8.0), 20 mM Imidazole (pH 8.0), 150 mM NaCl, 0.01% NP-40, 5 mM βME. Kindlin was eluted with 20 mM Tris-HCl (pH 8.0), 300 mM Imidazole (pH 8.0), 150 mM NaCl, 0.01% NP-40, 5 mM βME. The elute was further purified over a Source 15 Q column using a gradient of 0 → 300 mM NaCl in 20 mM Tris-HCl (pH 8.0), 1 mM DTT. The Collected fractions were pooled and the His$_6$-sumo tag was removed by Ulp1 sumo protease for 2 hr at room temperature. Protein was concentrated using an Amicon Ultra 10 k concentrators and further purified by size exclusion chromatography to remove any protein aggregates using a Superdex 200 column (GE Healthcare) in 25 mM HEPES (pH 7.5), 300 mM NaCl, 10% glycerol, and 1 mM βME.

## Talin head purification:

BL21(DE3) cells expressing talin head-His$_6$ were collected by centrifugation and lysed by cell disruption (Emulsiflex-C5, Avestin) in 20 mM Tris-HCl (pH 8.0), 5 mM Imidazole (pH 8.0), 500 mM NaCl, 1% TritonX, 5 mM βME, 1 µg/ml antipain, 1 µg/ml benzamidine, 1 µg/ml leupeptin, and 1 µg/ml pepstatin.

Centrifugation-cleared lysate was applied to Ni-NTA agarose (Qiagen), washed with 20 mM Tris-HCl (pH 8.0), 30 mM Imidazole (pH 8.0), 500 mM NaCl, 5 mM βME, 1 µg/ml benzamidine, and eluted with 20 mM Tris-HCl (pH 8.0), 100 mM Imidazole (pH 8.0), 500 mM NaCl, 5 mM βME, 1 µg/ml benzamidine. The his6-tag was removed using TEV protease treatment for 16 hr at 4 °C. Cleaved protein was applied to a Source 15 S cation exchange column and eluted with a gradient of 150 mM→500 mM NaCl in 20 mM Immidazole (pH 7.0), 10% glycerol, and 2 mM DTT followed by size exclusion chromatography to remove any protein aggregates using a Superdex 75 column (GE Healthcare) in 25 mM HEPES (pH 7.5), 150 mM NaCl, and 2 mM βME.

## Integrin purification

BL21(DE3) cells expressing integrin (MBP-his$_{10}$-Integrin, MBP-his$_6$-Integrin, or MBP-his$_6$-Integrin-GFP) were collected by centrifugation and lysed by cell disruption (Emulsiflex-C5, Avestin) in 20 mM Tris-HCl (pH 8.0), 20 mM Imidazole (pH 8.0), 150 mM NaCl, 0.01% NP-40, 10% glycerol, 5 mM βME, 1 µg/ml antipain, 1 µg/ml benzamidine, 1 µg/ml leupeptin, and 1 µg/ml pepstatin. Centrifugation-cleared lysate was applied to Ni-NTA agarose (Qiagen), washed with 20 mM Tris-HCl (pH 8.0), 20 mM Imidazole (pH 8.0), 150 mM NaCl, 10% glycerol, 5 mM βME, and eluted with 20 mM Tris-HCl (pH 8.0), 300 mM Imidazole (pH 8.0), 150 mM NaCl, 10% glycerol, 5 mM βME. Eluate was applied to a Source 15Q anion exchange column and eluted with a gradient of 0–300 mM NaCl in 20 mM Tris-HCl (pH 8.0), 10% glycerol and 1 mM DTT. The MBP-tag or MBP-his$_6$-tag was removed using TEV protease treatment for 16 hrs at 4 °C. Cleaved protein was applied to a Source 15 S cation exchange column and eluted with a gradient of 150 mM→500 mM NaCl in 20 mM Bis-Tris (pH 6.0), 10% glycerol, and 1 mM DTT followed by size exclusion chromatography to remove any protein aggregates using a Superdex 75 column (GE Healthcare) in 25 mM HEPES (pH 7.5), 1 M NaCl, 10% glycerol, 1 mM βME.

## FAK purification

His$_6$-FAK was expressed from baculovirus in Spodoptera frugiperta (Sf9) cells. Cells were collected by centrifugation and lysed by douncing on ice in 25 mM HEPES (pH 7.5), 20 mM Imidazole (pH 7.5), 500 mM NaCl, 10% glycerol and 5 mM βME + cOmplete(TM), EDTA-free Protease Inhibitor tablet. Centrifugation-cleared lysate was applied to Ni-NTA agarose beads (Qiagen), washed with 25 mM HEPES (pH 7.5), 20 mM Imidazole (pH 7.5), 500 mM NaCl, 10% glycerol, 5 mM βME, 1 µg/ml benzamidine, and eluted with 25 mM HEPES (pH 7.5), 400 mM Imidazole (pH 7.5), 1 M NaCl, 10% glycerol, 5 mM βME, 1 µg/ml benzamidine. The his6-tag was removed using TEV protease treatment for 16 hr at 4 °C. Cleaved protein was further purified with size exclusion chromatography to remove any protein aggregates applied to a Superdex 200 column (GE Healthcare) in 25 mM HEPES (pH 7.5), 300 mM NaCl, and 1 mM βME.

## Paxillin purification

BL21(DE3) cells expressing GST-Paxillin were collected by centrifugation and lysed by cell disruption (Emulsiflex-C5, Avestin) in 20 mM Tris-HCl (pH 8.0), 300 mM NaCl, 0.01% NP-40, 10% glycerol, 1 mM DTT, 1 µg/ml antipain, 1 µg/ml benzamidine, 1 µg/ml leupeptin, and 1 µg/ml pepstatin. Centrifugation-cleared lysate was applied to Glutathione Sepharose 4B (GE Healthcare) and washed with 20 mM Tris-HCl (pH 8.0), 300 mM NaCl, 0.01% NP-40, 10% glycerol, 1 mM DTT, 1 µg/ml benzamidine. GST was cleaved from protein by TEV protease treatment for 16 hr at 4 °C. Cleaved protein was applied to a Source 15 Q anion exchange column and eluted with a gradient of 5 → 500 mM NaCl in 20 mM imidazole (pH 8.0), 10% glycerol and 1 mM DTT. Eluted protein was concentrated using Amicon Ultra 10 k concentrators and further purified by size exclusion chromatography to remove any protein aggregates using a Superdex 200 column (GE Healthcare) in 25 mM HEPES (pH 7.5), 150 mM NaCl, and 1 mM βME.

## FAK c-term purification

BL21(DE3) cells expressing His$_6$-sumo-FAK c-term were collected by centrifugation and lysed by cell disruption (Emulsiflex-C5, Avestin) in 20 mM Tris-HCl (pH 8.0), 20 mM Imidazole (pH 8.0), 300 mM NaCl, 0.01% NP-40, 5 mM βME, 1 µg/ml antipain, 1 µg/ml benzamidine, 1 µg/ml leupeptin, and 1 µg/ml pepstatin. Centrifugation-cleared lysate was applied to Ni-NTA agarose (Qiagen), washed 20 mM Tris-HCl (pH 8.0), 300 mM Imidazole (pH 8.0), 300 mM NaCl, 0.01% NP-40, 5 mM βME, and eluted

with 20 mM Tris-HCl (pH 8.0), 20 mM Imidazole (pH 8.0), 300 mM NaCl, 5 mM βME,. Eluate was applied to a Source 15 Q anion exchange column and eluted with a gradient of 150 mM→300 mM NaCl in 20 mM Tris-HCl (pH 8.0), 10% glycerol, and 1 mM DTT. Collected fractions were pooled and the His$_6$-sumo tag was removed by Ulp1 sumo protease for 2 hr at room temperature. Protein was concentrated using an Amicon Ultra 10 k concentrators and further purified by size exclusion chromatography to remove any protein aggregates using a Superdex 75 column (GE Healthcare) in 25 mM HEPES (pH 7.5), 300 mM NaCl, and 1 mM DTT.

## Fluorophore conjugation

For conjugation with Maleimide chemistry (Nck, N-WASP, Integrin, FAK) recombinant proteins to be labeled with Alexa fluorophores were concentrated using Amicon Ultra Centrifugal Filter units (Millipore) to ~100 μM. 5 mM βME was added to reduce cysteine residues followed by buffer exchange using a HiTrap 26/10 Desalting column (GE Healthcare) in 25 mM HEPES (pH 7.5) and 150 mM NaCl. Fractions containing protein were collected and concentrated to 100 μM. 500 μM Alexa Fluor 647 C$_2$ Maleimide (ThermoFisher, for N-WASP), Alexa Fluor 488 C$_5$ Maleimide (ThermoFisher, for his$_8$-pNephrin, N-WASP, and his$_8$-N-WASP) or Alexa Fluor 568 C$_5$ Maleimide (ThermoFisher, for Nck) was added, and the reaction was incubated with gentle mixing at 4 °C for 16 hr. The reaction was quenched with 1 μl 14.3 M βME followed by final buffer exchange using size exclusion chromatography (GE Healthcare) in 25 mM HEPES (pH 7.5), 150 mM NaCl, and 1 mM βME. We consistently achieve >98% labeling efficiency.

For conjugation of SNAP-tagged proteins (paxillin, kindlin) recombinant proteins to be labeled with fluorophores were concentrated using Amicon Ultra Centrifugal Filter units (Millipore) to ~50 μM. 1 mM DTT and twofold molar excess SNAP-tag Substrate (SNAP-Surface Alexa488 or SNAP-Surface Alexa546) were added. The mixture was incubated for 30 min at 37 °C. Labeled proteins were purified by final buffer exchange using size exclusion chromatography (Superdex 200 column, GE Healthcare) in the appropriate buffer (i.e. final buffer in the above purification protocol). We consistently achieve >98% labeling efficiency.

Final protein concentration and degree of labeling were calculated from the protein absorbance using the following formulas:

Concentration (M) = A$_{280}$ – (A$_{494}$ × 0.11)/ ExtCoef
Alexa 488
Degree of labeling = A$_{494}$/(71,000 x [protein conc.])
Alexa 568
Concentration (M) = A$_{280}$ – (A$_{577}$ × 0.46)/ExtCoef
Degree of labeling = A$_{577}$/(91,300 x [protein conc.])
Alexa 647
Concentration (M) = A$_{280}$ – (A$_{650}$ × 0.03)/ExtCoef
Degree of labeling = A$_{650}$/(239,000 x [protein conc.])

## Mass spectrometry analysis of p130Cas phosphorylation

Protein samples were analyzed by LC/MS using a Sciex X500B Q-ToF mass spectrometer running Sciex OS v.1.6.1, coupled to an Agilent 1,290 Infinity II HPLC. Samples were injected onto a POROS R1 reverse-phase column (2.1 × 30 mm, 20 μm particle size, 4000 Å pore size), desalted, and the amount of buffer B was manually increased stepwise until the protein eluted off the column. Buffer A contained 0.1% formic acid in water and buffer B contained 0.1% formic acid in acetonitrile. The mobile phase flow rate was 300 μL/min. The acquired mass spectra for the protein of interest were deconvoluted using BioPharmaView v. 3.0.1 software (Sciex) in order to obtain the molecular weight.

## Turbidity measurements

Unlabeled proteins were diluted in 50 mM HEPES (pH 7.3), 50 mM KCl, 1 mM TCEP ('Buffer A'), and 0.1% (1 mg/mL BSA), unless a different buffer is indicated in figure legends. After a 30–60 min incubation, solution was transferred to Quartz cuvette and Absorbance at 350 nm was measured in a Spectrophotometer (Agilent or Thermo Fisher). While we use 1 mg/mL BSA (0.1%) to prevent nonspecific interactions with surfaces, we note that this is well below the BSA concentration necessary to induce crowding (typically 100 mg/mL or greater). For temperature-controlled measurements,

Absorbance at 350 nm was measured using a Cary 100 UV-Visible spectrophotometer equipped with a Peltier thermal controller (Agilent Technologies, Australia). The reaction mixture was incubated in a microcentrifuge tube at the desired temperature for 30 min, and then placed into a pre-equilibrated cuvette and spectrophotometer for measurement.

## 3D droplet assays

384-well glass-bottomed plates (Brooks) were washed with 5% Hellmanex III (Hëlma Analytics) for 3.5 hrs at 55 °C and thoroughly rinsed with MilliQ $H_2O$. Plates were then washed with 1 M NaOH for 1 hr at 55 °C and thoroughly rinsed with MilliQ $H_2O$. 50 µ L of 20 mg/mL mPEG silane MW 5 k (Creative PEGworks) in 95% EtOH was added to each well. The plate was covered in parafilm and incubated overnight at room temperature. The plate was thoroughly rinsed with MilliQ $H_2O$, dried, and sealed with foil. Prior to experiment, individual wells were washed 3 X with MilliQ $H_2O$, blocked with Buffer A containing 0.1% BSA for 30 min at room temperature. For droplet experiments, proteins were combined at 1 µM concentration in Buffer A containing 0.1% BSA and a glucose/glucose oxidase/catalase $O_2$-scavenging system. 80 µL of the mixture was added to the well and incubated for 30 min prior to imaging.

## Small unilamellar vesicle preparation

The general protocol we follow to make small unilamellar vesicles (SUVs) and supported phospholipid bilayers is described in *Su et al., 2017*. Synthetic 1-palmitoyl-2-oleoyl-glycero-3-phosphocholine (POPC), 1,2-dioleoyl-*sn*-glycero-3-[(*N*-(5-amino-1-carboxypentyl)iminodiacetic acid)succinyl] (nickel salt, DGS-NTA-Ni), and 1,2-dioleoyl-*sn*-glycero-3-phosphoethanolamine-N-[methoxy(polyethylen eglycol)–5000] (ammonium salt) (PEG5000 PE) were purchased from Avanti Polar Lipids as chloroform suspension. Using glass Hamilton syringes, lipids were mixed to make a chloroform suspension containing 98% POPC, 2% DGS-NTA-Ni and 0.1% PEG5000 PE. Chloroform was evaporated with gentle stream of Argon Gas, desiccated in a vacuum overnight, and resuspended in PBS (pH 7.3) with vortexing. To promote the formation of small unilamellar vesicles (SUVs), the lipid solution was repeatedly frozen in liquid $N_2$ and thawed using a 37°C water bath until the solution cleared (~35 freeze-thaw cycles). SUV-containing solution was centrifuged at 33,500 g for 45 min at 4°C to remove large vesicles. Cleared supernatant containing SUVs was collected and stored at 4°C covered with Argon for up to two weeks.

## Reconstitution on supported phospholipid bilayers

Briefly, 96-well glass-bottomed plates (Brooks) were washed with 5% Hellmanex III (Hëlma Analytics) for 3.5 hr at 55°C. The plate was thoroughly rinsed with MilliQ $H_2O$, dried, and sealed with foil. Prior to experiment, individual wells were washed with 6 M NaOH for 30 min at 50°C two times, and thoroughly rinsed with MilliQ $H_2O$ followed by equilibration with 50 mM HEPES (pH 7.3), 50 mM KCl, and 1 mM TCEP ('Buffer A'). Twelve µL SUVs were added to cleaned wells covered by Buffer A and incubated for 40 min hr at 40°C to allow SUVs to collapse on glass and fuse to form the bilayer. Bilayers were washed three times with Buffer A to remove excess SUVs, and then blocked with Buffer A containing 0.1% BSA for 30 min at room temperature. His-tagged proteins (10 nM his10-Integrin) were mixed in Buffer A containing 0.1% BSA, added to phospholipid bilayers, and incubated for 2 hr. Bilayers were then washed with Buffer A containing 0.1% BSA to remove unbound His-tagged proteins. Additional proteins were added to the well if required for the experiment. Microscopy experiments were performed in the presence of a glucose/glucose oxidase/catalase $O_2$-scavenging system to reduce photodamage and photobleaching. Unexpectedly, we found that FAK interacts with Nickel and competes with his-tagged proteins for Ni-lipid binding (data not shown). To reduce this effect, we lowered FAK concentration to 200 nM for experiments on supported phospholipid bilayers. Prior to any experiment, the fluidity of bilayers was indirectly assessed by imaging integrin Alexa-488. We photobleached a 5 micron region, and experiments were only performed if FRAP $t_{1/2}$ < 10 s. To confirm that our methods consistently give fluid, uniform bilayers, we also assessed bilayer fluidity directly in bilayers containing 1% PC-NBD, a fluorescent lipid (*Figure 4—figure supplement 1*).

## Microscopy

TIRF images were captured using a TIRF/iLAS2 TIRF/FRAP module (Biovision) mounted on a Leica DMI6000 microscope base equipped with a plan apo 100 × 1.49 NA TIRF objective and a 405/488/561/647 nm Laser Quad Band Set filter cube for TIRF applications (Chroma). Illumination was provided by an integrated laser engine equipped with multiple laser lines (405 nm-100mw/445 nm-75mw/488 nm-150mw/514 nm-40mw/561 nm-150mw/637 nm-140mw/730 nm-40mw, Spectral). Confocal images were captured using a Yokogawa spinning disk and a 405/488/561/647 nm Laser Quad Band Set filter cube (Chroma) with a plan apo 63 × 1.40 NA objective. Images were acquired using a Hamamatsu ImagEMX2 EM-CCD camera.

In *Figure 7*, TIRF images were captured using a Leica TIRF-module mounted on a Leica DMi8 equipped with a plan apo 100 × 1.47 NA TIRF objective and a Quad Band set filter cube for TIRF applications. Illumination was provided by an integrated laser system equipped with multiple laser lines (405 nm-50mw/488 nm-150mw/561 nm-120mw/638 nm-150mW). Images were acquired using LASX software a Hamamatsu Flash 4.0 V3 CMOS camera.

## Quantification of microscope PSF

Measurements were performed in 384-well glass-bottomed plates (Brooks) prepared identically to the 3D droplet assays. Five μL of 0.1 μm diameter TetraSpeck beads (Fisher) were diluted in 120 μL ethanol (final density of ~ 7.5 x 10$^9$ particles/mL). A total of 100 μL was added to a well and incubated for 10 min. Ethanol was removed and the well was rinsed 3X with Buffer A containing 0.1% BSA. Of Buffer A containing 0.1% BSA, 200 μL was added and images were acquired in all channels with identical acquisition settings to 3D droplet assays. Using spinning disk fluorescence microscopy, Z-stacks of beads were acquired. The PSF was measured for each channel in Image J. For each bead, the Z-plane with the highest intensity was identified (*Figure 1—figure supplement 5a*). A linescan through the bead was plotted and the full width half max (FWHM) was determined from a Gaussian fit (*Figure 1—figure supplement 5*). For each channel, 20 beads were measured and the mean FWHM was determined. A single-plane PSF with the calculated FWHM was generated using the Gaussian PSF 3D ImageJ plugin. Images with circles of known diameter (1 – 50 pixels) were generated. The images were convolved with the PSF. The convolved images were analyzes using MATLAB (Mathworks). A mask was generated by a global image threshold using Otsu's method. The diameter and mean intensity inside each masked region were measured. The original circle intensity (255) was divided by the measured intensity of the convolved image and plotted against the measured diameter of the convolved image (*Figure 1—figure supplement 5d*). The data were fit with a single exponential association. This equation was used to correct the intensity of small droplets.

## Measuring droplet partition coefficient

Quantitative image analysis was performed using Matlab (Mathworks). Background images were collected and subtracted from all images before processing. To correct for uneven illumination and detector sensitivity, pixel intensities across a solution containing dye were normalized to the maximum intensity of the image to obtain pixel-by-pixel correction factors (in a 0–1 range). Experimental images were then corrected by dividing by these factors. A mask of droplets was generated from the 647-channel (Either Nck-Alexa647 or FAK-Alexa647) by a global image threshold using Otsu's method. The diameter of each masked droplet was measured. For each individual masked droplet, the mean fluorescence intensity inside the masked region was calculated for each channel (Alexa488, Alexa546, Alexa647, although the number of channels in each experiment varied). The mask was dilated and inverted to calculate the mean intensity in the bulk solution (outside of droplets). Droplet intensity was plotted against droplet diameter. If intensity increased with increasing diameter, the intensity was corrected using the PSF correction as in *Figure 1—figure supplement 5*. Droplets with a diameter less than 12 pixels were discarded, since the intensity cannot be accurately measured or corrected. Partition coefficient was calculated for each droplet by dividing droplet mean intensity by the bulk mean intensity.

## Droplet FRAP measurements

FRAP analysis of droplets was performed using spinning disk confocal microscopy as described above. Droplets with a diameter of ~4-5 microns were used for FRAP measurements. A 19-pixel (4.8

µm) diameter circular region of interest was drawn around the droplet, and the entire droplet was bleached with 405 laser illumination. Images were acquired at 1s interval for 90 sec. Quantification of droplet intensity was performed using ImageJ. The mean intensity inside the ROI was measured, and an identical sized region in the bulk solution was used to correct for photobleaching during imaging. Background and photobleaching corrected intensities were used for normalization.

> Normalized Intensity = $(Intensity - Intensity_{postbleach})/(Intensity_{prebleach} - Intensity_{postbleach})$.
> $Intensity_{prebleach}$ = average of three measurements in the ROI before 405 laser.
> $Intensity_{postbleach}$ = intensity within the ROI immediately after 405 laser (at t=0).

Measurements were repeated at least 8 times, and the curves were averaged and fit using Prism software. To determine the best fit, a single exponential and double exponential fit were statistically compared with an extra sum-of-squares F Test. The values of the best fit are displayed in *Supplementary file 4*. For several molecules, the percent recovery could not be accurately fit from these data, as the recovery did not sufficiently plateau within the 90 second experiment. These are listed as nd.

## His-kindlin pull-down

$His_6$-sumo-kindlin was purified as described above, skipping the Ulp1 cleavage step to keep the $his_6$ tag attached. For the pull-down assay, we made a wash Buffer (20 mM Tris pH 7.5, 150 mM NaCl, 10 mM Immidazole) and elution buffer (20 mM Tris pH 7.5, 150 mM NaCl, 300 mM Immidazole). We added 50 µL of resuspended Ni-NTA Resin to Pierce spin columns (Thermo Scientific). Resin was washed 5 x with 400 µL of wash buffer, removing buffer with centrifugation at 10,000 x g after each wash. A 10 µM solution of his-kindlin was prepared in wash buffer. 100 µL of 10 µM Kin or 100 µL of wash buffer control was added to resin and incubated with gentle shaking for 1 hr at 4°C. After 1 hr, flowthrough was collected by centrifugation at 10,000x g. Resin was washed 5 x with 400 uL of wash buffer, removing buffer with centrifugation at 10,000 x g after each wash. A 10 µM solution of Cas or pCas was prepared in wash buffer. 100 µL of 10 µM Cas, 10 µM pCas or wash buffer control was added to resin and incubated with gentle shaking for 2 hr at 4 °C. The flowthrough was collected by centrifugation at 10,000 x g. Resin was washed 5 x with 400 µL of wash buffer, removing buffer with centrifugation at 10,000 x g after each wash. Finally, 250 µL elution buffer was added to resin and incubated for 5 min at room temperature. Eluted proteins were collected by centrifugation at 10,000 x g. The final elution was mixed with equal volume 2 X SDS-PAGE loading buffer and boiled for 10 min to denature proteins. Samples were loaded onto a 10% bis-Acrylamide gel for SDS-PAGE (200 V for 40 min). Gels were stained with Coomassie Blue and imaged. The following conditions were tested: (1) 10 µM his-kindlin+ buffer control; (2) Buffer control +10 µM Cas; (3) 10 µM his-kindlin +10 µM Cas; (4) Buffer control +10 µM pCas; (5) his-kindlin +10 µM pCas.

## Cell culture and transfection

Mouse Embryonic Fibroblasts (MEFs) were grown in DMEM supplemented with 10% FBS, 2 mM GlutaMAX (Gibco), 100 U/mL penicillin, and 100 µg/mL Streptomycin. *Ptk2* +/+ and *Ptk2* -/- MEFs were obtained from ATCC. *Bcar1*-/- MEFs with Control, CasWT, or CasY15F vectors stably expressed were obtained from Steve Hanks (Vanderbilt) and Larisa Ryzhova (Maine Medical Center). MEF phenotypes were confirmed with westernblot analysis of Cas or FAK protein (*Figure 6—figure supplement 3*) and cells were routinely tested for mycoplasm with the Lonza mycoplasma detection kit. For expression of GFP-tagged Paxillin or FAK variants, cells were transiently transfected with 0.5 µg DNA of EGFP-tagged proteins using lipofectamine 2000 and incubated for 24 hr prior to experiment. For siRNA experiments, cells were transfected using lipofectamine RNAiMAX with pooled oligos (Nontargeting, BCAR1 or PTK2, Dharmacon) and incubated for 48 hr prior to experiment. Details of plasmids used for expression in cells found in *Supplementary file 6*.

## Quantification of total adhesion area

To determine the transient effect of solvent perturbations on adhesion size and/or number, we quantified total adhesion area after a 10-min incubation in specified buffers or media. Cells were plated on glass coverslips coated with 10 µg/mL fibronectin and incubated overnight. The experimental buffer or media was added, and cells were incubated for 10 mins. For temperature experiments, the media was pre-incubated at 4 °C, 22 °C, or 37 °C prior to adding to cells. After a 10-min incubation,

cells were fixed with 3% paraformaldehyde in Cytoskeleton Buffer (CB: 10 mM MES pH 6.1, 150 mM NaCl, 5 mM EGTA, 5 mM MgCl$_2$, 5 mM glucose) for 20 min at room temperature. Cells were then permeabilized for 8 min with CB +0.5% triton-X, quenched for 10 min with CB +0.1 M glycine, and rinsed 2 × 5 min in TBS-T. Coverslips were blocked with 2% BSA in TBS-T for 1 hour. Coverslips were incubated with primary antibody (Ms anti Paxillin, 1:100) in TBS-T with 2% BSA overnight at 4 °C. Coverslips were rinsed 3 × 5 min in TBS-T and incubated with secondary antibody (anti Ms Alexa568, 1:250) in in TBS-T with 2% BSA for 45 min at room temperature. Coverslips were rinsed 3 × 5 min with TBS-T and mounted on slides with Vectashield (Fischer Scientific, H1000NB). Cells were imaged with spinning disk fluorescence microscopy. To quantify total adhesion area, paxillin images were analyzed in ImageJ. Images were thresholded with a lower threshold level of 5500 and an upper threshold level of 65,535. The total thresholded area was then calculated.

### Adhesion formation assay

Cells were trypsinized and plated on glassbottom Matek dishes (5-min timepoint; #1.5 glass) or glass coverslips (20-min timepoint; #1.5 glass) coated with 10 µg/mL fibronectin. For a 5-min time point, unbound cells were gently washed away with fresh media after 1 min. After another 4 min (total spreading time of 5 min), cells were fixed. For a 20-min time point, unbound cells were gently washed away with fresh media after 10 min. After another 10 min (total spreading time of 20 min), cells were fixed. For all experiments, cells were fixed with 3% paraformaldehyde in Cytoskeleton Buffer (CB: 10 mM MES pH 6.1, 150 mM NaCl, 5 mM EGTA, 5 mM MgCl$_2$, 5 mM glucose) for 20 min at room temperature. Cells were then permeabilized for 8 min with CB +0.5% triton-X, quenched for 10 min with CB +0.1 M glycine, and rinsed 2 × 5 min in TBS-T. Coverslips were blocked with 2% BSA in TBS-T for 1 hr. Coverslips were incubated with primary antibody (Ms anti Paxillin, 1:100) in TBS-T with 2% BSA overnight at 4 °C. Coverslips were rinsed 3 × 5 min in TBS-T and incubated with secondary antibody (anti Ms Alexa568, 1:250) in in TBS-T with 2% BSA for 45 min at room temperature. Coverslips were rinsed 3 × 5 min with TBS-T. For the 5-min timepoint, PBS was added to the dish and cells were imaged with TIRF microscopy using 561 illumination. For the 20-min timepoint coverslips were mounted on slides with Vectashield (Fischer Scientific, H1000NB) and imaged with spinning disk fluorescence microscopy. Images in the 488 channel (GFP-tagged proteins) and 561 channel (endogenous paxillin) were acquired.

### Counting adhesions

To reduce experimental noise due to differences in expression levels, GFP-transfected cells were only analyzed if the mean GFP intensity within the cell fell within a defined range (intensity between 1000 and 5000 a.u. following background subtraction, at least 50% of imaged cells were retained with these cutoffs). Adhesions were segmented and counted with ImageJ macros based on a previously published ImageJ workflow (*Horzum et al., 2014*). First macro: run("Subtract Background...", "rolling = 50 sliding"); run("Enhance Local Contrast (CLAHE)", "blocksize = 19 histogram = 256 maximum = 6 mask=*None* fast_(less_accurate)"); run("Exp"); run("Enhance Contrast", "saturated = 0.35"); Then manually run LoG 3D plugin with sigma = 2. Final macro: setAutoThreshold("Default dark"); setOption("BlackBackground", false); run("Convert to Mask"); run("Invert"); run("Analyze Particles...", "size = 5–1000 circularity = 0.00–1.00 summarize"). The final mask was compared to the original image to visually confirm the results. We also validated the final results of automated analysis by manually segmenting adhesions (data not shown).

### Measuring adhesion partition coefficient

Cells expressing GFP-FAK variants or GFP-Paxillin variants were further analyzed to quantify adhesion partitioning of GFP-tagged protein. The same set of images were analyzed for partitioning and counting adhesions. In ImageJ, the image was manually segmented by thresholding using Otsu's method for dark background. The mean intensity within the threshold was measured. Then the thresholding was manually adjusted to segment the cytoplasm (i.e. exclude the adhesions) and the mean intensity was measured. The background was subtracted from intensity measurements, and the intensity within adhesions was divided by the intensity within the cytoplasm.

### Western blot analysis

Cells were plated on 10 cm round tissue culture dish and grown until ~75% confluent. Media was removed and plate was gently washed with 1 mL cold PBS. 1 mL of cold PBS was added to the dish

and cells were scraped for 30 sec. Cells were pelleted with centrifugation at 100 g for 4 min at 4°C. PBS was aspirated and cells were resuspended and lysed in 300 µL cold RIPA buffer (10 mM Tris-HCl (pH 7.6), 1 mM EDTA, 0.1% SDS, 0.1% Na-Deoxycholate, 1% TritonX-100)+ cOmplete, EDTA-free Protease Inhibitor tablet (Roche). Lysates were centrifuged for 10 min at 10,000 rpm and the supernatant was collected. 100 µL of supernatant was combined with 100 µL of 2 X SDS-PAGE loading buffer and boiled for 10 min to denature proteins. Samples were loaded onto a 10% bis-Acrylamide gel for SDS-PAGE (200 V for 40 min). Samples were transferred onto Immobilon-P PVDF Membrane (90 V for 90 min). Membrane was blocked with 5% BSA in TBS-T for 1 hr. Membrane was incubated with primary antibody in TBS-T +2% BSA overnight at 4 °C (Ms anti FAK: 1:1000; Rb anti Cas: 1:1000; Rb anti actin: 1:5000). Membrane was rinsed 3 × 5 min with TBS-T and incubated with secondary antibody 45 min at room temperature (HRP tagged antibodies, 1:10,000). Membrane rinsed 3 × 5 min with TBS-T. Membrane was treated with ECL (Immobilon Western chemiluminescence HRP substrate) for 1 min and visualized with the ChemiDoc MP imaging system (BioRad). Exposure was optimized for each blot.

## Acknowledgements

We thank Steve Hanks (Vanderbilt) and Larisa Ryzhova (Maine Medical Center) for sharing *Bcar1*-/- cell lines, Robert Liddington (Sanford Burnham Institute) for sharing talin and FAK DNA plasmids, W Todd Miller (Stony Brook University) for sharing p130Cas DNA plasmid, Jun Qin (Cleveland Clinic) for sharing kindlin DNA plasmid, and David Corey and Jiaxin Hu (UT Southwestern Medical Center) for use of their Cary 100 UV-Visible spectrophotometer and Peltier thermal controller. We thank the UT Southwestern Proteomics Core Facility and the Koch Institute Biopolymers & Proteomics Facility for performing mass spectrometry analysis. L.B.C. was a Robert Black Fellow of the Damon Runyon Cancer Research Foundation (DRG-2249–16) and is further supported (in part) by the Damon Runyon Cancer Research Foundation (DFS-38–20). This work was also supported by the Howard Hughes Medical Institute and the Welch Foundation (grant I-1544).

## Additional information

### Competing interests

Michael K Rosen: is a co-founder of Faze Medicines. The other authors declare that no competing interests exist.

### Funding

| Funder | Grant reference number | Author |
|---|---|---|
| Damon Runyon Cancer Research Foundation | postdoctoral fellowship | Lindsay B Case Michael K Rosen |
| Damon Runyon Cancer Research Foundation | Dale Frey Scientist Award | Lindsay B Case |
| Howard Hughes Medical Institute | Investigator | Michael K Rosen |
| Welch Foundation | I-1544 | Michael K Rosen |
| Damon Runyon Cancer Research Foundation | DRG-2249-16 | Lindsay B Case Michael K Rosen |
| Damon Runyon Cancer Research Foundation | DFS-38-20 | Lindsay B Case |

The funders had no role in study design, data collection and interpretation, or the decision to submit the work for publication.

### Author contributions

Lindsay B Case, Conceptualization, Data curation, Formal analysis, Funding acquisition, Investigation, Methodology, Project administration, Validation, Visualization, Writing – original draft, Writing

– review and editing; Milagros De Pasquale, Investigation, Performed experiments during revision; Lisa Henry, Investigation; Michael K Rosen, Conceptualization, Formal analysis, Funding acquisition, Investigation, Methodology, Project administration, Supervision, Writing – review and editing

**Author ORCIDs**
Lindsay B Case http://orcid.org/0000-0003-3166-1138
Michael K Rosen http://orcid.org/0000-0002-0775-7917

**Decision letter and Author response**
Decision letter https://doi.org/10.7554/eLife.72588.sa1
Author response https://doi.org/10.7554/eLife.72588.sa2

## Additional files

### Supplementary files

• Supplementary file 1. Curation of published Fluorescence Recovery After Photobleaching (FRAP) measurements of integrin adhesion components. Proteins used in this study are in bold.~ indicates the percent recovery was estimated from graph.

• Supplementary file 2. Curation of published studies characterizing the interactions between proteins used in this study. Unless the specific domain is indicated in parentheses, studies used full-length protein. Technique abbreviations: ITC: Isothermal Titration Calorimetry; FCS: Fluorescence correlation spectroscopy; Co-IP: Co-immunoprecipitation; SPR: Surface Plasmon Resonance; NMR HSQC: Nuclear Magnetic Resonance Heteronuclear Single Quantum Coherence; AUC: Analytical ultracentrifugation.

• Supplementary file 3. Curation of published estimations of cellular concentrations for proteins used in this study.

• Supplementary file 4. Fluorescence Recovery After Photobleaching (FRAP) curve exponential fit of data. Biexponential and single exponential fits were statistically compared with an extra sum-of-squares F Test to determine the best fit. Values of best fit are shown in table. nd = not determined. For several molecules, the percent recovery could not be accurately fit from these data, as the recovery did not sufficiently plateau within the 90 second experiment.

• Supplementary file 5. Protein domains of consensus adhesome components (Adhesome components identified in *Horton et al., 2015*; domains identified using http://pfam.xfam.org). Proteins with multivalent domains shaded grey. Proteins with intrinsically disordered regions longer than 100 amino acids (IDR >100) also indicated.

• Supplementary file 6. Recombinant DNA used in this study. DNA used for expression in bacteria, SF9 cells (pFastBac vector), or mammalian cells (mEGFP-C1 vector). For bacteria and SF9 cell plasmids, the protein sequence of final protease-cleaved and purified protein is shown. For mammalian expression plasmids, the expressed protein sequence is shown.

• Transparent reporting form

### Data availability

All imaging data has been deposited on Dryad under https://doi.org/10.5061/dryad.9p8cz8wj0.

The following dataset was generated:

| Author(s) | Year | Dataset title | Dataset URL | Database and Identifier |
|---|---|---|---|---|
| Case LB, Rosen MK | 2022 | Synergistic Phase Separation of Two Pathways Promotes Integrin Clustering and Nascent Adhesion Formation | https://doi.org/10.5061/dryad.9p8cz8wj0 | Dryad Digital Repository, 10.5061/dryad.9p8cz8wj0 |

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
