## [Editor Report]

This paper identifies phase separation as underlying mechanism that imitates/contributes to the formation of Integrin-containing adhesion sites. The authors demonstrate that p130Cas and FAK undergo phase separation, concentrate kindlin and eventually integrins. The experiments and findings are well described and controlled. This paper is an important contribution to the understanding of how integrins and focal adhesion proteins cluster to form cell attachment sites.

---

## [Decision Letter]

**Decision letter after peer review:**

[Editors’ note: the authors submitted for reconsideration following the decision after peer review. What follows is the decision letter after the first round of review.]

Thank you for submitting your work entitled "Synergistic Phase Separation of Two Pathways Promotes Integrin Clustering and Focal Adhesion Formation" for consideration by *eLife*. Your article has been reviewed by 3 peer reviewers, including Reinhard Fässler as the Reviewing Editor and Reviewer #1, and the evaluation has been overseen by a Senior Editor.

The reviewers spent quite a bit of time reviewing and then discussing this paper, in part because the paper lacks important control experiments and not all conclusions appear to have been directly tested. The reviewers considered the possibility of having you resubmit a revised manuscript after clarifying some of these issues but it was ultimately decided that there were likely too many experiments that needed to be done so it would be better to give you the necessary time to perform the missing experiments and resubmit a new manuscript to *eLife*, or to give you the opportunity to resubmit elsewhere.

*Reviewer #1:*

Cell-ECM adhesion sites contain clusters of integrins associated with plethora of cytoplasmic adaptor and signaling proteins. How integrins cluster to form small and large adhesion sites is not understood. Case and colleagues demonstrate that p130Cas and FAK commence adhesion assembly by undergoing phase separation, which in turn concentrates kindlin and eventually integrins. This work represents a major conceptual advance as it provides interesting evidence for a mechanism of how integrin adhesion site formation is probably initiated. Still, important control experiments and explanations (DLS experiments, protein concentrations, buffer composition, FRAP measurements, SLB studies, characterization of mutant proteins, etc.) are missing, and certain claims (p130Cas-kindlin binding, FAs versus nascent adhesions, etc) would require further exploration.

Case and colleagues report that multivalent interactions of p130Cas with Nck and N-WASP, and FAK with paxillin undergo phase separation in vitro. Furthermore, the authors demonstrate that p130Cas and FAK synergize to induce in vitro condensates, enrich the integrin-activating and clustering adaptor protein kindlin and induce nascent, cell-ECM adhesions. The paper is well written and interesting for a broad readership. The experiments are carried out by a lab with outstanding accomplishments in deciphering the formation of molecular condensates by liquid-liquid phase separation. Still, important control experiments and explanations (DLS experiments, protein concentrations, buffer composition, FRAP measurements, SLB studies, characterization of mutant proteins, etc.) are missing, and certain claims (p130Cas-kindlin binding, FAs versus nascent adhesions, etc.) require further exploration.

1) The authors should perform dynamic light scattering measurements of the protein stock solutions to exclude that the individual recombinant proteins (in particular FAK) form aggregates before applying them in their assays. Ideally, this is done over time and at different temperatures.

2) Phase diagrams demonstrating the influence of protein concentration versus pH or salt concentrations on the formation of condensates are missing.

3) The concentrations of the recombinant proteins used for in vitro phase separation experiments are far above their concentrations in cells.

4) It is not clear from the result and method sections whether the buffer used in the in vitro studies has the appropriate osmolarity.

5) The FRAP experiments of FAK and paxillin in phase droplets substantially differ from those reported in cellular adhesion sites. Therefore, the authors should determine the material properties (viscosity) of the droplets in vitro and ideally compare them to nascent adhesions in cells, e.g. by observing fusion events of droplets.

6) The influence of p130Cas on kindlin depends on an interaction between the two proteins. A direct binding has not been reported so far, which should therefore be demonstrated.

7) Since kindlin binds paxillin in Zn+2-dependent manner, the authors should compare the influence of EDTA and Zn2+ on FAK-mediated condensates and in particular kindlin enrichment.

8) The SLB experiments are carried out without phospholipids and strangely, in the presence of PEG and BSA. In light of the PH domain that is present in kindlin, this experimental set-up is not plausible.

9) The mutant p130Cas and FAK proteins need to be characterized and tested in phase separation experiments in vitro.

10) The authors conclude that 'that pCas- and FAK-driven phase separation provides an intracellular trigger for integrin clustering and nascent adhesion formation.' In contrast to this conclusion, the cellular studies investigate mature focal adhesion rather than nascent adhesion formation. Ideally, the cellular studies are carried out with cells undergoing isotropic spreading (around 5 min upon seeding, depending on cell type and ECM substrate) which is associated with the assembly (and turn over) of nascent adhesions.

*Reviewer #2:*

In the present work the authors report the formation of condensates of adhesion complex proteins. They show in a series of experiments that the condensation of pCas and FAK appears to orchestrate the phase separation and the binding of the these complexes to membranes via kindlin.

Although a bit sensational in the title the paper gives a nice proposal of activation of nascent adhesion formation. The title is a bit misleading since focal adhesions are not observed on the SLB and no actin filament binding is involved. A clear distinction between the here observed condensates and the complexes which are focal adhesion sites should be made throughout the manuscript. Maybe the condensates come closer to nascent adhesion sites.

The interpretation of two distinct phase separation pathways working together to promote integrin clustering seems to be to be an overinterpretation and not entirely consistent with literature. It seems unlikely that the p130Cas+NcK^+^N-WASP would be involved in FA formation, as N-WASP is an actin nucleator and the actin layer in FAs is spatially separated from FAK (https://www.nature.com/articles/nature09621 this paper was cited, but no comment was made to that discrepancy). Which leads to the structure of the complex – which is certainly beyond the scope of the paper but still somehow important to argue a function of the formed aggregates. Neither p130Cas, Nck or N-WASP are components of the consensus adhesome and are not found in nascent FAs.: https://www.ncbi.nlm.nih.gov/pmc/articles/PMC4663675/ The cited paper of Meenderink et al. 2010 states that p130Cas effect is most pronounced in FA disassembly. In this paper a pre-existing phosphorylation pattern is used to show complex formation with FAK. But it was shown that a FAK^+^Src complex leads to the phosphorylation of p130Cas, which is needed for complex formation (https://www.ncbi.nlm.nih.gov/pmc/articles/PMC2700377/). In summary, in the present study the protein system of Case et al. 2019 was used, basically only switching out the phosphotyrosine-protein, without really focusing on the context of FA formation process or if the second phase separation mix is even needed. It is not examined if only adding phosphorylated or dephosphorylated p130Cas to the FAK mix is sufficient for increasing the molecular condensates. This would be needed to claim real synergistic behavior – which is actually a still loosely used term here. I am even not sure if really just an increase of condensate size and protein content (fluorescence intensity) at 1µM with undefined stoichiometry is a strong point – or better how to relate this to the emergence of nascent adhesions – which have a well defined size range and are tightly controlled.

It is not clear how the FRAP measurements are done. The condensates are freely diffusing (also rotational) in 3D solution which would make it difficult to obtain proper FRAP analysis. The stable fluorescence before bleaching need to be reported and contribution of recovery by rotational and translational diffusion of the particles themselves need to be experimentally excluded. The presented curves do not seem to look like a single exponential, and the rapid recovery at t=0 and almost no recovery for pCas and NcK bothers quite a bit.

Unambiguous FRAP data and analysis would seem to be key for any claims on the existence of functional condensates.

Additional specific concerns:

– It is unclear to me how the FRAP measurements are done? The condensates are freely diffusing (also rotational!) in 3D solution which would make it difficult to obtain proper FRAP analysis. The stable fluorescence before bleaching need to be reported and contribution of recovery by rotational and translational diffusion of the particles themselves need to be experimentally excluded. The presented curves do not look like a single exponential, and the rapid recovery at t=0 and almost no recovery for pCas and NcK bothers quite a bit.

Unambiguous FRAP data and analysis seem to be key for any claims on the existence of functional condensates.

– In Figure 1b, the authors show the solution turbidity as a function of the concentration of pCas. Could the authors comment on why the turbidity increase from 0nM to 50nM and decrease from 50 nM on. Do the aggregates disappear or is this a different scattering regime? If they disappear, why are the microscopy measurements work at 1µM? why does the solution turbidity of pCas system does not decay above 100 nM thus a different trend than FAK^+^Paxillin system (Figure 2d)? Does the difference in multivalency matter or explain the differences?

– The pCas system in Figure 4b should have 1 µm of kindlin, or not? It seems it is missing from the caption

Figure 3: In this figure as well as in the text it should be clearer, that "pCas" or "FAK" stands for the whole condensation mix from above This should be explicitly stated and the color code introduced in 3a used in the subsequent mentions of theses protein mixes.

E There is a massive difference in Kindlin Intensities between pCas-droplets and FAK-droplets, but F shows about the same partition coefficient?

Figure 4: No, quality control of SLB is presented, as no fluorescent lipids are present in the SLBs, hence no images shown of SLB or FRAPs of SLB. SLBs are formed in 96 wells which makes the essential washing steps difficult. Well cleaning protocol seems to be insufficient to get clean SLBs from what is commonly state of the art. Control measurements on the quality of the lipid layer needs to be provided (e.g. FRAP of lipids). As a matter of fact, such configuration could also help to perform the FRAP measurements of the condensates.

Surprisingly, and contrary to most reports, no PIP2 is required in the membrane to mediate the adhesion of the complexes to the membrane.

In methods section the written name of the main lipid is DOPC, whereas the given abbreviation is POPC. FAK concentration was lowered, because there was unspecific binding to NiNTA, which is a bit worrisome for the interpretations and comparisons. Increased salt concentration could prevent unspecific binding, but might alter protein condensate dynamics at the same time. Added low concentration (<100mM) of imidazole may help here to prevent unspecific Ni-binding.

Figure 5: In humans, temperature normally varies around 1 {degree sign}C and pH varies 0.1. That means the observed different conditions are far from physiological. It seems a bit far to explain differences in focal adhesion size with the protein condensate properties, as at these conditions the phosphorylation state as well as the lipid composition should be massively altered: https://www.molbiolcell.org/doi/10.1091/mbc.e12-04-0296

Figure 6: The paper is very focused on an effect of phase separation on forming FAs, but the presented in vivo data looks at FAs at a static time point and does not take into account the FA dynamics.

*Reviewer #3:*

Case, Henry and Rosen investigate the influence of the Cas- and FAK-driven phase separation on the formation of integrin-based focal adhesions. Using a combination of in vitro reconstituted and cellular systems, the authors could show that phase separation is apparently sufficient to recreate kindlin-dependent integrin clustering. Interestingly, the authors were able to develop an in vitro system, which is able to reconstitute nascent focal adhesions on supported lipid bilayers consisting of seven proteins. The study is of great general interest, the data quality in most parts sufficient and the manuscript is well written so it is accessible for the large and interdisciplinary readership of *eLife*. The results of this manuscript have the potential to provide the community with interesting clues towards understanding focal adhesion formation based on phase separation of a multitude of interactor proteins.

[Editors’ note: further revisions were suggested prior to acceptance, as described below.]

Thank you for submitting your article "Synergistic Phase Separation of Two Pathways Promotes Integrin Clustering and Integrin Adhesion Complex Formation" for consideration by *eLife*. Your article has been reviewed by 3 peer reviewers, including Reinhard Fässler as the Reviewing Editor and Reviewer #1, and the evaluation has been overseen Anna Akhmanova as the Senior Editor. The following individual involved in review of your submission has agreed to reveal their identity: Daniel Lietha (Reviewer #3).

This newly submitted manuscript has been reviewed at *eLife* before. It was rejected due to the vast number of experiments that needed to be done. In the new submission, several concerns have been addressed. However, a few remain after the reviewers have discussed their reviews with one another. The Reviewing Editor has drafted this to help you prepare a revised submission.

Summary:

This paper provides evidence for phase separation as underlying mechanism to initiate the formation of integrin adhesion sites. The experiments show that p130Cas and FAK undergo phase separation, concentrate kindlin and eventually integrins. The experiments and findings are well described, while several controls are missing and certain conclusions not sufficiently substantiated by the experimental results.

Essential revisions:

1) The reviewers request experiments that show whether the membrane and/or integrins influence the formation of condensates seen by TIRF in Figure 4, or if the condensates would also form in absence of integrin/membrane (ie, comparable to in solution). To test this questions the existing quantifications in Figure 4 should be compared with (a) quantifications of condensates on membranes without integrin tails and (b) quantification of condensates without integrin tails and without lipids (i.e. on the glass surface or some inert coating).

2) The reviewers are not convinced that simply counting nascent adhesions provides a good correlation for the in vitro experiments with purified components. It would be better to measure partitioning between cytosol and nascent FAs. This should be done for FAK-WT vs FAK-W266A, Cas-WT vs Cas-Y15F, and should be straight forward with the available images.

3) in vitro, the components of the pCAS or FAK mix increase the partitioning of other components of the same mix into condensates. To correlate this with cells, partitioning of paxillin into nascent adhesions of FAK-/- , FAK-WT (and perhaps FAK-W266A) cells should be measured, and also partitioning of N-WASP or Nck into adhesions of Cas-/-, Cas-WT (and perhaps Cas-Y15F) cells.

4) The reviewers also request phase separation experiments with mutant Cas and FAK. It is important to confirm also in vitro that condensates are smaller when these mutations are introduced into the proteins. Cas carries 15 mutations, and this experiment will exclude at least in part that the cellular effects occur via phase separation-independent mechanisms.

---

## [Author Response]

[Editors’ note: the authors resubmitted a revised version of the paper for consideration. What follows is the authors’ response to the first round of review.]

Reviewer #1:Case and colleagues report that multivalent interactions of p130Cas with Nck and N-WASP, and FAK with paxillin undergo phase separation in vitro. Furthermore, the authors demonstrate that p130Cas and FAK synergize to induce in vitro condensates, enrich the integrin-activating and clustering adaptor protein kindlin and induce nascent, cell-ECM adhesions. The paper is well written and interesting for a broad readership. The experiments are carried out by a lab with outstanding accomplishments in deciphering the formation of molecular condensates by liquid-liquid phase separation. Still, important control experiments and explanations (DLS experiments, protein concentrations, buffer composition, FRAP measurements, SLB studies, characterization of mutant proteins, etc.) are missing, and certain claims (p130Cas-kindlin binding, FAs versus nascent adhesions, etc.) require further exploration.1) The authors should perform dynamic light scattering measurements of the protein stock solutions to exclude that the individual recombinant proteins (in particular FAK) form aggregates before applying them in their assays. Ideally, this is done over time and at different temperatures.

Our final purification step for every protein preparation is a size exclusion column, to separate potential aggregates from monomers. All our purified proteins, including kindlin and FAK, elute as a single, gaussian peak on the size exclusion column, indicating they are not aggregated (as aggregates would elute in the column void volume). We have optimized purification conditions and final buffers to ensure we have pure, unaggregated protein for our studies. We now more explicitly describe this throughout the methods: the protein is purified by…. “followed by size exclusion chromatography to remove any protein aggregates” …

For example, Author response image 1 is a FAK chromatogram from an SD200 column:

You can see the narrow peak of FAK, eluting after the void volume (~9 mL), followed by a smaller peak of a lower molecular weight contaminant (~60 KDa). Thus, our rigorous purification protocols ensure our starting materials are pure and not aggregated.

**Author response image 1. sa2fig1:** 

2) Phase diagrams demonstrating the influence of protein concentration versus pH or salt concentrations on the formation of condensates are missing.

With a simple, one-component system, it is possible to map a phase diagram of protein concentration versus salt. However, with complex multi-component systems, this becomes less straightforward as changing the concentration of a single component can have non-intuitive effects. Even with only two proteins phase separating together, the simple features of single-component phase separation are not observed (this is explained in more detail in this recent review: Lyon et al., 2020 https://doi.org/10.1038/s41580-02000303-z).

In the revised manuscript, we performed additional experiments to characterize the dependence of droplet formation on salt, in addition to our previous experiments showing the dependence on pH (Figure 5). We titrated NaCl concentration for two different concentrations of proteins and measured solution turbidity (new Figure 3 —figure supplement 1). We find that turbidity decreases with increasing salt concentrations, consistent with phase separation driven by electrostatic interactions (such as the FAK FERM:FAT interaction). With a higher protein concentration of 250 nM each of pCas, Nck, N-WASP, FAK, Paxillin, and Kindlin, Abs_350_ approaches 0 around 300 mM NaCl. With a lower protein concentration using the cytoplasmic concentrations of each component, Abs_350_ approaches 0 around 150 mM NaCl.

3) The concentrations of the recombinant proteins used for in vitro phase separation experiments are far above their concentrations in cells.

Table S3 lists the measured cellular concentrations for all of the proteins used in our studies. In Figure 1 —figure supplement 3 and Figure 2 —figure supplement 2 we examined our systems at these cellular concentrations, and observed small droplets. We also now characterize the effect of salt on droplets formed with cellular concentrations of pCas, Nck, N-WASP, paxillin, and FAK (Figure 3 —figure supplement 1). However, the droplets formed at low concentrations are smaller than the resolution limit of our microscope, and thus not amenable to quantitative fluorescence imaging. Thus, in other parts of the manuscript we used higher concentrations to generate larger droplets, which enable quantification of partition coefficients and FRAP dynamics. Nevertheless, we are confident that the molecules undergo phase separation at physiologic concentrations/conditions, and that the behaviors we observe are reflective of those conditions.

4) It is not clear from the result and method sections whether the buffer used in the in vitro studies has the appropriate osmolarity.

We list the buffers used throughout the results, figure legends and methods section. The standard buffer for most in vitro assays is: 50 mM Hepes pH 7.3, 50 mM KCl, 0.1% BSA. Our buffers contain 50 mM KCl as the salt, which is commonly used for in vitro reconstitutions, especially those involving the actin cytoskeleton (see for example, Hansen et al., 2013: https://doi.org/10.1007/978-1-62703-538-5_9). One of our long-term goals is to examine coupling of actin to the phase separated integrin clusters. Thus, we used a buffer that will be compatible with future studies interrogating the role of actin polymers in regulating in vitro clusters.

As described above, we have performed additional experiments to test the effect of NaCl concentration on condensates. We see that droplet formation becomes less favored between 150-300 mM NaCl. We speculate that at cellular concentrations and osmolarity, spontaneous phase separation would be less likely to occur in the cytoplasm, which could ensure that macromolecular assembly occurs specifically at sites of integrin activation on the membrane.

Finally, we note that there is no universally agreed upon buffer to best represent the interior of the cell, and various biochemical reconstitutions vary widely in composition and osmolarity. For example, as distinct from the actin buffer above, studies of microtubule dynamics are typically performed in BRB80 buffer (80 mM PIPES, 1 mM MgCl2, 1 mM EGTA, pH 6.8; e.g. Hyman and Mitchison, J. Cell Biol., 110, 1607 (1990)), studies of transcription often use acetate buffers (e.g. 10 mM Hepes, 6.5 mM MgSO_4_, 5 mM magnesium acetate, 70 mM potassium acetate, 5 mM EGTA, 2.5 mM dithiothreitol,4 mM phosphoenolpyruvate,10% glycerol; Lue and Kornberg, PNAS, 84, 8839 (1987)), and reconstitution of vesicle trafficking uses buffers containing KCl and acetate (25 mM HEPES (pH 7.0), 15 mM KCI, 2.5 mM magnesium acetate, and 0.2 M sucrose; Orci, Glick and Rothman, Cell, 46, 1986). In all these cases, meaningful biochemical data that inform on cellular processes have been obtained. We feel that, especially with the new salt dependence data, our biochemical studies will be similarly informative on biology even if our buffers are only approximations of the cytoplasm.

5) The FRAP experiments of FAK and paxillin in phase droplets substantially differ from those reported in cellular adhesion sites. Therefore, the authors should determine the material properties (viscosity) of the droplets in vitro and ideally compare them to nascent adhesions in cells, e.g. by observing fusion events of droplets.

We agree that ideally one would be able to observe similar material properties of condensates reconstituted in vitro with their counterparts in cells. However, numerous considerations makes such comparisons difficult to perform and hard to interpret. First, we obviously are lacking components in our reconstitution that likely contribute strongly to material properties of adhesions in cells—attachment to the extracellular matrix, attachments to the actin cytoskeleton, multi-component attachments to lipids (see below), molecular crowding, etc. Adding these components is well beyond the scope of the present work, and moreover, it is unclear even which might play important roles in dictating material properties.

Second, viscosity cannot be measured simply by observing fusion events, since fusion rates depend on both viscosity and surface tension. Rheological measurements would be necessary. These would be extremely difficult to perform on the inside of the plasma membrane in a live cell, in a short-lived structure like a nascent adhesion. Further, given that our condensates are attached to membranes, membrane fluidity (at the site of adhesion) will play a significant role in their physical properties, and it is unclear how one might reproduce this in vitro.

Third, FRAP measurements in cells vary quite widely in different studies, and can even vary for identical condensates within a single cell. FRAP measurements within focal adhesions vary widely in the literature. FRAP measurements are also only easily done in larger adhesions that remain stable for several minutes (i.e. not in nascent adhesions). Because of these issues is not clear to us how one would measure the material properties of cellular nascent adhesions and meaningfully compare them to properties measured in vitro, and this challenge is beyond the scope of this study.

This being said, we believe that our FRAP experiments on in vitro condensates are, in fact, qualitatively consistent with those previously reported in cells. Although the in vitro droplets described in this study lack many of the core components of cellular adhesion sites, such as talin and vinculin, their FRAP dynamics are within an order of magnitude of reported cellular measurements. Our in vitro data, which fits best with a biexponential model, show FAK t_1/2_ fast = 6 s; FAK t_1/2_ slow = 56 s; FAK 62% recovery; paxillin t_1/2_ fast = 7 s; paxillin t_1/2_ slow = 78 s; pCas t_1/2_ fast = 2 s; pCas t_1/2_ slow = 39s. While our time-courses were too short to reliably determine a fraction recovery, it is clear qualitatively that the condensates also contain an immobile fraction. By comparison, a recent cellular study fit curves with a single exponential and found FAK t_1/2_ = 10s with 60% recovery, paxillin t_1/2_ = 16s with 65% recovery and Cas t_1/2_ = 14s with 70% recovery (Stutchbury et al., 2017). Thus, like our reconstituted droplets, cellular adhesions contain a dynamic pool of molecules that rapidly exchange with the cytoplasm and a more stable population of molecules that do not recover. in vitro condensates, while not perfect, do reasonably recapitulate the published properties of cellular adhesions. To help readers better understand and compare FRAP measurements, we have now included detailed information about the fit in supplemental table 4, and made more direct comparisons between our data and published cellular data.

For example, we now include the following discussion of FRAP data in the results:

“We found that pCas, Nck and N-WASP rapidly exchanged between droplets and bulk solution (Figure 1e; Table S4; pCas t_1/2_ fast = 2 s; pCas t_1/2_ slow = 39s; Nck t_1/2_ = 8 s; N-WASP t_1/2_ fast = 1 s; N-WASP t_1/2_ slow = 37 s ), although a fraction of Nck and pCas molecules did not recover in the 100 s timeframe of the experiment, suggesting a slower phase of recovery and/or an immobile fraction (Table S4). In cells, IAC-associated proteins, including Cas, often contain a population of fast exchanging molecules (t_1/2_ < 30 s) as well as an immobile fraction (up to 50% immobile for some proteins; Table S1). Thus, similar to cellular IACs, droplets exhibit liquid-like material properties with some solid-like elements as well.”

6) The influence of p130Cas on kindlin depends on an interaction between the two proteins. A direct binding has not been reported so far, which should therefore be demonstrated.

To address this concern, we have performed additional biochemical experiments. These demonstrate a direct interaction between kindlin and p130Cas. As the reviewer suggested, we performed a pull-down assay with purified recombinant proteins. We found that hisKindlin could pull down both unphosphorylated Cas and phosphorylated Cas (Figure 3 —figure supplement 2). These new data provide evidence that kindlin interacts directly with p130Cas, and that the interaction does not require phosphorylation of the Cas substrate domain.

7) Since kindlin binds paxillin in Zn+2-dependent manner, the authors should compare the influence of EDTA and Zn2+ on FAK-mediated condensates and in particular kindlin enrichment.

A recent structural study used NMR to investigate the structural basis of the paxillin – kindlin interaction (Zhu et al., 2019: https://doi.org/10.1016/j.str.2019.09.006). They found that the paxillin LIM4 domain binds directly to the Kindlin F0 domain. Furthermore, they showed that the LIM4 domain adopts a canonical LIM domain fold, with two classic zinc fingers. Zinc fingers require Zn^2+^ to maintain their structure, and EDTA is known to irreversibly denature and aggregate these domains (see for example doi: 10.1042/BJ20041096). Considering this data, treating these condensates with EDTA will most likely denature and aggregate paxillin, making any results difficult to interpret. Thus, we have not explored this issue further.

8) The SLB experiments are carried out without phospholipids and strangely, in the presence of PEG and BSA. In light of the PH domain that is present in kindlin, this experimental set-up is not plausible.

Our SLBs are phospholipid bilayers, and we apologize for not describing our bilayer composition in sufficient detail previously. We have edited the methods to hopefully be less opaque (including adding the lipid information to the resource table) and also now refer to the bilayers in the text and phospholipid bilayers. We use bilayers containing 98% POPC, 2% DGS-NTA(Ni), and 0.1% PE-PEG, and each component has a specific purpose:

POPC is a phosphatidylcholine (1-palmitoyl-2-oleoyl-sn-glycero-3-phosphocholine) commonly used to make phospholipid bilayers, liposomes, and nanodiscs for biophysical/biochemical assays

(e.g. Pautot et al., 2005 https://doi.org/10.1038/nchembio737; Jiang et al., 2019 https://doi.org/10.1073/pnas.1916204116; Libby and Su, 2020, 10.1007/978-1-07160266-9_13; Tay and Du, 2017 https://doi.org/10.1016/bs.mie.2016.09.043; Ritchie et al., 2009, 10.1016/S0076-6879(09)64011-8; Carbone et al., PNAS, 2017, https://doi.org/10.1073/pnas.1710358114).

DGS-NTA(Ni) (1,2-dioleoyl-sn-glycero-3-[(N-(5-amino-1-carboxypentyl)iminodiacetic acid)succinyl] (nickel salt)) is a Ni^2+^-chelating lipid that enables attachment of His_10_-tagged proteins to the bilayer. In Author response image 3, we used it to attach the b-integrin tail.

**Author response image 3. sa2fig3:** 

PE-PEG-5000 1,2-dioleoyl-sn-glycero-3-phosphoethanolamine-N-[methoxy(polyethylene glycol)-5000] is a phospholipid with a PEG moiety on the head group. Adding very small amounts of PEGylated phospholipids ensures the bilayers remain fluid and prevents their tight adhesion to the glass below. Again, doping with PEGylated lipid is widely used in biochemistry performed on supported lipid bilayers (Jiang et al., 2019 https://doi.org/10.1073/pnas.1916204116; Libby and Su, 2020, 10.1007/978-1-0716-0266-9_13; Carbone et al., PNAS, 2017, https://doi.org/10.1073/pnas.1710358114).

**Author response image 4. sa2fig4:** 

We also include a small amount of BSA (0.1% = 1 mg/mL) in all our in vitro buffers to prevent non-specific interactions of proteins with plastic or lipid surfaces. Again, this is standard practice in a wide swath of biochemical analyses (Kwiatkowski et al., PNAS, 2010; Nair et al., Nat Protoc, 2010; Risca et al., PNAS, 2012; Acker and Auld, Perspectives in Science, 2014; Nyren, Analytical Biochemistry, 1987; Carbone et al., PNAS, 2017, https://doi.org/10.1073/pnas.1710358114). The buffers used in the bilayer assays is the same as all in vitro experiments. Importantly, 1 mg/mL BSA is well below the concentrations used to induce molecular crowding (typically 100 mg/mL or higher).Although our bilayers are a phospholipid bilayer, they do not mimic the lipid composition of the plasma membrane. We agree that the role of specific phospholipids, especially Phosphatidylinositol 4,5-bisphosphate (PIP2) and Phosphatidylinositol 3,4,5bisphosphate (PIP3) is of particular interest to integrin adhesions as FAK and full-length N-WASP can directly interact with PIP2 and kindlin directly interacts with PIP3. To address the Reviewers’ questions about the role of PIP2, we did try to make supported bilayers containing 3% PIP2 using our standard approach. However, these bilayers were not uniform and were immobile. Studies have shown that PIP2 lipids in supported bilayers are rapidly immobilized due to interactions with the underlying glass surface (https://dx.doi.org/10.1021/jacs.0c03800). We are actively developing new technical approaches to enable to incorporation of PIP2 into fluid lipid bilayers. However, this will involve careful optimization and validation of a new technical approach and is better suited for a separate, PIP2-focused study/methods paper.

Our data suggest that phosphoinositides are not explicitly required for macromolecular assembly at integrin adhesions. However, we predict that including PIP2 or PIP3 phospholipids would increase protein density at the membrane through direct binding to FAK (PIP2), N-WASP (PIP2) and kindlin (PIP3), likely promote clustering at lower protein concentrations. If so, this would represent a novel mechanism to control LLPS of membrane-associated condensate systems. Because of this potential, Dr. Case is developing new technologies to examine phosphatidyl inositols in her recently inaugurated independent lab. To address these issues in the revised manuscript we have elaborated on the potential contribution of phosphoinositides towards nascent adhesion assembly and/or focal adhesion maturation in the discussion.

We now include an expanded discussion about phosphoinositides to highlight these possibilities:

“Furthermore, many IAC-associated proteins can interact with negatively charged phosphoinositides. FAK and N-WASP preferentially bind PI(4,5)P_2_ via basic amino acids (Goni et al., 2014; Papayannopoulos et al., 2005), while kindlin preferentially binds PI(3,4,5)P_3_ via its PH domain (Liu et al., 2011). We find that phosphoinositides are not necessary to reconstitute clusters on phospholipid bilayers that resemble nascent adhesions. Consistent with our observations, reduction of PI(4,5)P_2_ synthesis at IACs in cells leads to the formation of small IACs that contain paxillin and kindlin, but lack talin and vinculin (Legate et al., 2011). Furthermore, talin has been shown to directly interact with and activate PIP5K1C (Di Paolo et al., 2002), and PIP4K2A was identified in as a constituent of IACs whose local concentration potentially changes during adhesion assembly and growth (Horton et al., 2015). Thus, PI(4,5)P_2_ may be locally generated during IAC formation or maturation, and additional interactions between phosphoinositides and cytoplasmic adaptor proteins could be important to reduce the threshold concentration of phase separation during nascent adhesion assembly or subsequent adhesion maturation and growth (Mitchison, 2020).”

9) The mutant p130Cas and FAK proteins need to be characterized and tested in phase separation experiments in vitro.

We performed the corresponding in vitro measurements in Figure 1b-c and Figure 6. While we did not purify Y15F p130Cas, we performed an analogous experiment in vitro using fully dephosphorylated p130Cas (which is functionally very similar to Y15F). We do not observe phase separation with unphosphrylated p130Cas (Figure 1b-c). Similarly, we found that purified FAK W266A failed to phase separate even at 5 µM concentration (assessed by turbidity, figure 6c). Since unphosphorylated Cas and W266A FAK fail to undergo phase separation, further in vitro characterization such as measuring PC or FRAP recovery are not possible (since droplets do not form). To make figure 6 more clear, we labeled the graphs in 6c and 6h with “in vitro”.

10) The authors conclude that 'that pCas- and FAK-driven phase separation provides an intracellular trigger for integrin clustering and nascent adhesion formation.' In contrast to this conclusion, the cellular studies investigate mature focal adhesion rather than nascent adhesion formation. Ideally, the cellular studies are carried out with cells undergoing isotropic spreading (around 5 min upon seeding, depending on cell type and ECM substrate) which is associated with the assembly (and turn over) of nascent adhesions.

Thank you for drawing our attention to this issue. The standard approach to examine nascent adhesion formation or adhesion dynamics is to perform live cell imaging with overexpressed Paxillin-GFP. However, we find that paxillin can enhance phase separation (Figure 2e-f), and thus overexpression would likely alter phase separation-dependent processes. Thus, we decided to use immunostaining to visualize cellular adhesions without altering endogenous protein levels. We initially performed experiments at a 20 min time point, but as the reviewer suggests this could confound effects due to nascent adhesion formation and subsequent maturation. Thus, we have repeated the experiments in the original Figure 7 at a 5-minute time point. We now report these data in a new Figure 7 and Figure 7 —figure supplement 1, and have moved the 20 minute timepoints to Figure 7 —figure supplement 2. At 5 minutes the MEFs were sufficiently adhered to enable fixation and immunostaining, but still appeared to be undergoing isotropic spreading. We also used TIRF microscopy instead of spinning disk microscopy to improve our resolution of nascent adhesions.

These new data were consistent with our initial 20-minute timepoint observations. That is, single knockout or knockdown of FAK or Cas only partially reduces the number of adhesions, while double knockdown leads to severe reduction in adhesion formation. With the 5 minute and 20-minute timepoints, we can more confidently conclude that the effects of Cas and FAK on adhesion number during cell spreading occur primarily due to defects in nascent adhesion formation.

Reviewer #2:In the present work the authors report the formation of condensates of adhesion complex proteins. They show in a series of experiments that the condensation of pCas and FAK appears to orchestrate the phase separation and the binding of the these complexes to membranes via kindlin.Although a bit sensational in the title the paper gives a nice proposal of activation of nascent adhesion formation. The title is a bit misleading since focal adhesions are not observed on the SLB and no actin filament binding is involved. A clear distinction between the here observed condensates and the complexes which are focal adhesion sites should be made throughout the manuscript. Maybe the condensates come closer to nascent adhesion sites.

We agree with the reviewer that we are reconstituting clusters that are similar to nascent adhesions. To make this clearer, we have tried to be more specific and consistent with our language throughout the paper. We were using the term focal adhesion to broadly refer to any integrin adhesion complex. We now refer to “integrin adhesion complexes (IACs)” to broadly refer to cellular integrin macromolecular complexes, “nascent adhesions” to refer to newly formed integrin adhesion complexes, and “mature focal adhesion” to refer to the stabilized, elongated integrin adhesion complexes associated with actin stress fibers. We hope it is clear from these changes that our goal in the study was to identify protein interactions involved in nascent adhesion assembly.

The interpretation of two distinct phase separation pathways working together to promote integrin clustering seems to be to be an overinterpretation and not entirely consistent with literature. It seems unlikely that the p130Cas+NcK^+^N-WASP would be involved in FA formation, as N-WASP is an actin nucleator and the actin layer in FAs is spatially separated from FAK (https://www.nature.com/articles/nature09621 this paper was cited, but no comment was made to that discrepancy). Which leads to the structure of the complex – which is certainly beyond the scope of the paper but still somehow important to argue a function of the formed aggregates.

While N-WASP is an actin nucleator, it is also a membrane-associated protein. Full length N-WASP binds to PIP2 phospholipids and is known to nucleate actin filaments specifically at the plasma membrane (Papayannopoulos et al., 2005; 10.1016/j.molcel.2004.11.054). N-WASP is also phosphorylated by FAK in an adhesion-dependent manner (Wu et al., 2004; 10.1074/jbc.M310739200) and colocalizes with phosphorylated Cas in lamellipodia (Zhang et al., 2014; 10.1242/jcs.134692), and these interactions regulate cell adhesion and spreading.

To our knowledge, no one has looked at the specific nanoscale architecture of nascent adhesions. iPALM experiments are biased towards large, mature adhesions, as confident localization requires measuring the position of 100s – 1000s of individual molecules within each adhesion. Thus, it is unknown when this layered architecture arises during adhesion formation and maturation. Evidence suggests that talin length determines the height of the FA-stress fiber interface (Liu et al., PNAS, 2015; https://doi.org/10.1073/pnas.1512025112). It is possible that axial stratification does not occur until talin starts stretching under mechanical force. While the initiation of the nanoscale organization is an interesting question, it is well beyond the scope of the current paper. We have included a more detailed discussion of adhesion nanoscale architecture, and how the condensates we observe may be modulated by force to generate mature focal adhesions, in the discussion:

“Mature IACs (called focal adhesions) are organized into vertical layers, with paxillin and FAK localized near the membrane and actin and actin binding proteins localized > 50 nm above the membrane (Kanchanawong et al., 2010). At mature focal adhesions, the plasma membrane and actin cytoskeleton provide an inherent polarity; the membrane is physically connected to the ECM through integrins on one side and actin generates forces on the other. These forces are transmitted across the components of the mature focal adhesion. While nascent adhesions may behave like an isotropic liquid condensate, the actin cytoskeleton could induce molecular reorganizations to produce order in the direction perpendicular to the membrane and generate a state more akin to a liquid crystal. While it is not known when this layered organization emerges during IAC assembly and maturation, talin plays a critical role in controlling the distance between actin and the plasma membrane (Liu et al., 2015). Additionally, talin undergoes force dependent conformational changes that extend the protein and expose additional vinculin binding sites (del Rio et al., 2009). Thus, force-dependent talin extension likely plays an important role in patterning the mature focal adhesion. Many other condensates interact with actin, and the layered organization of mature focal adhesions may be more common than previously appreciated (Beutel et al., 2019; Bresler et al., 2004; Zeng et al., 2016).”

Neither p130Cas, Nck or N-WASP are components of the consensus adhesome and are not found in nascent FAs.: https://www.ncbi.nlm.nih.gov/pmc/articles/PMC4663675/ The cited paper of Meenderink et al. 2010 states that p130Cas effect is most pronounced in FA disassembly. In this paper a pre-existing phosphorylation pattern is used to show complex formation with FAK. But it was shown that a FAK^+^Src complex leads to the phosphorylation of p130Cas, which is needed for complex formation (https://www.ncbi.nlm.nih.gov/pmc/articles/PMC2700377/). In summary, in the present study the protein system of Case et al. 2019 was used, basically only switching out the phosphotyrosine-protein, without really focusing on the context of FA formation process or if the second phase separation mix is even needed. It is not examined if only adding phosphorylated or dephosphorylated p130Cas to the FAK mix is sufficient for increasing the molecular condensates. This would be needed to claim real synergistic behavior – which is actually a still loosely used term here. I am even not sure if really just an increase of condensate size and protein content (fluorescence intensity) at 1µM with undefined stoichiometry is a strong point – or better how to relate this to the emergence of nascent adhesions – which have a well defined size range and are tightly controlled.

While the proteomics work leading to the development of the consensus adhesome is an immensely powerful resource, the techniques used to develop the consensus are not infallible. Mass spectrometry approaches are more likely to identify proteins that are highly concentrated, stable constituents of integrin adhesions. Lack of detection by these approaches does not *de facto* mean the protein does not play an important role in integrin adhesions. As the authors of that paper themselves make this point explicitly: “some well characterised IAC components were not enriched in all seven datasets (e.g. β3 integrin, FAK, kindlin, paxillin and talin) or were observed in the meta-adhesome but not the consensus adhesome (e.g. p130Cas and Src family kinases). These omissions may be due to cell-type-specific expression, cell-type-specific IAC maturation, protein abundance at IACs, preferential use of β1 integrin or non-specific detection in negative controls.”

Furthermore, Nck was detected in the Meta-Adhesome and has since also been identified by proximity ligase-based approaches (Dong et al., 2016; 10.1126/scisignal.aaf3572 and Chastney et al., 2020: https://doi.org/10.1083/jcb.202003038). While N-WASP was not detected by proteomics studies, it has been observed to colocalize with pCas in newly formed adhesions via fluorescence microscopy (Zhang et al., 2014; doi: 10.1242/jcs.134692). Furthermore, WASF2, a WASP family member that has 97% amino acid sequence identity to the N-WASP protein used in this study, was recently identified in Kindlin and Paxillin BioID experiments (Dong et al., 2016).

Nck and N-WASP are not the only multivalent adaptor proteins that interact with p130Cas. In cells, there are likely multiple adaptor proteins capable of interacting in a similar fashion with pCas. However, Nck and N-WASP are the adaptor proteins with strongest evidence showing potential role in nascent adhesion formation. We have edited the manuscript to include more of this nuance:

“Within the substrate domain, there are 15 YXXP motifs that, when phosphorylated, can bind to the SH2 domain-containing adaptor proteins, Crk-II and Nck (Iwahara et al., 2004; Pellicena and Miller, 2001; Schlaepfer et al., 1997). In cells, Cas is robustly phosphorylated on 10 of its YXXP motifs, and proper cell migration requires a minimum of four phosphorylation sites (Shin et al., 2004). CrkII contains two SH3 domains which can bind adaptor proteins such as SOS and C3G that contain multiple proline rich motifs (PRMs) (Birge et al., 2009). Similarly, Nck contains three SH3 domains which can bind adaptor proteins such as NWASP, SOS and Abl that contain multiple PRMs (Li et al., 2001). … Nck has been localized within IACs (Goicoechea et al., 2002; Horton et al., 2015), N-WASP colocalizes with Cas within the lamellipodia where nascent adhesions form (Zhang et al., 2014), and both Nck and N-WASP have been implicated in regulating cell adhesion to fibronectin (Misra et al., 2007; Ruusala et al., 2008). Thus, we chose to use Nck as the SH2/SH3 domain containing adaptor protein and N-WASP as the PRM containing adaptor protein for our studies (Figure 1a, Table S2).”

Regarding p130Cas, as the reviewer cites, Cas and FAK directly interact (Wisniewska et al., 2005), and FAK binding to the Cas SH3 domain helps recruit Src kinase to phosphorylate the Cas substrate domain (Hamasaki et al., 1996; Tachibana et al., 1997). However, the related kinase Pyk2 can also promote Src-dependent Cas phosphorylation (Astier et al., 1997), and Cas is robustly phosphorylated in FAK-/- MEFs (Hamasaki et al., 1996). Thus, the phosphorylation of Cas can occur by both FAK-dependent and FAK independent mechanisms. In this study, we do not attempt (or claim) to capture the detailed pathway by which Cas becomes phosphorylated, whether by FAK, Src or a complex of the two. Rather, we examine the consequences of that phosphorylation, which is the ability to assemble with other proteins in the adhesion structure and contribute to its formation. One strength of an in vitro approach is the ability to differentiate between the act of phosphorylation and the downstream consequences of phosphorylation. Since we phosphorylate Cas during purification and do not include ATP in our experiments, we know that once Cas is phosphorylated, kinase activity is not necessary for the macromolecular assembly we observe. While understanding the spatiotemporal dynamics of Cas phosphorylation during nascent adhesion assembly is important, to work out this pathway would be a project unto itself beyond the scope of the present work.

In summary, we believe that based on current literature data our reconstitution does contain a reasonable collection of molecules to support our claim that they capture important mechanistic aspects of nascent adhesion assembly.

It is not clear how the FRAP measurements are done. The condensates are freely diffusing (also rotational) in 3D solution which would make it difficult to obtain proper FRAP analysis. The stable fluorescence before bleaching need to be reported and contribution of recovery by rotational and translational diffusion of the particles themselves need to be experimentally excluded. The presented curves do not seem to look like a single exponential, and the rapid recovery at t=0 and almost no recovery for pCas and NcK bothers quite a bit.Unambiguous FRAP data and analysis would seem to be key for any claims on the existence of functional condensates.

As detailed in our reply to the next question, in our FRAP experiments the droplets are not freely diffusing, but rather are weakly attached to the PEG-coated slide surface. So they do not move laterally in solution during FRAP bleaching or recovery; it is a reasonable assumption, then, that they are not rotating either. We also bleach the entire droplet and integrate its total volume during recovery. Because of this, corrections for motion are not necessary. We have included additional supplemental figures demonstrating representative images of droplets during FRAP recovery and raw intensity measurements. We also include pre-bleach timepoints in these examples to illustrate stable fluorescence. We note that the intensity at t=0 does not reflect rapid recovery, but rather how much of the molecules were bleached relative to pre-bleach intensity. To more appropriately display our data, the average FRAP recovery curve has been normalized such that the yaxis represents the percent recovery (with normalized intensity of 0 at t=0). We now clarify this point in the methods:

“Quantification of droplet intensity was performed using ImageJ. The mean intensity inside the ROI was measured, and an identical sized region in the bulk solution was used to correct for photobleaching during imaging. Background and photobleaching corrected intensities were used for normalization.

Normalized Intensity = (Intensity – Intensity_postbleach_)/(Int_prebleach_ – Intensity_postbleach_).

Int_prebleach_ = average of three measurements in the ROI before 405 laser. Intensity_postbleach_ = intensity within the ROI immediately after 405 laser (at t=0).”

At the referee’s suggestion we have reanalyzed the recovery curves using bi-exponentials, and found that indeed these fit better than single exponentials with the exception of Nck. We now report both time constants in the Figures 1E and 2G, and include more detailed information about the fit in a new supplemental table (Table S4). With the exception of NWASP, most proteins show a percentage of immobile (non-recovering) molecules, although for some molecules we are unable to accurately determine the percent recovery. We and others have observed this for other phase separated condensates both in vitro (Nephrin condensates: Banjade & Rosen, eLife, 2014; cGas condensates: Du and Chen, Science, 2020; Post-synaptic densities: Zeng et al, Cell, 2018) and in cells (Nephrin condensates: Kim et al., Mol. Biol. Cell, 2019; P-bodies: Xing et al., eLife, 2020; Stress Granules: Niewidok et al., 2018; doi: 10.1083/jcb.201709007; cGas condensates: Du and Chen, Science, 2020; Post-synaptic densities: Ying et al, Journal of Neuroscience, 2010). The functional significance, if any, of the immobile fraction remains unknown in virtually all cases.

In fact, the immobile fractions that we observe (between 60 – 45% immobile for different proteins) in our reconstitution are reasonably close to those observed for the same molecules in cellular IACs. For example, one recent study found that 40% of FAK molecules, 35% of paxillin molecules, and 30% of Cas molecules are immobile within IACs (Stutchbury et al., 2017). This behavior indicates that many condensates, including IACs, contain both dynamic and immobile elements; the physical and function relationships between these elements is an open question in the field.

Additional specific concerns:- It is unclear to me how the FRAP measurements are done? The condensates are freely diffusing (also rotational!) in 3D solution which would make it difficult to obtain proper FRAP analysis. The stable fluorescence before bleaching need to be reported and contribution of recovery by rotational and translational diffusion of the particles themselves need to be experimentally excluded. The presented curves do not look like a single exponential, and the rapid recovery at t=0 and almost no recovery for pCas and NcK bothers quite a bit.Unambiguous FRAP data and analysis seem to be key for any claims on the existence of functional condensates.

FRAP measurements of in vitro condensates is a standard approach commonly used by many labs to assess general material properties of condensates (Alberti et al., 2020 10.1016/j.cell.2018.12.035). To better illustrate our FRAP data, we have now included additional supplemental figures with representative individual FRAP measurements, including images of the droplet during recovery and raw fluorescence intensity measurements (Figure 1 —figure supplement 5 and Figure 2 —figure supplement 3). We note that the intensity at t=0 does not reflect rapid recovery, but rather how much of the molecules were bleached relative to pre-bleach intensity. To more appropriately display our data, the average FRAP recovery curve has been normalized such that the y-axis now represents the percent recovery (with 0% recovery at t=0).

We bleach the entire droplet (~5-micron diameter), and the droplets do not significantly move over the course of our experiment. We observe consistent droplet intensity in unbleached droplets during timelapse imaging of similar duration, making us confident that movement or rotation of the droplet is not greatly affecting our fluorescence intensity measurements (Over the course of a control timelapse, the average droplet intensity = 4252 with a standard deviation of 64, in the representative graph in Author response image 5).

**Author response image 5. sa2fig5:** 

In the revised manuscript, we include three pre-bleach intensity measurements in the new representative FRAP graphs (Figure 1 —figure supplement 5 and Figure 2 —figure supplement 3). We have also included the fit of the data on our graphs in Figure 1e and 2 g. We also include a new supplemental table with detailed information about the fits (Table S4). We have added a more nuanced discussion of the FRAP data, including more discussion of the population of molecules that does not recover.For example, we now include the following discussion of FRAP data:

“We found that pCas, Nck and N-WASP rapidly exchanged between droplets and bulk solution (Figure 1e; Table S4; pCas t_1/2_ fast = 2 s; pCas t_1/2_ slow = 39s; Nck t_1/2_ = 8 s; N-WASP t_1/2_ fast = 1 s; N-WASP t_1/2_ slow = 37 s ), although a fraction of Nck and pCas molecules did not recover in the 100 s timeframe of the experiment, suggesting a slower phase of recovery and/or an immobile fraction (Table S4). In cells, IAC-associated proteins, including Cas, often contain a population of fast exchanging molecules (t_1/2_ < 30 s) as well as an immobile fraction (up to 50% immobile for some proteins; Table S1). Thus, similar to cellular IACs, droplets exhibit liquid-like material properties with some solid-like elements as well.”

– In Figure 1b, the authors show the solution turbidity as a function of the concentration of pCas. Could the authors comment on why the turbidity increase from 0nM to 50nM and decrease from 50 nM on. Do the aggregates disappear or is this a different scattering regime? If they disappear, why are the microscopy measurements work at 1µM? why does the solution turbidity of pCas system does not decay above 100 nM thus a different trend than FAK^+^Paxillin system (Figure 2d)? Does the difference in multivalency matter or explain the differences?

In previous work (Yoshizawa et al., Cell, 2018; Cinar and Winter, SciReports, 2020; Schuster et al., PNAS, 2020; Aleberti et al., Cell 2019) we and others have observed a strong correlation between turbidity values and the number and size of droplets observed by microscopy. Thus, turbidity is generally used as a measure of the total summed volume of all droplets in solution—essentially, the degree of phase separation in the system. So we believe that the bell-shaped curve in Figure 1b reflects increasing, and then decreasing phase separation as pCas concentration increases. The microscopy measurements were done with all three components at 1 µM (rather than just N-WASP, as in Figure 1b), where droplet formation is robust.

This behavior is expected for multi-component phase separating systems based on modular domain interactions (Li et al., Nature 2012; Case et al., Science 2019). Essentially, the relative concentrations of the domains must be balanced in order to produce oligomerization and consequent phase separation. When the concentrations become imbalanced as one component increases to large excess, the systems titrate out of the 2phase regime. As the referee intuits, this behavior is modulated by the valence of the interacting species, and differences in valence (e.g. pCas system vs FAK system), and also molecular drivers of phase separation (heterotypic interactions for pCas vs heterotypic + homotypic for FAK) can produce different behaviors. In the pCas system, pCas has 6 Nck SH2 binding sites and Nck has only 1 SH2 domain + 3 SH3 domains. Maximum phase separation likely occurs at an optimal ratio of the three components that enables a highly interconnected, oligomerized network. It is interesting that we observe highest phase separation near the physiological concentration of Cas (70 nM).

– The pCas system in Figure 4b should have 1 µm of kindlin, or not? It seems it is missing from the caption.

Yes everything in 4B should have kindlin (except control). This was an error in our figure legend, and has been corrected.

Figure 3: In this figure as well as in the text it should be clearer, that "pCas" or "FAK" stands for the whole condensation mix from above This should be explicitly stated and the color code introduced in 3a used in the subsequent mentions of theses protein mixes.

Thank you for these helpful suggestions. We now refer to “pCas mix” “FAK mix” and “pCas + FAK mix” to distinguish these terms from the individual proteins. We have also colorcoded Figure 3e-h and Figure 4b-I to match the color scheme introduce in Figure 3a. We hope these changes make our experiments and text clearer.

E There is a massive difference in Kindlin Intensities between pCas-droplets and FAK-droplets, but F shows about the same partition coefficient?

Both example images fall within the range of what we observe in the quantification shown in Figure 3f. However, to make the figure more intuitive, we have selected images more representative of the population average to display in Figure 3e.

Figure 4: No, quality control of SLB is presented, as no fluorescent lipids are present in the SLBs, hence no images shown of SLB or FRAPs of SLB. SLBs are formed in 96 wells which makes the essential washing steps difficult. Well cleaning protocol seems to be insufficient to get clean SLBs from what is commonly state of the art. Control measurements on the quality of the lipid layer needs to be provided (e.g. FRAP of lipids). As a matter of fact such configuration could also help to perform the FRAP measurements of the condensates.

Our lab and others have been using this approach to successfully generate fluid, POPC bilayers for many biophysical and cellular studies (See the detailed methods papers we follow, now cited in the manuscript: Su et al., 2017; https://doi.org/10.1007/978-1-49396881-7_5). While laborious, we do wash 96-well plates thoroughly, well by well. Additionally, we have performed many quality-control experiments throughout the course of these studies that are not reported. In fact, we find that integrin clustering does not occur if the bilayer is poorly made or immobile. Prior to any experiment, we confirm the fluidity and uniformity of the bilayer by imaging the distribution and dynamics of His_10_integrin. Bilayers must exhibit t_1/2_ recovery < 10 s, indirectly assessed by His_10_-integrin FRAP, to be used for clustering experiments. For clustering experiments, we do not include fluorescent lipids in the bilayer, as this could potentially change the chemistry of the bilayer. However, we frequently make bilayers with fluorescent lipids as a qualitycontrol step, to confirm our methods generate fluid, uniform bilayers.

To address the reviewer’s concerns, we included a new supplemental figure (Figure 4 —figure supplement 1) with data characterizing the bilayer with fluorescent lipids. We include 1% PC-NBD (a fluorescent phospholipid with excitation/emission at 464/531 nm) to directly visualize the lipid bilayer. While NBD is less photostable than Alexa fluorophores, we can correct for photobleaching that occurs during the timelapse. We consistently generate bilayers with FRAP t_1/2_ recovery < 10 s for a 5 µm diameter bleach ROI. This is now specifically described in the methods (under “Reconstitution of Supported Phospholipid Bilayers”):

“Prior to any experiment, the fluidity of bilayers was indirectly assessed by imaging integrin Alexa-488. We photobleached a 5-micron region, and experiments were only performed if FRAP t_1/2_ < 10 s. To confirm that our methods consistently give fluid, uniform bilayers, we also assessed bilayer fluidity directly in bilayers containing 1% PC-NBD, a fluorescent lipid (Figure 4, supplement 1).”

Surprisingly, and contrary to most reports, no PIP2 is required in the membrane to mediate the adhesion of the complexes to the membrane.

To clarify one point, in our system the integrin tail is attached to the bilayer through an Nterminal His_10_-tag, which binds with high affinity to the 2% Ni-NTA lipid in the bilayer. So the mechanism by which the condensates associate with the bilayer is through binding the membrane-attached integrin.

This being said, as described above, we anticipate that the adhesion of the condensate to the membrane would be stronger if we had incorporated PIP_2_ or PIP_3_ into the bilayer as well. This is because FAK and full-length N-WASP can directly bind PIP_2_ and kindlin directly binds PIP_3_. In cells, it may be that cooperative binding to phosphatidylinositols and integrin tails is needed for condensates to be bright enough to readily observe. To address the Reviewers’ questions about the role of PIP2, we did try to make supported bilayers containing 3% PIP2 using our standard approach. However, these bilayers were not uniform and were immobile. Recent studies have shown that PIP2 lipids in supported bilayers are rapidly immobilized due to interactions with the underlying glass surface (https://dx.doi.org/10.1021/jacs.0c03800). We are actively developing new technical approaches to enable to incorporation of PIP2 into fluid lipid bilayers. However, this will involve careful optimization and validation of a new technical approach and is better suited for a separate, PIP2-focused study/methods paper.

Alternatively, PIP_2_ may be needed primarily in later stages of adhesion formation/maturation. In cells, depletion of PIPKIγ from IACs (to prevent local PIP2 synthesis) lead to the formation of small IACs that contain paxillin and kindlin, but lack talin and vinculin (Legate et al., 2011). This is consistent with our findings that phosphoinositides are not required for macromolecules (including paxillin and kindlin) to assemble and cluster integrin tails in vitro, and our assertion that these condensates may best represent nascent adhesions. To our knowledge, no cellular studies have definitively investigated the specific timing of PIP2 generation during nascent adhesion assembly and focal adhesion growth. Talin has been shown to directly interact with and activate PIP5K1C (Paolo et al., 2002) and PIP4K2A was identified in as a constituent of integrin adhesion complexes whose local concentration potentially changes during adhesion assembly and growth (Horton et al., 2015). Thus, while PIP_2_ may influence adhesions by binding to adaptor proteins, adhesions likely also influence the local concentration of PIP_2_. This could positively reinforce adhesion formation and/or growth. We have elaborated on the potential contribution of PIP_2_ interactions towards nascent adhesion assembly and/or focal adhesion maturation in the discussion:

“Furthermore, many IAC-associated proteins can interact with negatively charged phosphoinositides. FAK and N-WASP preferentially bind PI(4,5)P_2_ via basic amino acids (Goni et al., 2014; Papayannopoulos et al., 2005), while kindlin preferentially binds PI(3,4,5)P_3_ via its PH domain (Liu et al., 2011). We find that phosphoinositides are not necessary to reconstitute clusters on phospholipid bilayers that resemble nascent adhesions. Consistent with our observations, reduction of PI(4,5)P_2_ synthesis at IACs in cells leads to the formation of small IACs that contain paxillin and kindlin, but lack talin and vinculin (Legate et al., 2011). Furthermore, talin has been shown to directly interact with and activate PIP5K1C (Di Paolo et al., 2002), and PIP4K2A was identified as a constituent of IACs whose local concentration potentially changes during adhesion assembly and growth (Horton et al., 2015). Thus, PI(4,5)P_2_ may be locally generated during IAC formation or maturation, and additional interactions between phosphoinositides and cytoplasmic adaptor proteins could be important to reduce the threshold concentration of phase separation during nascent adhesion assembly or subsequent adhesion maturation and growth (Mitchison, 2020).”

Addressing these various possibilities in vitro and in cells will require non-trivial technical innovations and is beyond the scope of our current study. We believe that these open questions do not detract from our findings, but simply say that there are additional aspects of the system to explore beyond this initial report.

In methods section the written name of the main lipid is DOPC, whereas the given abbreviation is POPC.

We apologize for this typo in the methods section. We used POPC, not DOPC, for all experiments. We have corrected the methods section and added more detailed information on the lipids used in the resource table.

FAK concentration was lowered, because there was unspecific binding to NiNTA, which is a bit worrisome for the interpretations and comparisons. Increased salt concentration could prevent unspecific binding, but might alter protein condensate dynamics at the same time. Added low concentration (<100mM) of imidazole may help here to prevent unspecific Ni-binding.

After the initial observation that FAK was interacting with Nickel, we screened many different buffers to reduce this effect (Assessed with Ni-NTA resin binding). The only addition that reduced the FAK-Ni interaction was 20 mM or 40 mM Imidazole. However, 20 mM Imidazole also disrupted the attachment of His_10_-Integrin to the bilayer (assessed with microscopy in Author response image 6):

**Author response image 6. sa2fig6:** 

From these experiments, we concluded that using His_10_-integrin (instead of His_8_-integrin) in combination with 200 nM FAK was the most effective approach to ensure integrin remained attached to the bilayer while reducing the effect of nonspecific FAK-Ni interactions on the observations.

**Author response image 7. sa2fig7:** 

Figure 5: In humans, temperature normally varies around 1 {degree sign}C and pH varies 0.1. That means the observed different conditions are far from physiological. It seems a bit far to explain differences in focal adhesion size with the protein condensate properties, as at these conditions the phosphorylation state as well as the lipid composition should be massively altered: https://www.molbiolcell.org/doi/10.1091/mbc.e12-04-0296

The main goal of Figure 5 is to use environmental perturbations to directly compare in vitro droplets and integrin adhesions, even if these perturbations are not physiologically relevant. Such non-physiologic treatments are widely used in cellular experiments—e.g. heat shock, osmotic shock, various drug or small molecule treatments. It is in this spirit that we perform these experiments. We specifically look at shorter timepoints (10 min) to increase the likelihood that effects are due to phase separation (which should be a fast response) and not cellular responses that rely on slower changes in transcription, translation, or other longer-term adaptations. The specific paper the reviewer cites is looking at timepoints hours (10-24 hr) after temperature shock, which is a very different experiment from the 10 minute treatments we use here.

Furthermore, we disagree that these pH differences are not physiological, since the local pH at FAs has been observed to vary in cells between ~ 7.1 – 7.7 due to the local concentration and activation of NHE1, a Na/H^+^ exchanger (Ludwig et al., 201310.1002/jcp.24293). Thus, of the possible environmental perturbations that can occur in cells, regulated fluctuations in pH are the most likely to be relevant to integrin adhesions.

Figure 6: The paper is very focused on an effect of phase separation on forming FAs, but the presented in vivo data looks at FAs at a static time point and does not take into account the FA dynamics.

The standard approach to examine integrin adhesion dynamics is to perform live cell imaging of Paxillin-GFP. However, we find that paxillin can enhance phase separation (Figure 2e-f), and thus overexpression of paxillin would likely alter phase separation dependent processes. Thus, we decided to use immunostaining to visualize adhesions and infer information about adhesion formation without altering endogenous protein levels. We initially performed experiments at a 20 min time point, but this could confound effects due to adhesion formation and adhesion maturation.

Thus, taking the suggestion of reviewers, we have repeated the experiments at a 5-minute time point. At 5 minutes the MEFs were sufficiently adhered to enable fixation and immunostaining, but still appeared to be undergoing isotropic spreading. We also used TIRF microscopy instead of spinning disk microscopy to improve our resolution of nascent adhesions. With our new data, we now have a 5 min (Figure 7b-c) and a 20 min (Figure 7 —figure supplement 2) snapshot of endogenous integrin adhesions during cell spreading, and find similar behavior in both. This new data strengthens our conclusion that Cas and Fak are required for integrin adhesion assembly.

[Editors’ note: what follows is the authors’ response to the second round of review.]

Essential Revisions:1) The reviewers request experiments that show whether the membrane and/or integrins influence the formation of condensates seen by TIRF in Figure 4, or if the condensates would also form in absence of integrin/membrane (ie, comparable to in solution). To test this questions the existing quantifications in Figure 4 should be compared with (a) quantifications of condensates on membranes without integrin tails and (b) quantification of condensates without integrin tails and without lipids (i.e. on the glass surface or some inert coating).

We have performed the requested experiment and the results are now included in Figure 4f-g. This experiment was extremely insightful, revealing that integrin specifically increases the rate of droplet appearance in the TIRF field compared with an empty lipid bilayer. The effect of integrin was even more pronounced when we repeated this experiment with physiological protein concentrations (70 nM pCas, 180 nM Nck (15% Alexa647 labeled), 140 nM N-WASP, 60 nM Kin, 60 nM Paxillin, 40 nM FAK). Thus, while condensates are observed in the TIRF field in the absence of integrin (especially with higher 1 μM protein concentrations), we also observe that integrin specifically increases condensate formation, likely through increasing the rate of nucleation on the membrane.

2) The reviewers are not convinced that simply counting nascent adhesions provides a good correlation for the in vitro experiments with purified components. It would be better to measure partitioning between cytosol and nascent FAs. This should be done for FAK-WT vs FAK-W266A, Cas-WT vs Cas-Y15F, and should be straight forward with the available images.

We have reanalyzed the images from these figures to measure paxillin partitioning in adhesions. These data are now in Figure 6g (Cas+/+, Cas-/-, Cas WT, CasY15F) and Figure 7g (FAK +/+, FAK-/-, WT FAK, FAK W266A).

3) in vitro, the components of the pCAS or FAK mix increase the partitioning of other components of the same mix into condensates. To correlate this with cells, partitioning of paxillin into nascent adhesions of FAK-/- , FAK-WT (and perhaps FAK-W266A) cells should be measured, and also partitioning of N-WASP or Nck into adhesions of Cas-/-, Cas-WT (and perhaps Cas-Y15F) cells.

With new in vitro measurements (see point 4 below), we find that paxillin partitioning decreases in pCas+FAK droplets with *both* FAK W266A and Cas Y15F proteins, likely because Cas interacts with both FAK and paxillin in addition to Nck. Thus, we can address this question by correlating paxillin partitioning in vitro and in cellular experiments. These comparisons can be found in Figure 6 (Cas Y15F) and Figure 7 (FAK W266A), and indeed show parallel responses in vitro and in cells to these perturbations.

We did also try to measure Nck partitioning into Cas-/- cells. However, this proved to be a confounding experiment. We did not have reliable antibodies to measure endogenous N-WASP or Nck at adhesions, so we transiently expressed Nck-mCherry in cells and labeled adhesions with paxillin antibodies. However, we found that cells overexpressing Nck often failed to spread and form adhesions, suggesting the Nck overexpression was somehow inhibiting adhesion formation or spreading. See two examples of this phenotype in Cas +/+ MEFs in Author response image 8:

**Author response image 8. sa2fig8:** 

Despite this, we were able to find some cells expressing Nck-mCherry that formed adhesions (imaged in Author response image 9 and (Author response image 1) with spinning disk microscopy). We do see Nck colocalized within Paxillin-labeled adhesions in cells with WT Cas but not with Y15F Cas, although Nck has a higher cytosolic localization and lower adhesion partition compared with paxillin in WT Cas cells.

**Author response image 9. sa2fig9:** 

**Author response image 10. sa2fig10:** 

However, due to the unexpected and not understood effect of Nck expression on cell spreading and/or adhesion formation, we did not include these data in the manuscript, as they would require analysis of a small subpopulation of cells, which we feel are not reliable.

4) The reviewers also request phase separation experiments with mutant Cas and FAK. It is important to confirm also in vitro that condensates are smaller when these mutations are introduced into the proteins. Cas carries 15 mutations, and this experiment will exclude at least in part that the cellular effects occur via phase separation-independent mechanisms.

We have included new in vitro experiments with Cas Y15F. Just like unphosphorylated Cas, CasY15F did not undergo any phase separation when combined with Nck and N-WASP, which we have demonstrated with both solution turbidity (Figure 6a) and microscopy (Figure 6 —figure supplement 1). Similarly, FAK W266A does not undergo any phase separation on its own (Figure 7a, Figure 7 —figure supplement 2). We have also quantified the effects of mutant Cas and FAK on pCas+FAK droplets, which more specifically mirror the perturbations we do in cells. Both Cas Y15F and FAK W266A reduce phase separation in the Cas+FAK mix and significantly decrease paxillin partitioning into droplets. These new experiments provide a more direct correlation between in vitro experiments and cellular data.